



# An evaluation of the regional distribution and wet deposition of secondary inorganic aerosols and their gaseous precursors in IFS-COMPO cycle 49R1

Jason E. Williams[1], Swen Metzger[2], Samuel Rémy[3], Vincent Huijnen[1] and Johannes Flemming[4]

R&D Weather and Climate Modeling, Royal Netherlands Meteorological Institute, De Bilt, the Netherlands
ResearchConcepts Io GmbH, Freiburg, Germany
HYGEOS, Lille, France
European Centre for Medium-Range Weather Forecasts, Bonn, Germany

Correspondence to : Jason Williams (jason.williams@knmi.nl)

**Abstract**

Secondary Inorganic Aerosol (SIA) constitutes a considerable fraction of total particulate matter exposure, making it an important component of any atmospheric composition and air quality forecasting system. The subsequent loss of SIA to the surface, via both dry and wet deposition, determines the exposure time for humans and the extent of damage imposed on sensitive ecosystems due to increased surface acidity. This study provides a description and evaluation of recent updates to aerosol production, scavenging, and wet deposition processes in the global IFS-COMPO chemical forecasting system, used within the Copernicus Atmosphere Monitoring Service. The implementation of the EQSAM4Clim simplified thermodynamic module in IFS-COMPO cycle 49R1 alters the phase transfer efficiency of SIA precursor gases (sulphur dioxide, nitric acid, and ammonia), which significantly affects particulate SIA concentrations by modifying the fraction converted into aerosol form. Comparisons with surface observational data from Europe, the U.S., and Southeast Asia during 2018 indicate reductions in the global annual mean bias for both sulphates and nitrates. Updating the IFS-COMPO model to cycle 49R1 increases the burden and lifetime of sulphate and ammonium particles by one-third. Coupling EQSAM4Clim with IFS-COMPO improves the representation of ammonia-ammonium partitioning across regions, while the effect on sulphate is minimal. For nitric acid and nitrates, the phase partitioning is also significantly altered, with lower particulate concentrations leading to an excess of gas-phase nitric acid and an associated improvement in surface nitrate predictions. The impact on total regional wet deposition is generally positive, although sulphates in the U.S. and ammonium particles in Southeast Asia are strongly influenced by precursor emission estimates. Overall, these results provide confidence in the ability of IFS-COMPO cycle 49R1 to deliver accurate global-scale deposition fluxes of sulphur and nitrogen.



## 1. Introduction

Secondary Inorganic Aerosols (SIA) are found throughout the troposphere, where their concentrations depend on temperature (T), relative humidity (RH), and the concentrations of inorganic precursor gases, namely water vapor ($H_2O$), sulfur dioxide ($SO_2$), ammonia ($NH_3$), and nitric acid ($HNO_3$). High concentrations of SIA contribute to total particulate matter, accumulating in size bins of 1.0 μm ($PM_{1.0}$), 2.5 μm ($PM_{2.5}$), and 10 μm ($PM_{10}$) (Liu et al., 2022), and have detrimental effects on both human health and visibility (Sharma et al., 2020; Ting et al., 2021). The main types of SIA are ammonium sulfate (($NH_4$)$_2SO_4$), ammonium bisulfate ($NH_4HSO_4$), and ammonium nitrate ($NH_4NO_3$). Once formed, sulfates are very stable and deposit to the surface, while $NH_4NO_3$ is more unstable and can decompose back into precursor gases (Feick and Hainer, 1954), depending on T and RH. These particles can be transported out of source regions, influencing air quality in neighboring countries (e.g., Vieno et al., 2014; Chang et al., 2022). Anthropogenic activity significantly contributes to SIA formation through the emission of $NO_x$ (oxidized nitrogen in the form of NO and $NO_2$), $NH_x$ (reduced nitrogen), and $SO_2$. There has been a general trend of decreasing sulfur (S) and nitrogen (N) emissions in the EU, U.S., and China (Tørseth et al., 2012; Aas et al., 2019; Benish et al., 2022; Jiang et al., 2022), leading to an increasing fraction of SIA being $NH_4NO_3$. This results in a decrease in the lifetime of SIA due to the increased meteorological instability of $NH_4NO_3$ (e.g., Williams et al., 2015; Metzger et al., 2002, 2006), reducing the potential for long-range transport out of source regions (He et al., 2018).

At RH values above 50%, most SIA aggregates water and exists in a deliquescent state. At high RH, SIA formation is enhanced (Gao et al., 2020); therefore, under constant or changing emissions, SIA is likely to become more ubiquitous in a warming atmosphere. The hygroscopic growth of SIA alters its optical properties (scattering and absorption) and interactions with gas-phase trace species via changes in pH (e.g., Jayne et al., 1990; Shi et al., 2018). The concentrated salt solution produced typically has higher ionic strength than cloud droplets, with pH values ranging from -1 to 6 (Ault, 2020). The high solubility of SIA leads to scavenging into aqueous aerosols and clouds, which is a dominant loss mechanism. This has implications for the acidification of sensitive ecosystems and increased eutrophication due to high nitrogen loading in inland water bodies, potentially exceeding critical loads for vegetation (e.g., Sun et al., 2020). Nitrogen loading also enhances carbon uptake by land (Holland et al., 1997; Reary et al., 2008). Once dissolved in solution, SIA dissociates efficiently into its ionic constituents (e.g., nitrate ($NO_3^-$), ammonium ($NH_4^+$), and sulfate ($SO_4^{2-}$)), which are then deposited on land during precipitation events.

There are distinct differences in the primary source terms for various SIA species. For $NO_x$ and $NH_x$ species, particle formation is sensitive to resident gas-phase precursors, temperature, and RH in the absence of aqueous-phase droplets. For $SO_4^{2-}$, production occurs almost exclusively in the aqueous phase after $SO_2$ is scavenged into clouds and fog, with cumulative oxidation rates dependent on the prescribed pH in solution. Recent studies highlight the importance of accurately representing cloud pH for determining long-term trends in $SO_4^{2-}$ production (Thurock et al., 2019; Myriokefalitakis et al., 2022). The representation of acidity in tropospheric aerosols and clouds varies significantly across large-scale atmospheric models. The simplest approach is to assume a fixed cloud water pH between 5.0 and 5.6, effectively representing the impact of dissolved $CO_2$. A more accurate approach incorporates the influence of other dissolved species that either acidify (e.g., $HNO_3$, $H_2SO_4$) or buffer (e.g., $NH_3$) solution pH once scavenged through irreversible uptake. This is the method adopted in the Integrated Forecasting System with atmospheric composition extension (IFS-COMPO) for both cloud and precipitation. Other $SO_4^{2-}$ production mechanisms involving compounds such as methyl hydroperoxide ($CH_3OOH$) are of secondary importance (Myriokefalitakis et al., 2022). More buffering by $NH_3$ accelerates conversion rates, as the reaction of $HSO_3^-$ is slower than that of $SO_3^{2-}$ (Warneck, 1991).

A major loss mechanism for SIA is wet deposition through precipitation. Previous global tropospheric modeling studies have focused on the temporal accuracy and annual deposition totals at continental scales for $NH_x$ and $SO_x$ (Zhang et al., 2012; Kanikadou et al., 2016; Ge et al., 2021). Multi-model intercomparison studies have also examined variability across different models and identified the main assumptions causing such differences (Dentener et al., 2006; Bain et al., 2017; Tan et al., 2018). The accuracy of any model in capturing wet deposition depends on the precursor emission inventory's accuracy, the distribution of cloud liquid water content (defining cloud Surface Area Density, SAD), the formation and distribution of aerosol particles, phase transfer, and parameterizations for dry/wet deposition.



The IFS-COMPO model is a large-scale global model used for operational analyses and air quality forecasts (Peuch
et al., 2022; Williams et al., 2022; Rémy et al., 2024) as part of the Copernicus Atmosphere Monitoring Service
(CAMS). This service provides forecasts and reanalyses of trace gases and aerosols to inform national service
providers and policymakers. It delivers chemical/aerosol forecast products, including ozone ($O_3$), nitrogen dioxide
($NO_2$), $SO_2$, $PM_{2.5}$, $PM_{10}$, and aerosol optical depth. One of the recent updates to IFS-COMPO focused on reducing
biases and improving correlations for aerosol products (Rémy et al., 2024). As a result, acidic deposition and
nitrogen loading outputs from the model will likely improve as PM distribution accuracy increases, fostering the
development of future IFS-COMPO products.
This paper analyzes the regional performance of IFS-COMPO CY48r1 and CY49r1 in terms of surface
distributions of nitrogen and sulfur gaseous precursors for SIA, along with the associated particle concentrations
and distributions, evaluated against ground-based observation networks. Special emphasis is placed on the
application of the latest EQSAM4Clim updates (Metzger et al., 2024) in the global chemical forecasting model
IFS-COMPO CY49r1. This work complements a recent evaluation of the performance of IFS-COMPO CY48r1
and CY49r1 and the impact of using EQSAM4Clim with respect to regional $PM_{2.5}$ distributions and aerosol optical
depth, presented in Rémy et al. (2024). The influence of these updates on regional wet and dry deposition terms is
also evaluated to assess improvements to both EQSAM4Clim and the deposition schemes. Section 2 provides
details of the IFS-COMPO simulations used, a brief description of the latest model updates, and the emissions
used. Section 3 describes the observational networks against which surface evaluations are performed for precursor
gases and resulting SIA particulates. Section 4 details the changes in regional surface concentrations of precursor
gases and associated particulates, along with regional annual mean statistics. Section 5 presents the comparisons
of annual mean wet deposition fluxes for Europe, the U.S., and Southeast Asia, and discusses improvements.
Finally, Section 6 offers further discussion and conclusions from our study. Additional supporting information is
available in the supplementary material.

### 2. Model description of IFS-COMPO versions

The IFS-COMPO global composition model (formerly known as C-IFS) is used for operational air quality analyses
and forecasts as part of CAMS. The modeling and data assimilation framework is regularly updated. Since July
2023, IFS-COMPO has been based on CY48R1, using recently updated chemical and aerosol components for
near-real-time simulations of atmospheric composition (https://www.ecmwf.int/en/elibrary/, last accessed
21.07.23; Rémy et al., 2022; Williams et al., 2022). These updates have been shown to reduce biases in key
products such as $O_3$ and $NO_2$ compared to previous cycles (Huijnen et al., 2016; Huijnen et al., 2019). In this study,
we perform simulations using CY48R1 and compare them against a version of IFS-COMPO based on CY48R1,
but with updates to the atmospheric composition components to be included in CY49R1, which will be operational
in November 2024 (Rémy et al., 2024). These updates aim to improve the aerosol component, wet deposition
scheme, and description of pH in clouds and aerosols by applying the EQSAM4Clim approach (Metzger et al.,
2016; Metzger et al., 2024; Rémy et al., 2024). For brevity, we provide only a brief description of the updates
made to the wet deposition parameterization and the implementation of EQSAM4Clim in IFS-COMPO, which
determines surface deposition fluxes. A more comprehensive description of the CY49R1 updates is provided in
Rémy et al. (2024), and details of the EQSAM4Clim thermodynamic module are found in Metzger et al. (2024).

### 2.1 Updates in IFS-COMPO CY49R1

The CY49R1 version of IFS-COMPO is built on the previous operational cycle (CY48R1) and includes eight
distinct aerosol types with multiple bins for size segregation: sea salt, desert dust, organic carbon, black carbon,
$SO_4^{2-}$, fine and coarse $NO_3^-$, $NH_4^+$, and secondary organic aerosol. For CY49R1, updates have been made to the
aerosol component, including modifications to the description and properties of desert dust and sea salt. These
changes impact the resident lifetimes and long-range transport of each aerosol species. Modifications to the aerosol
optics description have also been implemented, improving simulations of aerosol optical depth (AOD) and the
Ångström exponent compared to regional observations (Rémy et al., 2024). The gas-phase chemistry, photolysis,
and dry deposition are identical to those described in Williams et al. (2022).
In CY49R1, EQSAM4Clim is used to estimate the gas/particle partitioning of the $HNO_3$-$NO_3^-$ and $NH_3$-$NH_4^+$
systems and to provide an estimate of aerosol pH. The pH of aqueous solutions, aquated aerosols, and precipitation
is updated at each time step using the EQSAM4Clim approach, which accounts for additional cations ($Ca^{2+}$, $Mg^{2+}$,
$Na^+$, $K^+$), anions ($SO_4^{2-}$, $HSO_4^-$, $NO_3^-$, $Cl^-$), and their solute interactions, as comprehensively described in Metzger
et al. (2012, 2016, 2024). This replaces the original pH estimate, which was based on summing the contributions
from dissolved $CO_2$ and strong acids ($HNO_3$, $HSO_3^-$, $H_2SO_4$, $NO_3^-$, and methane sulfonic acid), buffered by





dissolved NH₃. The contributions to solution pH from dissolved formic and acetic acids (HCOOH and CH₃COOH,
respectively) are also now included in CY49R1, as they have been shown to influence cloud droplet pH (Shah et
al., 2020). This update impacts phase transfer, speciation, and the subsequent aqueous-phase oxidation of SO₂ in
cloud droplets, which affects SO₄²⁻ formation. The loss of gas-phase species such as H₂O₂ and the corresponding
formation of SIA particles are also affected. Note that both the original (CY48R1) and updated (CY49R1)
approaches account for the dominant gaseous contributions to solution pH, namely SO₂, HNO₃, and NH₃.
Consequently, the differences in cloud pH are generally smaller than the changes in aerosol pH.
Below-cloud scavenging of gaseous precursors is also affected by solution pH (e.g., Seinfeld and Pandis, 2006).
In CY48R1, fixed values for cloud pH were used over land (pH = 5.0) and ocean (pH = 5.6), providing only limited
variability in regions affected by both high and low emissions. In CY49R1, the pH calculation is now coupled with
resident trace gas and aerosol concentrations, improving consistency within IFS-COMPO and providing variable
scavenging rates dependent on tropospheric composition.
In CY48R1, the wet deposition routines for aerosols and chemistry were distinct, though both utilized a scheme
adapted from Luo et al. (2019) for operational use. To ensure a consistent approach between aerosol and trace gas
wet deposition, and to simplify code maintenance, these separate implementations have been merged into a unified
routine. This new routine now represents the wet deposition processes for both aerosols and chemical species and
is executed with either chemical or aerosol tracers as inputs. Similar to CY48R1 and previous versions, the routine
in CY49R1 is executed twice: once for large-scale precipitation and once for convective precipitation. For
convective precipitation, the assumed precipitation fraction has been standardized to 0.05 (whereas in CY48R1, a
value of 0.1 was used for chemistry scavenging and 0.05 for aerosol scavenging).
Additional upgrades have been made for aerosol wet deposition as follows: (i) The aerosol activation
parameterization of Verheggen et al. (2007) has been implemented, which estimates the fraction of aerosols
scavenged through in-cloud processes as a function of temperature. It applies to mixed clouds, specifically for
temperatures between the freezing point and 233 K. For temperatures above 0°C, the consistency of the parameters
determining the fraction of aerosols subject to in-cloud wet deposition with the Verheggen parameterization results
has been verified. (ii) For below-cloud scavenging of aerosol species, scavenging rates have been updated to better
reflect particle size dependency, as described by Croft et al. (2009). This update includes adjustments to the below-
cloud scavenging parameters, which describe the efficiency with which aerosols are removed by rain and snow,
depending on species and size distribution. A below-cloud scavenging model has also been implemented.
**2.2 Setup of model simulations**
The IFS-COMPO simulations used to evaluate the impact of the atmospheric composition upgrades proposed
for cycle 49R1 on tropospheric composition, precursor gases, particle distributions, and wet deposition terms
employ both IFS cycles CY48R1 and CY49R1. Here, CY49R1 refers to IFS-COMPO cycle 48R1, including the
proposed updates to the aerosol/chemistry modules for IFS cycle 49R1. The meteorological component remains
the same across simulations and corresponds to CY48R1. The simulations presented here cover the year 2018,
with a one-month spin-up period. The vertical resolution uses 137 individual model levels, and the horizontal
resolution is TL511, corresponding to approximately 0.4° x 0.4°. These experiments do not include data
assimilation of observations. Meteorology is initialized every 24 hours based on ERA5 reanalysis data, meaning
IFS-COMPO is run in a cyclic forecast mode. A 15-minute chemical time step is used to solve a modified version
of the CB05 tropospheric chemistry scheme (Williams et al., 2022), excluding active stratospheric chemistry for
efficiency. Three-hourly, three-dimensional global output is used for the analysis.
The details of the sensitivity experiments are summarized in Table 1. The CY48R1 reference simulation pertains
to the 48R1 version of IFS-COMPO, while the CY49R1 simulation is based on the version described in Rémy
et al. (2024). The CY49R1_NOE4C simulation is identical to the CY49R1 simulation, except that the
EQSAM4Clim module (Metzger et al., 2016) is deactivated. For future reference, the experiment identities on
the ECMWF Multiversion Asynchronous Replicated Storage system (MARS) are hylm (CY48R1), i3bw
(CY49R1_NOE4C), and i3ad (CY49R1). These three simulations use a configuration similar to those described
in Rémy et al. (2024) for evaluating particulate matter (PM).
**Table 1:** Definitions of the IFS-COMPO simulations used in this study.

| Simulation | Experiment ID | Comments |
|---|---|---|
| | | |



| CY48R1 | hylm | Reference CY48R1 model version. |
|---|---|---|
| CY49R1 | i3ad | As CY48R1, but with all composition modeling updates for CY49R1, particularly activating EQSAM4Clim in both aerosols and cloud droplets. |
| CY49R1_NOE4C | i3bw | As CY49R1, but with the EQSAM4Clim module deactivated. |


The emissions used in these configurations are taken from the CAMS_GLOB_ANT v5.3 dataset (Soulie et al.,
2023), with biogenic emissions from the CAMS_GLOB_BIO v3.1 dataset (Sindelarova et al., 2022;
http://eccad.aeris-data.fr) and biomass burning emissions from GFAS v1.2 (Kaiser et al., 2012). All emissions
are applied using the methodology described in Ye et al. (2021). Apart from biomass burning (BB) and $SO_2$,
emissions are applied in the lowest model level. Currently, the emission of dimethyl sulfide (DMS) is based on
a climatology, i.e., it is not coupled to sea surface temperature, which controls biogenic activity (Deschaseaux
et al., 2019). Additionally, direct production of $SO_4^{2-}$ and $HNO_3$ from hot shipping exhausts is not accounted
for.
**3. Observations**
For the evaluation of the regional distribution and concentrations of SIA precursor gases, as well as the associated
particle concentrations and deposition fluxes, we use data freely available from various observational networks.
Here, we provide only a brief description of the chosen networks.
For gas-phase precursors, we use in-situ measurements of $SO_2$ from the AirBase (Europe,
https://www.eea.europa.eu/, last accessed 12 Aug 2024), AirNow (U.S., https://www.airnow.gov/about-airnow/,
last accessed 12 Aug 2024), and the China National Environmental Monitoring Center (CNEMC,
https://www.cnemc.cn/) networks. Only rural background stations have been selected, and filtering has been
applied to the AirNow data to remove spurious high values that are not representative of rural background
conditions.
For $NH_3(g)$ in the U.S., we compare both weekly and yearly mean values derived from in-situ measurements taken
from selected stations participating in the Ammonia Monitoring Network (AMoN,
https://nadp.slh.wisc.edu/networks/ammonia-monitoring-network/, last accessed 12 Aug 2024), selecting 18
individual sites across the continent. No filtering has been applied to these measurements, as quality control has
been adopted from the provider.
For $HNO_3(g)$ in Europe and the U.S., we use data provided by the European Monitoring and Evaluation Programme
(EMEP, Torseth et al., 2012; https://ebas.nilu.no/, last accessed 12 Aug 2024) and the Clean Air Status and Trends
Network (CASTNET; https://www.epa.gov/castnet, last accessed 12 Aug 2024), respectively. For Southeast Asia,
data from the Acid Deposition Monitoring Network in East Asia (EANET, https://www.eanet.asia/, last accessed
14 Aug 2024) is used. However, no corresponding measurements of $NH_3(g)$ and $HNO_3(g)$ are available for the
Southeast Asia domain. For evaluating particle concentrations, we use available data from the CASTNET, EMEP,
and EANET networks for $SO_4^{2-}$ and $NO_3^-$.
For wet deposition totals, we use data from the same measurement networks as those for the gaseous precursors,
thus removing any potential differences introduced by spatial sampling that might complicate the comparisons
discussed here. Specifically, these are the EMEP network for Europe, the CASTNET network for the U.S., and
the Acid Deposition Monitoring Network in East Asia (EANET, https://www.eanet.asia/, last accessed 14 Aug
2024) for Southeast Asia. No filtering of the data was performed before making the comparisons. Although
seasonal variability is of interest, the EANET wet deposition totals are only provided as annual mean values, which
limits the sampling frequency used for the analysis.
The averaging period chosen for the evaluation is primarily constrained by the availability of data from Southeast
Asia, which only provides annual mean values. For Europe and the U.S., data is provided on different timescales,
ranging from daily to weekly (CASTNET) and biweekly (AMoN). Averaging of model data is done using the
respective time intervals for each dataset to provide weekly composites at the selected stations used for $SO_2$, $NH_3$,
and $HNO_3$ comparisons.



## 4. The influence of pH on SIA chemical precursors and particulates

The efficacy of SIA formation is strongly governed by the concentrations of gaseous precursors. Therefore, changes introduced to the parameterizations for simulating particle formation also have feedback effects on the precursors, due to changes in fractional uptake governed by the solute pH. In this section, we evaluate the temporal and regional distribution, as well as biases, of both gaseous precursors ($SO_2$, $NH_3$, $HNO_3$) and SIA (namely $SO_4^{2-}$, $NH_4^+$, $NO_3^-$) simulated by IFS-COMPO for Europe, the U.S., and Asia. Mixing ratios and particle concentrations are strongly influenced by the description and distribution of primary emission sources, meteorology, deposition, aerosol pH (for $NH_x$ and $NO_x$), and atmospheric transport. To investigate IFS-COMPO's ability to capture observed distributions, we present both weekly and annual mean comparisons for CY48R1 and CY49R1 against corresponding measurement composites. Given that the differences between CY48R1 and CY49R1_NOE4C are smaller (as shown in the budget analysis of gaseous precursors), we limit the selection of results for brevity. A direct link exists between [$NH_4^+$] and [$NO_3^-$] because the Nitrate#1 tracer takes the form of $NH_4NO_3$. All observational data are used to calculate the statistics, so they represent the mean across different chemical regimes. However, the location of sampling sites is not homogeneous throughout the analysis region, meaning results can be weighted towards certain states/countries. For Mean Bias (MB) and Root Mean Square Error (RMSE), negative percentage differences indicate improvements in bias statistics, whereas for Pearson's R, a positive percentage difference indicates improvement.

To assess the scale of these feedbacks, we show monthly mean regional differences for July and December 2018 for the three selected regions, focusing on $SO_2$, $NH_3$, and $HNO_3$. To evaluate the performance of IFS-COMPO, we aggregate data at a weekly frequency. For SIA, we present annual mean values against observations.

### 4.1 $SO_2$ and $SO_4^=$

Figure 1 shows weekly comparisons of surface $SO_2(g)$ concentrations ($\mu g/m^3$) from the three regions for 2018 against observational composites assembled from background/rural stations, as selected from the relevant observational networks introduced in Section 3. The associated annual mean statistics are provided in Table 2. In Europe, observations show typical weekly values of 0.5-1 $\mu g/m^3$ with no significant seasonal cycle, indicating the effective mitigation of $SO_2$ emissions in the region over the last few decades (Aas et al., 2024). In contrast, IFS-COMPO exhibits a seasonal cycle with an amplitude of 2 $\mu g/m^3$, driven by the monthly variability in the bottom-up emission inventories used. Typically, there is a positive bias in the simulations throughout the year, varying between 0.5 and 1 $\mu g/m^3$. A notable reduction in the wintertime bias of 0.5-1 $\mu g/m^3$ between CY48R1 and CY49R1_NOE4C (see Fig. 1A in the Appendix) indicates an increase in $SO_4^{2-}$ production (quantified as a small increase, as shown in Table 3). For CY49R1, the MB with respect to surface [$SO_4^{2-}$] decreases by approximately 0.1-0.2 $\mu g/m^3$. The primary source of $SO_2$ being direct emissions suggests that emission estimates for Eastern Europe may be too high (see Fig. 1A in the Appendix). The annual MB value decreases by around 25%, with moderate correlation. In the U.S., weekly mean $SO_2$ values in the observations are typically around 2.5 $\mu g/m^3$, more than double those observed in Europe.

Figure A1 in the Appendix details the regional monthly mean distributions of surface $SO_2$ mixing ratios for July and December 2018 for CY48R1, along with percentage differences between CY48R1, CY49R1_NOE4C, and CY49R1. To assess the global integrated impact on $SO_4^{2-}$ formation, the associated global budget terms are provided in Table 3 in Tg S/year. Comparing spatial distributions across regions, Europe exhibits the lowest $SO_2$ mixing ratios in CY48R1, reflecting strong mitigation practices over the last few decades (e.g., Vestreng et al., 2007). The maps for December show higher mixing ratios toward the east, with a significant contribution from shipping. In the U.S., a stark east-west gradient exists, governed by the continental distribution of anthropogenic emissions, with higher emissions toward the East Coast, again showing a seasonal signature. Maximal surface mixing ratios are 5-10 times higher than those simulated for Europe and are distributed over a much larger area. As expected, China exhibits the highest mixing ratios, between 10-20 ppb across the country, which is approximately 20 times higher than those simulated for Europe in both months shown.

Comparing CY48R1 against CY49R1_NOE4C reveals reductions in surface [$SO_2(g)$] across all regions of between 0-10%, leading to limited increases in $SO_4^{2-}$ production of a few percent due to changes unrelated to aerosol and solution pH updates. This small increase in $SO_4^{2-}$ production is reversed when applying the EQSAM4Clim pH methodology (Metzger et al., 2024), where the conversion efficacy of $SO_2$ is faster at more alkaline pH. The global budget terms show that, in addition to primary emissions, approximately one-third of $SO_2$



in the troposphere comes from the oxidation of dimethyl sulfide (DMS) by hydroxyl radicals (OH), with DMS
originating from biogenic activity in the oceans. In CY48R1, approximately 20% of $SO_2$ is oxidized in the gas
phase and 43% in the aqueous phase, with the remaining 37% lost to the surface via dry and wet deposition. This
increase in gas-phase production via OH is linked to changes in $O_3$-$NO_x$ reaction cycles near anthropogenic source
regions, resulting in a small increase in $O_3$ of a few percent (not shown). The corresponding values for
CY49R1_NOE4C show changes of a few percent across terms, increasing the global burden by 1.5%, mostly in
the lower troposphere. For CY49R1, the application of EQSAM4Clim pH in cloud droplets reduces both the
uptake and oxidation of $SO_2$ by reducing aquated sulfite ($[SO_3^{2-}]aq$, $pKa(HSO_3^-) = 7.2$) and enhances gas-phase
oxidation due to increased OH, resulting in more gas-phase production of $H_2SO_4$, which is subsequently scavenged
into solution, further increasing solution acidity (lower pH values) in cases of excess $SO_4^{2-}$ (i.e., insufficient
cations to completely neutralize all $SO_4^{2-}$).
**Table 2**: The annual Mean Bias (MB), Root Mean Square Error (RMSE), and Pearson's R values for weekly mean
regional distributions and concentrations of gaseous $SO_2$ compared against observational composites for 2018, as
shown in Figure 1, for Europe, the U.S., and China. For China, statistics relate to seasonal means.

| Diagnostic | Europe (EMEP) | | US (CASTNET) | | China (CNEC) | |
|---|---|---|---|---|---|---|
| | CY48R1 | CY49R1 | CY48R1 | CY49R1 | CY48R1 | CY49R1 |
| MB (µg/m³) RMSE Pearsons R | 0.85 0.48 | 0.77 0.49 | -0.16 0.185 | -0.26 0.192 | 11.6 0.07 | 11.5 0.07 |


**Table 3:** The tropospheric $SO_2$ budget in Tg S/year for 2018, as calculated by CY48R1, CY49R1_NOE4C, and
CY49R1, with the associated relative differences provided in parentheses (e.g., ((CY49R1 - CY48R1) / CY48R1)
* 100).

| Process | CY48R1 | CY49R1_NOE4C | CY49R1 |
|---|---|---|---|
| **Emission** | 54.0 | 54.0 (-) | 54.0 (-) |
| $DMS + OH \rightarrow SO_2$ | 21.8 | 21.8 (-) | 21.5 (-1.6) |
| $SO_2 + OH \rightarrow H_2SO_4$ | 15.1 | 15.4 (+2.3) | 16.5 (+9.3) |
| $SO_2 (aq) \rightarrow SO_4 (aq)$ | 33.7 | 33.9 (+1.2) | 33.0 (-2.2) |
| Dry Deposition | 21.6 | 21.3 (-3.0) | 22.2 (+3) |
| Wet Deposition | 8.2 | 8.0 (-3.0) | 6.9 (-15.8) |
| Burden | 0.70 | 0.71 (+1.4) | 0.75 (+7.1) |
| Lifetime (days) | 3.25 | 3.29 (+1.2) | 3.48 (+7.1) |


Figure 1A shows that the region with the highest surface $SO_2$ concentrations is the northeastern U.S., with other
regions moderating the biases. There is little seasonality in the weekly observational composites. A positive bias
is observed during the winter and a negative bias during the summer, around 0.5-1.0 µg/m³ across all simulations.
In CY49R1, the annual mean negative bias increases by approximately 0.1 µg/m³, and there is poor correlation
with the observations. In China, weekly $SO_2$ concentrations are an order of magnitude higher than those observed
in the other regions, reaching 15-20 µg/m³ during winter. The simulated concentrations exhibit a very large
positive bias, between 10-20 µg/m³, suggesting that the regional $SO_2$ emissions in the global inventory are
significantly overestimated. Only negligible improvements are observed in CY49R1, where no detectable
correlation exists between the simulated and observed values.



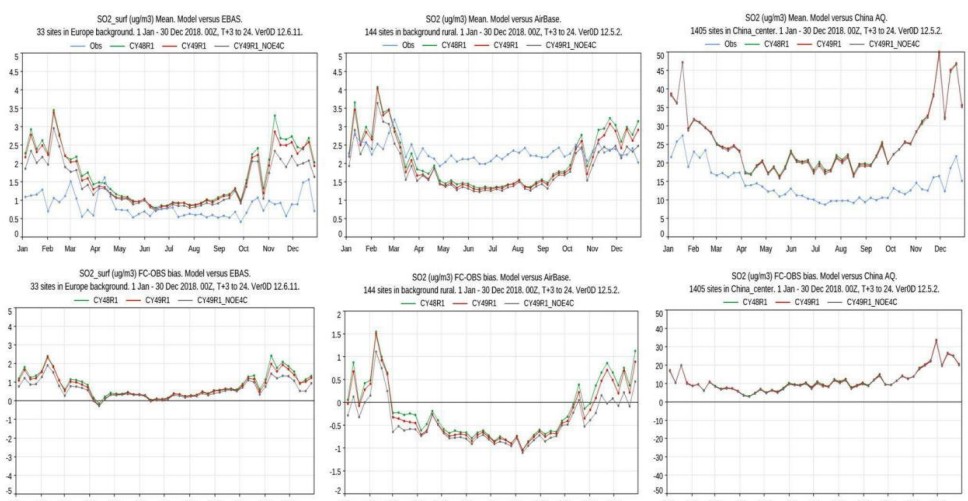


**Figure 1:** A comparison of weekly mean $SO_2$ concentrations for (a) Europe (AirBase, left), the U.S. (AirNow, middle), and China (CNEC, right), simulated in CY48R1 (green), CY49R1_NOE4C (grey), and CY49R1 (red), compared against measurement composites from stations representative of rural conditions in 2018.

Figure 2 compares weekly $SO_4^{2-}$ surface concentrations in Europe, using data from the EBAS archive and IFS-COMPO simulations, along with associated biases. There is a seasonal cycle in the observational composites, with higher surface $[SO_4^{2-}]$ during winter, despite the weaker seasonal cycle for $SO_2$ (cf. Figure 1). The simulations show a weak seasonal cycle for surface $SO_4^{2-}$, with fairly constant values during the summer and low weekly biases. However, significant weekly biases occur in winter, where observations exhibit high variability. CY48R1 shows a low annual MB of -0.41 µg/m³, which is reduced in CY49R1_NOE4C (-0.20 µg/m³) and further in CY49R1 (+0.01 µg/m³). The correlation coefficient improves marginally, from 0.37 (CY48R1/CY49R1_NOE4C) to 0.43 (CY49R1).

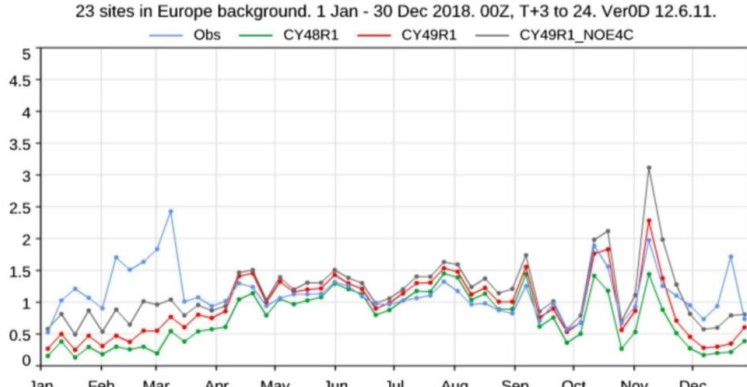

332

**Figure 2:** A comparison of weekly mean $SO_4^{2-}$ for Europe (µg/m³) simulated in CY48R1, CY49R1_NOE4C, and CY49R1, compared against measurement composites from stations representative of rural conditions in 2018.

Figure A2 in the appendix shows the corresponding comparison against weekly observational composites of surface $SO_4^{2-}$ from the CASTNET measurement network. No seasonal cycle is observed in the surface $[SO_4^{2-}]$ observations, with typical values around 0.7-1.0 µg/m³. During winter, lower weekly MB values are observed for both





CY49R1_NOE4C and CY49R1, with some degradation in weekly MB for certain weeks when applying EQSAM4Clim. In summer, much larger positive biases occur, reaching 100-150% for all simulations due to the strong seasonal cycle, with only marginal improvement in CY49R1 regarding weekly MB. The low MB for $SO_2$ shown in Figure 1 suggests that the rate of oxidation in IFS-COMPO is too fast.

Figure 3 presents the annual mean surface $[SO_4^{2-}]$ for CY48R1 and CY49R1 for Europe, the U.S., and Southeast Asia. The changes in surface $[SO_4^{2-}]$ are somewhat unaffected by the aerosol pH changes due to EQSAM4Clim, as shown in Rémy et al. (2024) for 2019. This is because $SO_4^{2-}$ production is dominated by aqueous-phase processes, with small increases from organic acids. One key difference for $SO_x$ is that $SO_4^{2-}$ production is irreversible, depending on cloud pH, dissolved $O_3$, and hydrogen peroxide ($H_2O_2$).

In Europe, sampling sites for this aerosol species in the EMEP network are such that comparisons for southern European countries are excluded from the regional mean statistics. A sharp north-south gradient exists, driven by variability in $H_2SO_4$ production between seasons, cloud cover for the wet production term, and the distribution of primary $SO_2$ emission sources. Mitigation measures have reduced the increase in emitted fluxes during winter months associated with domestic heating (Versteeg et al., 2007). Simulated concentrations in CY48R1 are lower in Scandinavia compared to countries like France, resulting in a low bias of around 1 µg/m³ in Finland and around the Baltic, related to missing shipping emissions. In other European sites, the agreement is better, with the low bias decreasing to approximately 0.5 µg/m³. One outlier exists at the most easterly station, which exhibits a significant high bias of 1.5 µg/m³. In CY49R1, simulated surface $[SO_4^{2-}]$ increases by 0.2-0.4 µg/m³, leading to improved bias, as shown in Table 7. However, only small improvements are made to the correlation coefficient due to identical emission estimates and the fact that $SO_x$ is the least affected by EQSAM4Clim, impacted only indirectly through changes in pH.

In the U.S., CASTNET observations show an east-west continental gradient in surface $[SO_4^{2-}]$, determined by the distribution of primary $SO_2$ emissions and transport (cf. Figure A1). There is a significant transport component for $SO_4^{2-}$, with surface $[SO_4^{2-}]$ in the marine boundary layer ranging from 1.0-2.5 µg/m³, where transport dominates local surface $[SO_4^{2-}]$ produced from DMS oxidation (Simpson et al., 2014). In CY49R1, surface $[SO_4^{2-}]$ decreases at the continental scale, reducing the annual MB from 0.67 to 0.20 µg/m³, with a corresponding increase in the correlation coefficient to 0.43, although it remains weakly correlated.

In the western U.S., a positive MB is introduced for rural background sites in CY49R1, ranging from 0.5-0.7 µg/m³, with a contribution from transport from the east. Therefore, reductions in the annual MB are driven by lower biases related to eastern sampling stations. The positive MB of approximately 1-1.5 µg/m³ around Kentucky/Tennessee suggests that local $SO_2$ emission estimates may be too high (see Discussion in Section 5).





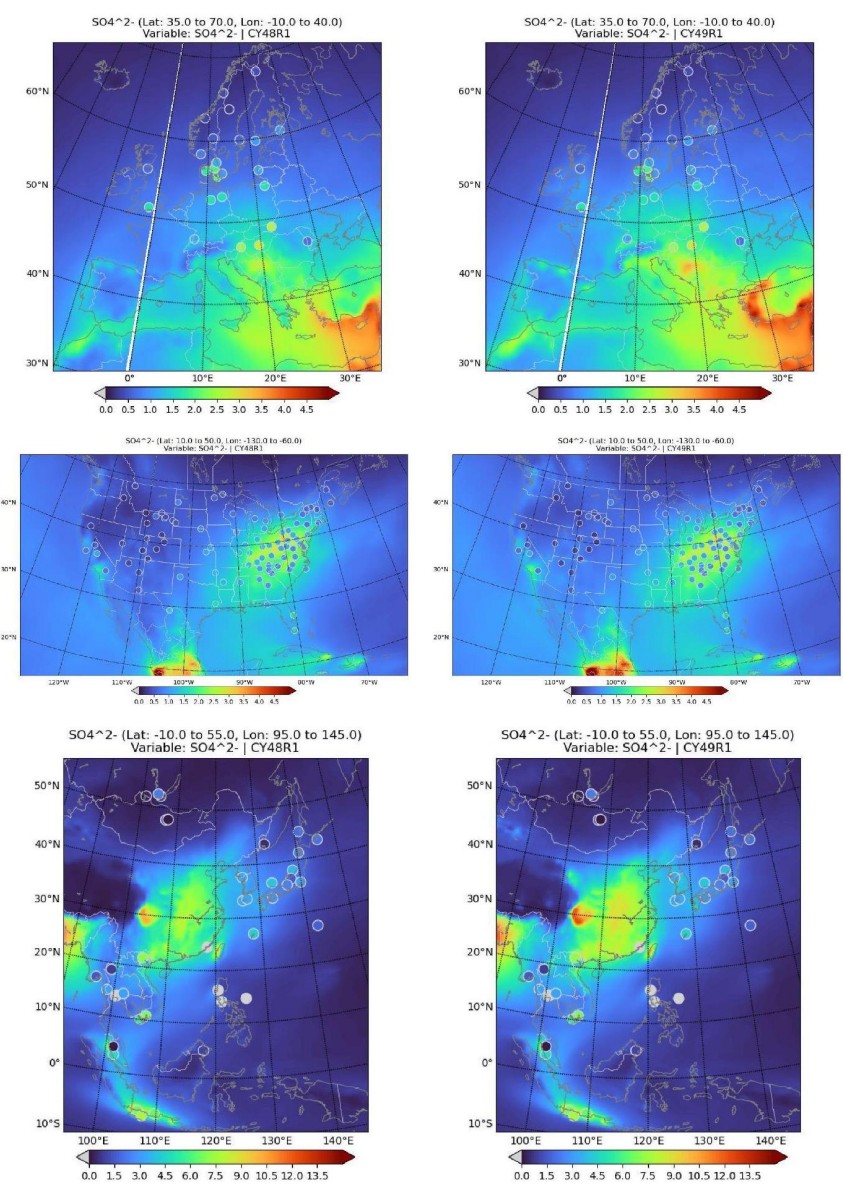


**Figure 3:** Comparisons of annual mean [SO₄²⁻] simulated in CY48R1 and CY49R1 compared to
measurements for the three selected regions, given in µg/m³.
In Asia, the scarcity of sampling sites in the EANET network results in a less robust evaluation. Many sampling sites
are located on the coast rather than inland, so changes in coastal regions significantly influence regional statistics.
Higher primary SO₂ emissions occur inland. Therefore, any positive MB near source regions is not included in the
statistics; the results shown here for surface SO₄²⁻ should be considered lower limits. Long-range transport of SO₄²⁻
in Asia has been shown to partially neutralize national SO₂ mitigation measures, such as those in Taiwan and South
Korea. This originates from changing trends in SO₂ emissions from mainland China, as captured by EANET
measurement sites (Chang et al., 2022). For CY48R1, the annual mean statistics show a very low MB and a good
correlation coefficient of 0.75. However, in CY49R1, performance degrades, with the MB increasing to 0.48 µg/m³,
showing a trend similar to that in the U.S. Notably, more remote sampling stations (e.g., oceanic) exhibit regional





negative biases (approximately -0.7 µg/m³), while sites near Mongolia and South Korea agree well, with low MB
values. In Thailand and Vietnam, large MB values suggest that regional $SO_2$ emission estimates are too high.
Unfortunately, there are no in-situ measurements available for better quantification. The correlation coefficient
degrades in CY49R1 compared to CY48R1, dropping to 0.66. Overall, improvements in the $SO_2$-$SO_4^{2-}$ couple are
mixed and less pronounced compared to other SIA species.
**Table 4**: The annual MB, RMSE, and Pearson's R values for daily (EMEP, Europe), weekly (CASTNET, U.S.), and
annual (EANET, Southeast Asia) mean regional distributions and concentrations of surface $SO_4^{2-}$ compared to
observational composites for 2018 shown in Figures 8-10 for Europe, the U.S., and Southeast Asia. Percentage
differences are calculated as ((CY49R1 - CY48R1) / CY48R1) * 100.

| | Europe (EMEP) | | US (CASTNET) | | SE Asia (EANET) | |
|---|---|---|---|---|---|---|
| $SO_4^{=}$ | CY48R1 | CY49R1 | CY48R1 | CY49R1 | CY48R1 | CY49R1 |
| MB (ug/m³) | -0.49 | -0.32 (-35) | 0.67 | 0.20 (-70) | -0.02 | 0.48 (+96) |
| RMSE | 1.35 | 1.31 (-3) | 0.93 | 0.46 (-50) | 1.64 | 2.28 (+39) |
| Pearsons R | 0.45 | 0.47 (+4) | 0.33 | 0.43 (+23) | 0.75 | 0.66 (-12) |


## 4.2 NH$_3$ and NH$_4^+$

**4.2 NH₃ and NH₄⁺**
Figure 4 compares weekly observational composites of $[NH_3(g)]$ from the EBAS archive against data extracted from
the various IFS-COMPO simulations for 2018. The observational composite shows a skewed seasonal cycle, with a
maximum in April due to agricultural activity. Wintertime values are around 0.5 µg/m³, increasing to 1.0-2.0 µg/m³
during spring and summer. This seasonal variability is captured across all simulations, albeit with a significant positive
summertime weekly bias of 1-2 µg/m³ in CY48R1 (annual MB: 1.04 µg/m³). A small increase in bias is simulated for
CY49R1 (annual MB: 1.21 µg/m³). There is high correlation across simulations, with values ranging from 0.71-0.73.
The occurrence of weekly increases in observed values is typically captured by IFS-COMPO, but there is a modest
degradation in performance compared to CY48R1.
The regional distribution of surface NH₃ for 2018 in the three chosen regions, and the changes resulting from both the
IFS cycle upgrades and the application of EQSAM4Clim, are shown in Figure A2 in the Appendix. The corresponding
global chemical budget terms are provided in Table 5. Despite a declining trend in European regional NH₃ emissions
(Tich? et al., 2023), a strong seasonal cycle exists in CY48R1. Maximal mixing ratios are found around Benelux and
northern Italy, with local differences of 8-20 ppb between July and December across regions. The
CAMS_GLOB_ANT v5.3 (Soulie et al., 2023) emission inventory has recently been validated for NH₃ against top-
down estimates, providing confidence in the estimates' quality (Ding et al., 2024). In the U.S., a similar seasonal
signature exists, especially in the northwest and southeast, associated with agricultural emissions (Wang et al., 2020),
with background mixing ratios of 0.5-2.0 ppb remaining relatively constant.
In China, where NH₃ emissions have increased over recent decades (Liu et al., 2019; Chen et al., 2023), surface mixing
ratios of 5-20 ppb occur in July over large areas, again associated with agricultural practices. Similarly, high mixing
ratios are observed around Bangladesh (> 20 ppb). In December, mixing ratios are typically an order of magnitude
lower, except in the southwest, where high mixing ratios (> 20 ppb) persist. Measurements of NH$_x$ over the ocean are
rare, so the large increases shown cannot be verified. Nevertheless, estimates range from 0.1-4.2 ppb depending on
season and location (Sharma et al., 2012), indicating a significant negative bias in CY48R1 that is somewhat improved
in CY49R1.
**Table 5**: The tropospheric NH₃ budget in Tg N/year for 2018, as calculated by CY48R1 and CY49R1, with relative
differences shown as (CY49R1-CY48R1)/CY48R1.

| Process | CY48R1 | CY49R1_NOE4C | CY49R1 |
|---|---|---|---|
| **Emission** | 51.1 | 51.1 (-) | 51.1 (-) |
| NH₃ + OH | 0.82 | 0.99 (+20) | 1.98 (+240) |



| | | | |
|---|---|---|---|
| $NH_3 \rightarrow NH_4^+$ | 30.6 | 30.3 (-1) | 17.3 (-44) |
| Dry Deposition | 16.3 | 16.6 (+2) | 22.4 (+37) |
| Wet Deposition | 7.0 | 6.2 (-13) | 10.6 (+51) |
| Burden | 0.13 | 0.16 (+19) | 0.29 (+118) |
| Lifetime (days) | 0.9 | 1.1 (+22.0) | 2.0 (+133) |


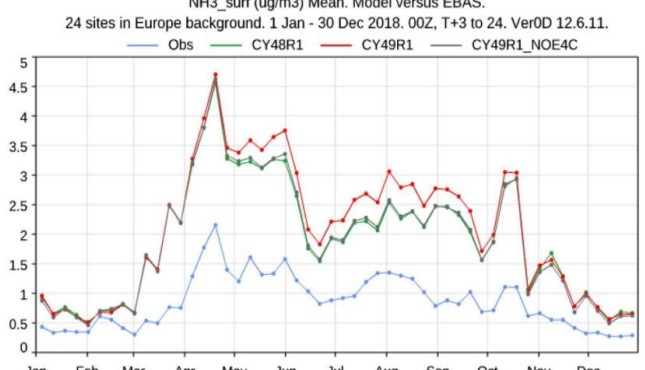

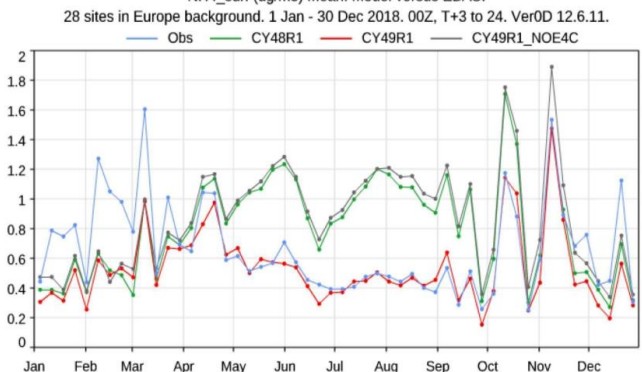


**Figure 4:** A comparison of weekly mean $[NH_3(g)]$ and $[NH_4^+]$ at the surface for Europe ($\mu g/m^3$), simulated in CY48R1 (green), CY49R1_NOE4C (grey), and CY49R1 (red), compared against measurement composites from stations representative of a rural scenario for 2018. The corresponding biases are shown in the bottom panel.

Comparing CY48R1 and CY49R1_NOE4C shows decreases of 5-20% over land in the chosen regions. Changes in $[NH_4^+]$ production are minimal in the absence of EQSAM4Clim, with most $NH_3(g)$ lost through conversion to $NH_4^+$. In CY49R1_NOE4C, there is a 13% decrease in the dissolved fraction of $NH_3$, which is subsequently lost as wet deposition, contributing to a nearly 20% increase in the tropospheric $NH_3$ burden. This is attributed to changes in scavenging and wet deposition. In CY49R1, improved gas/particle partitioning from EQSAM4Clim reduces particle-phase concentrations of semi-volatile aerosol species, increasing gas-phase concentrations and affecting aerosol pH, which governs $NH_3(g)$ solubility. This reduces its conversion into $NH_4^+$ (see Table 5, approx. 44% reduction), amplified by the inclusion of mineral cations ($Ca^{2+}$, $Na^+$, $K^+$, $Mg^{2+}$). The tropospheric lifetime of $NH_3$ more than doubles in CY49R1, in line with changes in the tropospheric burden. Both dry and wet deposition increase (by 37% and 51%, respectively) due to lower $NH_4^+$ particle production.



Figure 4 also presents weekly comparisons of observational composites of [$NH_4^+$] at the surface from the EBAS
archive against IFS-COMPO simulations for 2018. Although maximum observed $NH_3(g)$ in Figure 4 occurs in May,
higher [$NH_4^+$] is observed during winter. Both CY48R1 and CY49R1_NOE4C exhibit summertime mean biases of >
100% (0.5 µg/m³), which are removed in CY49R1 by applying EQSAM4Clim, resulting in a very low bias (< 0.1
µg/m³). This potentially improves PM2.5 and PM10 forecasts significantly by reducing cumulative bias across aerosol
types (see Rémy et al., 2024).
Similar comparisons are shown in Figure A5 in the Appendix for the U.S., using weekly composites from CASTNET
data. Unlike the seasonal cycle for [$NH_3(g)$], which peaks in May (see Fig. A4), weekly [$NH_4^+$] values peak in July,
remaining fairly consistent between 1.5-1.75 µg/m³ during summer. This suggests saturation in $NH_4^+$ particle
formation, likely linked to the availability of $HNO_3(g)$ (cf. Fig. A7 in the Appendix). Significant biases exist in
CY48R1 and CY49R1_NOE4C, reaching 1.25 µg/m³ during summer (600% of observational values). Applying
EQSAM4Clim halves this positive bias in CY49R1, resulting in better agreement during winter.
A comparison of weekly [$NH_3(g)$] variability between IFS-COMPO and measurements from rural AMON sites in
2018 is shown in Figure A5 in the Appendix, with site details provided in the figure legend. Sites were selected to
cover a wide area of the U.S. Measurements show that winter [$NH_3(g)$] concentrations are lower than summer
concentrations across most sites, except in California, where seasonal temperature variability (and agricultural
practices) is less pronounced. Maximal concentrations range from 1-6 µg/m³, occurring during spring (Alabama/New
York) or summer (Florida), depending on the extent and timing of agricultural activity in each state. Differences
between CY48R1 and CY49R1_NOE4C are negligible, but weekly bias is significantly reduced in CY49R1. Arizona
is an exception, where a large positive bias suggests a too-high local emission flux. In CY49R1, high [$NH_3(g)$] also
depends on local [$HNO_3(g)$] via $NO_x$, the other important precursor for [$NH_4(NO_3)$] (cf. New York vs. Alabama). The
seasonality of weekly variability is well captured, with substantial improvements in cycle amplitude in CY49R1.
The corresponding statistics for 2018 against AMoN composites for all three simulations are shown in Table A2 in
the Appendix. Differences between CY48R1 and CY49R1_NOE4C in the U.S. are not appreciable (cf. Figure A2),
despite the global increase in the tropospheric $NH_3$ burden in Table 4. Without the aerosol pH changes in
EQSAM4Clim, limited repartitioning occurs. Therefore, further discussion is limited to changes in CY49R1 statistics.
The negative annual MB in CY49R1 is approximately half that in CY48R1, decreasing to 0.26 µg/m³, reflecting $NH_x$
repartitioning into $NH_3(g)$ at higher aerosol pH, with a modest improvement in the correlation coefficient.
Figure 5 presents the annual mean [$NH_4^+$] distribution for Europe, the U.S., and Southeast Asia in CY48R1 and
CY49R1 during 2018, with regional annual mean statistics in Table 6. Measurement site locations are also shown,
with respective annual mean values within each circle. Significant decreases in $NH_3$ conversion in CY49R1 result in
lower [$NH_4^+$] concentrations, driven by improved gas/aerosol partitioning and increased aerosol pH when applying
EQSAM4Clim (see Table 4; Rémy et al., 2024). $NH_3(g)$ depositional loss to the surface increases in CY49R1 due to
its longer residence time. Aerosol pH varies widely between regions, with Europe exhibiting values of 3-4, while the
southern U.S. and northern China exhibit pH values of 2-3 (Pan et al., 2024; Rémy et al., 2024), indirectly affecting
$NH_4^+$ production variability. Once formed, regional transport contributes to the continental distribution of $NH_4^+$ away
from strong source regions (Simpson et al., 2010; Renner and Wolke, 2010; Du et al., 2020).
In Europe, most observational annual mean values are between 0.2-1.2 µg/m³, exceeded by > 50% in CY48R1. In
CY49R1, annual mean [$NH_4^+$] decreases by 0.5-1.0 µg/m³, resulting in low annual mean [$NH_4^+$] values for Spain and
the UK, while reducing maximal concentrations by approximately 50% in northern Italy. This contributes to a
reduction in cumulative PM2.5 bias in the region, as shown in Rémy et al. (2024) for 2019. The associated MB values
in Table 6 show a significant bias reduction (> 80%) and an increase in the correlation coefficient, although the
simulated $NH_4^+$ distribution is still only moderately correlated (r=0.62). Unfortunately, no available measurements
allow for quantification of IFS-COMPO performance around the Mediterranean. It should be noted that with a more
realistic distribution and seasonal variability in $NH_3(g)$ emissions (Shepard et al., 2011; Dammers et al., 2019), the
[$NH_4^+$] distributions would likely not be affected, as other SIA species govern $NH_3$-$NH_4^+$ gas/aerosol partitioning (see
Discussion).
In the U.S., similar decreases in annual mean [$NH_4^+$] values occur in CY49R1, with very low concentrations (0.1-0.4
µg/m³) in the western U.S., reducing bias compared to observational mean values. This reduces the annual mean
regional bias by approximately 0.7 µg/m³, as shown in Table 6. A gradient exists in aerosol pH from EQSAM4Clim,
with values ranging from pH 3.0 in the northwest U.S. to more acidic values of pH 2.0 in the southwest (Rémy et al.,
2024). This reduces $NH_3(g)$ transfer, thus moderating $NH_4^+$ production (cf. Table 4). In the northeast U.S., with high





NO$_x$ emissions, reductions of 0.5-1.0 µg/m³ occur. In the southwest U.S., with high [NH$_3$(g)] from agriculture (cf.
Figure A2), reductions of 0.3-1.0 µg/m³ are observed. The correlation coefficient degrades, showing a moderate
annual mean correlation with significant overestimates in the southwest U.S., as shown in comparisons of [NH$_3$(g)]
at selected sites in Figure 6.
In Southeast Asia, simulated annual mean [NH$_4^+$] over land is typically much higher than in Europe or the U.S., with
maximal values of 7.0-9.0 µg/m³ in eastern China, despite similar surface NH$_3$(g) mixing ratios between Europe and
China (see Figure A3). This difference is driven primarily by higher HNO$_3$(g) in China, due to a more polluted
chemical regime (O$_3$, NO$_2$, and OH determine gas-phase HNO$_3$ production). Applying EQSAM4Clim in CY49R1
reduces [NH$_4^+$] by 1-2 µg/m³, particularly where annual mean [NH$_4^+$] exceeds 6.0 µg/m³. This reduces the annual
mean regional bias by 0.4 µg/m³, with a corresponding reduction in correlation due to less transport. The lack of
sampling sites in regions with high primary NH$_3$(g) emissions skews the annual mean biases. In more remote locations
(e.g., Mongolia/South China Sea), low values of < 0.5 µg/m³ are well captured in both CY48R1 and CY49R1.
**Table 6** As for Table 4, but for NH$_4^+$.

| | Europe (EMEP) | | US (CASTNET) | | SE Asia (EANET) | |
|---|---|---|---|---|---|---|
| NH$_4^+$ | CY48R1 | CY49R1 | CY48R1 | CY49R1 | CY48R1 | CY49R1 |
| MB (ug/m³) | 0.26 | -0.05 (-81) | 0.95 | 0.23 (-48) | 0.96 | 0.55 (-43) |
| RMSE | 0.94 | 0.72 (-23) | 1.71 | 0.46 (-73) | 1.73 | 1.30 (-25) |
| Pearsons R | 0.46 | 0.62 (+29) | 0.59 | 0.43 (-27) | 0.59 | 0.44 (-25) |





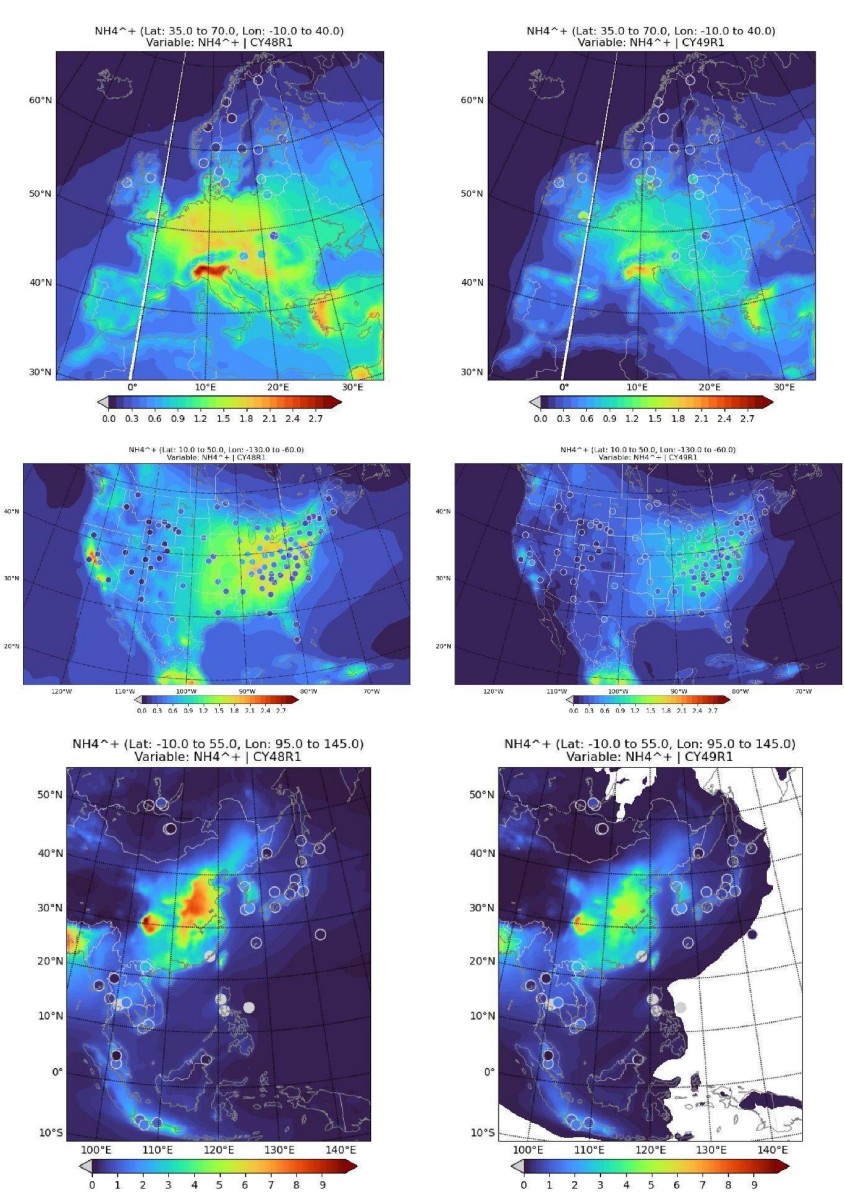

**Figure 5:** Comparisons of annual mean surface $NH_4^+$ particle concentrations simulated in CY48R1 and CY49R1, compared to measurements for the three selected regions during 2018 (µg/m³). The corresponding regional statistics are provided in Table 7.



### 4.3 HNO₃ and NO₃⁻

Figure 6 shows the resulting changes in surface [HNO₃(g)] between simulations, compared against weekly composites assembled from a selection of background measurement sites in Europe that participate in the EMEP measurement network. The location of the sampling sites results in a significant bias towards northern Europe, where seasonality is more pronounced. However, the observations exhibit only a weak seasonal cycle, with weekly values ranging from 0.4-0.8 µg/m³, as many sites are located away from strong NO$_x$ sources. Both CY48R1 and CY49R1_NOE4C (see Figure A7) show negative biases, underestimating concentrations by around 100% during summer. In CY49R1_NOE4C, there is a bias reduction of approximately 0.1-0.2 µg/m³, indicating that other changes made between IFS cycles cause alterations in the gas-phase production term in addition to the changes in NH₄NO₃ from EQSAM4Clim (cf. Table 8). In CY49R1, there is a significant excess of [HNO₃(g)] due to enhanced production and reduced transfer into the particulate phase, despite an increase in cumulative deposition terms. Such changes are associated with relatively low [HNO₃(g)] values in IFS-COMPO, around < 0.1 ppb (see Figure A7).

Figure A7 in the Appendix shows the monthly mean regional distribution of HNO₃(g) for July and December 2018 for the three selected regions for CY48R1, along with percentile differences when compared with CY49R1_NOE4C and CY49R1. The corresponding global chemical budget terms for HNO₃(g) are provided in Table 7. No direct emission of HNO₃ occurs in IFS-COMPO, as is often prescribed in global chemistry transport models to represent chemistry in ship plumes. Instead, the main source is the oxidation of NO₂ by OH in the gas phase, as shown in Table 7. This production term increases by approximately 14% in CY49R1 due to enhanced OH from changes in O₃ (not shown). For heterogeneous conversion, the cumulative HNO₃ production term is approximately 50% that of the gas-phase production term, remaining relatively constant between simulations. A shift occurs between fine mode NO₃⁻ (NH₄NO₃) and coarse mode NO₃⁻ (CaNO₃/NaNO₃), strengthening the link between NH₄⁺ and NO₃⁻ in IFS-COMPO. Both dry and wet loss terms increase significantly due to the increased availability of HNO₃(g), reducing the fraction converted to NO₃⁻. The temporal variability of HNO₃(g) is influenced by the magnitude and extent of regional NO$_x$ emissions, photochemical activity (via OH formation), gas/aerosol partitioning (where particles with high SO₄²⁻ content have an associated low NO₃⁻ content), and scavenging in clouds and aerosols.

In Europe, very low surface mixing ratios occur over land during both months in CY48R1 (< 0.1 ppb), which is surprising given that Benelux is known for high NO$_x$ levels (van der A, 2024), suggesting correspondingly high HNO₃(g) mixing ratios. Higher mixing ratios of 0.25-0.5 ppb occur around the coasts and the Mediterranean, originating from direct shipping emissions. This can lead to elevated NO₃⁻ concentrations due to the uptake of HNO₃(g) on sea salt, which may be overestimated, as EQSAM4Clim currently assumes thermodynamic equilibrium without accounting for dynamic limitations. A coupling with a dynamic aerosol model is foreseen. In contrast, applying EQSAM4Clim in CY49R1 results in large increases in surface HNO₃(g) at the continental scale during July. In December, strong latitudinal variability occurs, with decreases of 25-75% in HNO₃(g) at latitudes higher than 60°N, due to lower temperatures and lower RH in a relatively low NO$_x$ environment.

In the U.S., the highest HNO₃(g) mixing ratios in CY48R1 occur in the eastern states and California (1-2 ppb), with much lower values in the more remote central U.S. (0.1-0.2 ppb), and a strong seasonal cycle with maximum values peaking in July. Comparing the relative differences between simulations shows a significant increase in surface HNO₃(g) in CY49R1 (100-6000 ppt) across the continent for both months, with the largest increases occurring in the northern states. In contrast to Europe, no seasonal decreases are observed at any location.

In Southeast Asia, surface mixing ratios are the highest across all regions, with maximum values of 4-5 ppb along the eastern coast (July) and in central regions (December). Comparing the relative differences between simulations shows significant increases of 50-5000%, except in the more remote northern regions. As in Europe, strong seasonality is observed, with decreases above 30°N, regardless of the NO$_x$ regime. As shown for NH₃ (see Figure A3 in the Appendix), significant increases in HNO₃ over the ocean occur for both months, associated with lower [NO₃⁻] (as shown by the cumulative 50% reduction in global conversion).



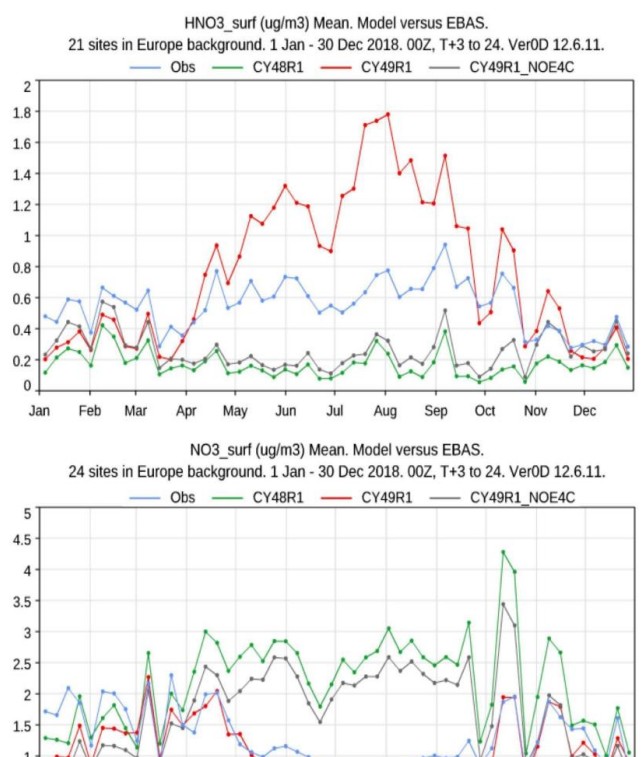


**Figure 6**: A comparison of weekly mean [HNO₃(g)] and [NO₃⁻] for Europe (µg/m³) at the
surface, simulated in CY48R1 (green), CY49R1_NOE4C (grey), and CY49R1 (red),
compared against measurement composites from stations representative of a background
scenario in 2018. The evolution of the corresponding bias values is shown in the bottom panel.
Comparisons of weekly [HNO₃(g)] from the CASTNET measurement network in the U.S. are shown in the top
panel of Figure A8 in the Appendix and reveal similar conclusions. As in Europe, both the concentrations and
seasonal variability in the observations are low, with typical weekly concentrations around 0.5 µg/m³. The
relatively even distribution of measurement sites in the U.S. means that the evaluation presented does not have
significant regional bias. It is surprising that measured weekly mean concentrations are relatively constant, given
that variability in the gas-phase chemical production term involves OH, which exhibits strong seasonality due to
day length differences. In contrast to Europe, both CY48R1 and CY49R1_NOE4C show moderately good
agreement with the measurements, with weekly biases around 0.2-0.25 µg/m³. However, CY49R1 introduces a
large positive bias from EQSAM4Clim due to a limitation in HNO₃'s ability to condense on particle surfaces, as
condensed HNO₃ does not contribute to NH₄NO₃ formation (this requires coupling EQSAM4Clim with a dynamic
aerosol model, as described in Metzger et al., 2018). It also shows that although cumulative global dry and wet
deposition terms in CY49R1 have increased markedly compared to CY48R1 (cf. Table 8), this is insufficient to
compensate for the reduced aerosol formation.
The bottom panel of Figure 6 shows the corresponding changes in surface [NO₃⁻] for Europe, similar to the changes
for HNO₃. Typical [NO₃⁻] values are almost twice those of [HNO₃(g)]. Unlike for HNO₃(g), a concave seasonal
cycle is evident in the weekly observational composites, with lower concentrations of around 1 µg/m³ during
summer compared to winter. Both CY48R1 and CY49R1_NOE4C fail to capture the correct seasonality, showing
higher concentrations in summer, resulting in substantial positive biases of 1-2 µg/m³. The associated biases in
[HNO₃(g)] indicate that the HNO₃-NO₃⁻ partitioning is poorly captured. In CY49R1, the seasonal cycle description



is improved by EQSAM4Clim, resulting in much lower biases (< 0.5 µg/m³) throughout the year, highlighting the
importance of better gas/particle partitioning representation. The bottom panel of Figure A8 in the Appendix shows
the corresponding changes in [NO₃⁻] against weekly composites from the CASTNET measurement network.
Strong similarities are seen with the improvements observed in the European comparison. In CY48R1 and
CY49R1_NOE4C, no seasonal variability occurs in [NO₃⁻], leading to significant positive biases of 1.5-2.0 µg/m³.
In CY49R1, biases decrease by an order of magnitude, and seasonal variability improves markedly.

**Table 7:** The tropospheric HNO₃(g) budget in Tg N/year for 2018, as calculated by CY48R1 and CY49R1, with relative differences shown as (CY49R1 - CY48R1)/CY48R1.

| Process | CY48R1 | CY49R1_NOE4C | CY49R1 |
|---|---|---|---|
| $NO_2 + OH \rightarrow HNO_3$ | 11.0 | 11.6 (+5) | 12.6 (+14) |
| $N_2O_5 + Liq \rightarrow HNO_3$ | 2.2 | 2.5 (+12) | 2.6 (+17) |
| $N_2O_5 + Aer \rightarrow HNO_3$ | 3.2 | 2.3 (-28) | 2.3 (-27) |
| $NO_3 + Aer \rightarrow HNO_3$ | 0.8 | 0.4 (-47) | 0.5 (-42) |
| $HNO_3 \rightarrow$ Fine NO₃- | 1.4 | 1.2 (-18) | 2.0 (+41) |
| $HNO_3 \rightarrow$ Coarse NO₃- | 9.3 | 5.9 (-36) | 3.6 (-17) |
| Dry Deposition | 2.0 | 2.4 (+17) | 5.1 (+150) |
| Wet Deposition | 6.8 | 5.9 (-13) | 9.3 (+38) |
| Trop. Burden | 0.31 | 0.30 (-3) | 0.32 (+3) |

In Figure 7, we show the regional distributions of annual mean [NO₃⁻] for CY48R1 and CY49R1 during 2018 for
the three chosen regions, with the associated changes in regional annual mean statistics provided in Table 8. Some
commonality exists between the changes shown for annual mean [NH₄⁺] and [NO₃⁻], due to the speciation of the
SIA involved. The cumulative sums of smaller nitrate particles (fine mode NO₃⁻ in Table 4, in the form NH₄NO₃)
and larger nitrate particles (coarse mode NO₃⁻ in Table 4, in the form of CaNO₃ and NaNO₃) are included in the
plots. Therefore, the changes evaluated here represent a combination of changes in both fine and coarse mode
NO₃⁻, rather than changes in individual particle sizes. Unlike the reduced nitrogen analysis provided above, which
is impacted more directly by changes in fine mode NO₃⁻, [NO₃⁻] is also indirectly affected by coarse mode
assumptions through the effect of cations on neutralization levels, which subsequently control gas/aerosol
equilibrium partitioning. Changes in HNO₃ partitioning result in a reduction of NOₓ in particulate form, due to a
higher dry deposition component.
In Europe, the simulated annual mean [NO₃⁻] in CY48R1 generally ranges from 0.2-1 µg/m³ over Scandinavia,
Spain, and surrounding seas, and from 2-6.3 µg/m³ over northwestern and central Europe and the Mediterranean,
with lower values towards the northeast and southwest. The highest European NOₓ emissions occur in the southeast
UK, Benelux, the Ruhr, and Po valleys (e.g., Liu et al., 2021; van der A, 2024). This, and the relatively
homogeneous distribution within central Europe, shows significant transport once NO₃⁻ particles are formed. No
such continental gradient in annual mean [NO₃⁻] exists in the observational mean values, indicating an
overestimate in IFS-COMPO. Nevertheless, in CY49R1, decreases of 2-4 µg/m³ in [NO₃⁻] occur for the Baltic
states, France, Germany, and the Mediterranean Sea (from relatively high shipping NOₓ emissions), resulting in
better agreement with the annual mean observed values at individual measurement stations. The annual regional
MB decreases by ~90%, dropping to 0.1 µg/m³ in CY49R1, with an associated increase in the correlation
coefficient due to lower transport of [NO₃⁻] out of the main source regions. A large impact is observed due to the
acidification of sea salt aerosols under relatively high NOₓ emissions from dense shipping lanes, which can be seen
by the similar [NO₃⁻] reductions over the sea, though these are difficult to evaluate due to insufficient
measurements.





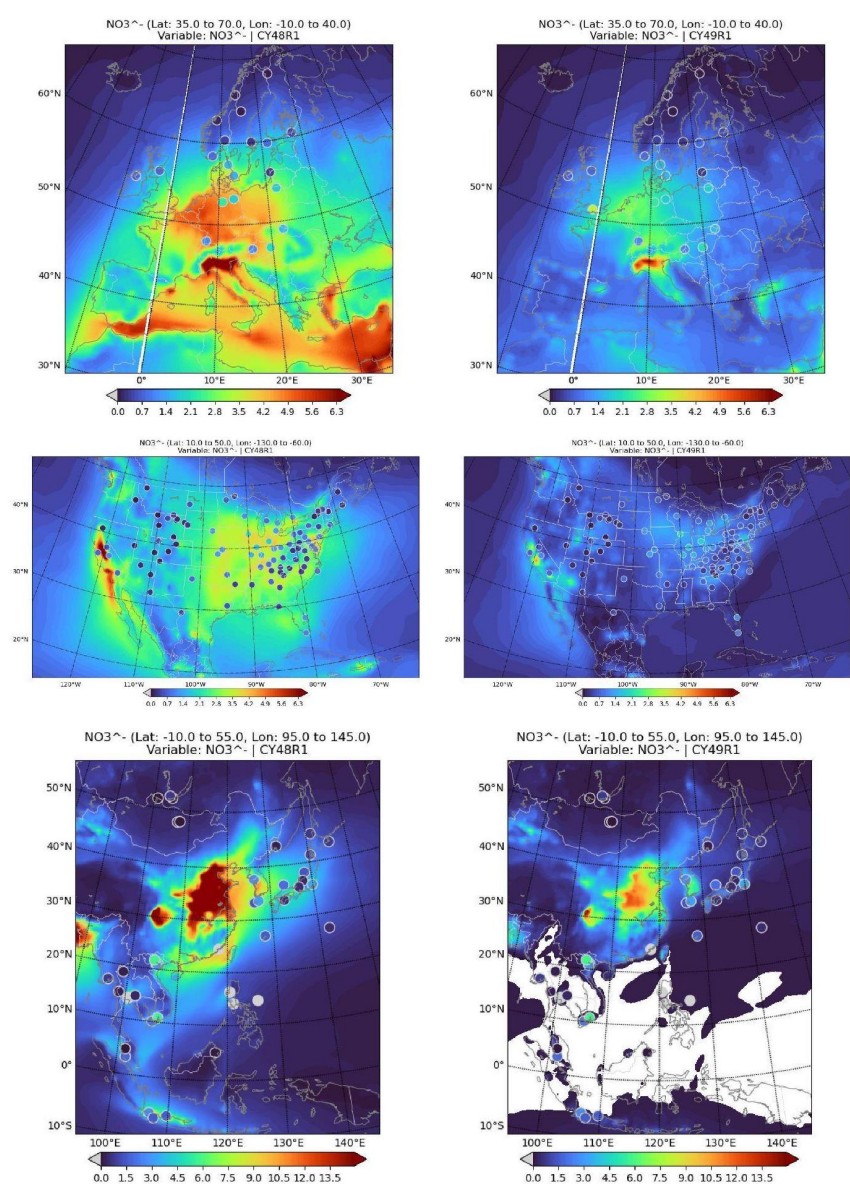

**Figure 7:** Annual mean comparisons of [NO₃⁻] simulated in CY48R1 and CY49R1, compared against measurements for the three selected regions, given in µg/m³. The corresponding regional statistics are provided in Table 8.





**Table 8**: As for Table 2, but for NO₃⁻.

| NO₃⁻ | Europe (EMEP) | | US (CASTNET) | | SE Asia (EANET) | |
|---|---|---|---|---|---|---|
| | CY48R1 | CY49R1 | CY48R1 | CY49R1 | CY48R1 | CY49R1 |
| MB (ug/m³) | 0.95 | 0.10 (-90%) | 1.71 | 0.10 (-94%) | 2.68 | -0.16 (-94%) |
| RMSE | 2.37 | 1.60 (-32%) | 2.20 | 0.83 (-62%) | 3.62 | 1.41 (-61%) |
| Pearsons R | 0.41 | 0.58 (+29%) | 0.31 | 0.57 (+46%) | 0.65 | 0.52 (-20%) |

In the U.S., a similar impact on [NO₃⁻] is observed as in Europe, where the high annual MB in [NO₃⁻] decreases significantly (94%) from CY48R1 to CY49R1. In CY48R1, [NO₃⁻] typically ranges from 2–4 µg/m³, with medium‑range transport resulting in appreciable concentrations over the surrounding oceans. Considering the precursors, there is surprisingly little variability in the observed annual mean [NO₃⁻], despite the large difference in resident [HNO₃(g)] across different states of the U.S., related to the distribution of NOₓ emissions (see Figure 5; Goldberg et al., 2021). Only in the southwest, around California, are annual mean [NO₃⁻] values > 2.0 µg/m³, whereas typical annual mean [NO₃⁻] values in CY49R1 are ≤ 1.0 µg/m³ for most of the U.S. This implies that the cations used as input for EQSAM4Clim impose a limit on the phase transfer of HNO₃(g) into more acidic aerosols through neutralization of anions by cations in the particle phase.

**5. The changes in regional wet deposition**

In this section, we evaluate the temporal distribution and biases associated with the annual wet deposition of soluble trace gas species and particulates. All three SIA species are lost to the surface via both dry and wet deposition processes. Over the last few decades, the main source of acidification has shifted from SOₓ-based to NOₓ-based, following the reduction measures for SOₓ and increased emissions from sectors such as road transport. Here, we assess whether the current version of IFS-COMPO captures the correct wet scavenging for the various dissolved precursors and SIA. Evaluations are based on comparisons of model output against annual wet deposition totals from observational networks. The concentrations of the dissolved precursors (i.e., SO₂(aq), NH₃(aq), and HNO₃(aq)) also undergo wet deposition (in IFS-COMPO) and cannot be differentiated in the observational networks, but are included in the measured totals. The wet deposition term is influenced by meteorological parameters such as simulated large-scale and convective mixing, liquid and solid precipitation droplet size, SAD (Surface Area Density), and the frequency and intensity of precipitation provided by the IFS model.

In Table 9, we present the changes in the global tropospheric burden, lifetime, and dry and wet deposition totals for SO₄²⁻, NH₄⁺, and NO₃⁻ (fine and coarse) during 2018 across all simulations. The corresponding statistics for the annual wet deposition means of SOₓ, reduced N, and oxidized N are provided for the three selected global regions in Table 10. The locations of the measurement sites are similar to those used for the SIA concentration evaluations and have similar constraints with respect to representativity for the area. The stations' locations are shown in the following figures, allowing for a direct comparison of the annual values without complications from different sampling regimes regarding spatial representation.

**Table 9**: The global budget values for the burden, tropospheric lifetime, wet and dry deposition terms for SO₄²⁻, NH₄⁺, and NO₃⁻ in 2018. Totals are provided in Tg S/year and Tg N/year. Percentage difference changes are given in parentheses.

| | CY48R1 | CY49R1_NOE4C | CY49R1 |
|---|---|---|---|
| SO₄²⁻ | | | |
| Burden | 0.4 | 0.6 (+30) | 0.6 (+30) |





| | CY48R1 | CY49R1_NOE4C | CY49R1 |
|---|---|---|---|
| Lifetime (days) | 3.4 | 4.4 (+29) | 4.4 (+29) |
| Dry dep | 5.8 | 5.4 (-7) | 5.4 (-7) |
| Wet dep | 43.1 | 43.9 (+2) | 44.2 (+3) |
| $NH_4^+$ | | | |
| Burden | 0.3 | 0.4 (+32) | 0.2 (-33) |
| Lifetime (days) | 3.5 | 4.6 (+34) | 4.1 (+18) |
| Dry dep | 5.1 | 5.1 (-) | 1.8 (-64) |
| Wet dep | 27.5 | 27.1 (-2) | 20.1 (-40) |
| $NO_3^-$ (fine) | | | |
| Burden | 0.01 | 0.01 (-) | 0.02 (+86) |
| Lifetime (days) | 4.9 | 5.4 (+12) | 6.1 (+25) |
| Dry dep | 0.2 | 0.2 (-21) | 0.1 (-32) |
| Wet dep | 0.6 | 0.5 (-13) | 1.0 (+64) |
| $NO_3^-$ (coarse) | | | |
| Burden | 0.01 | 0.01 (-) | 0.01 (-) |
| Lifetime (days) | 3.4 | 3.8 (+11) | 2.4 (-29) |
| Dry dep | 1.5 | 2.5 (+73) | 1.2 (-18) |
| Wet dep | 3.5 | 2.8 (-20) | 0.7 (-79) |


**Table 10**: The annual MB, RMSE, and Pearson's R values for the comparisons of weekly mean regional wet
deposition totals of dissolved $SO_2 + SO_4^{2-}$, $NH_3 + NH_4^+$, and $HNO_3 + NO_3^-$, compared against composites
assembled from the regional observation networks for 2018 shown in Figures 8-10 for Europe, the U.S., and
Southeast Asia. Percentage difference changes are calculated as $((CY49R1 - CY48R1)/CY48R1) * 100$.

| | Europe (EMEP) | | US (CASTNET) | | SE Asia (EANET) | |
|---|---|---|---|---|---|---|
| $SO_X$ | CY48R1 | CY49R1 | CY48R1 | CY49R1 | CY48R1 | CY49R1 |
| MB (mgS/m²/yr) | -42 | -38 (-9) | 137 | 190 (+39) | -44.2 | 8.7 (-80) |
| RMSE | 88.2 | 85 (-3) | 203 | 270 (+33) | 447 | 500.3 (+12) |
| Pearsons R | 0.55 | 0.58 (+6) | 0.68 | 0.66 (-3) | 0.72 | 0.65 (-10) |
| Reduced N | | | | | | |
| MB (mgN/m²/yr) | 61 | 25.9 (-58) | 8.4 | 6.8 (-21) | 12 | -44 (+260) |
| RMSE | 114 | 93.4 (-18) | 76.0 | 81.3 (+7) | 318 | 302 (-5) |
| Pearsons R | 0.69 | 0.68 (-1.4) | 0.77 | 0.72 (-16) | 0.75 | 0.71 (-1) |
| Oxidised N | | | | | | |
| MB (mgN/m²/yr) | 9.7 | -1.4 (-86) | 130 | 99.7 (-23) | 142 | 98.3 (-31) |
| RMSE | 69 | 72 (+4) | 153 | 122.6 (-20) | 324 | 274.3 (-15) |
| Pearsons R | 0.50 | 0.47 (-6) | 0.86 | 0.85 (-1) | 0.67 | 0.68 (+2) |







### 5.1 Total annual wet S deposition

Figure 11 shows the regional distribution of annual wet S deposition for Europe, the U.S., and Southeast Asia in both CY48R1 and CY49R1 during 2018. To allow direct comparison across regions, we use a color scale covering values up to 1000 mg S/m²/year. The global budget terms for $SO_4^{2-}$ are presented in Table 9, showing that despite the global burden increasing by one third, only small increases of a few percent occur in the annual wet $SO_4^{2-}$ totals (Rémy et al., 2024). However, the significant increase in the tropospheric $SO_4^{2-}$ lifetime means that more remains in the aerosol phase, impacting the degree of scattering in IFS-COMPO, as shown in AOD comparisons in Rémy et al. (2024). The most significant change is in the direct gas-phase production of $H_2SO_4(g)$, where increases in $[SO_2(g)]$ subsequently increase the total mass scavenged into aqueous cloud droplets. This results in some acidification (slowing in-situ oxidation, cf. Table 3), buffered somewhat by increased phase transfer of $NH_3(g)$ (cf. Table 5). Although there is a 15% reduction in global $SO_2(aq)$ wet deposition, increases in $[SO_4^{2-}(aq)]$ result in an increase in the cumulative wet S deposition totals.

In Europe, the changes between model simulations are similar to those for $SO_2(g)$ and $SO_4^{2-}$ particle concentrations discussed in Section 4. Compared to the annual EMEP observational mean values, which range from 100-900 mg S/m²/year, CY48R1 generally underestimates values by approximately 100-150 mg S/m²/year in northwest Europe, Poland, and the Iberian Peninsula. In other regions, agreement is good, capturing the observed deposition gradient from Germany into Austria and northern Italy. A limited number of measurement stations exhibit very high localized values (e.g., southwest Ireland, Palma), indicating missing primary emission sources in the global inventory. In CY49R1, strong similarities are observed for Benelux, Denmark, and Italy, with negative biases of around 50-100 mg S/m²/year. A significant negative annual MB exists in Europe, decreasing by around 10 mg S/m²/year in CY49R1 (cf. Table 10), with a marginal increase in correlation. This is influenced by the associated negative MB for $SO_2(g)$ during summer (cf. Figure 1) and the large values observed at selected stations influencing the regional mean.

In the U.S., there is a stark contrast to Europe. CASTNET annual mean values show an observational gradient in wet deposition totals, similar to the primary $SO_2$ emission sources (cf. Figure A1 in the Appendix), with maximum values reaching 300-400 mg S/m²/year toward the East Coast. CY48R1 captures this gradient well but with large positive biases of >100 mg S/m²/year, resulting in maximum values of 700-900 mg S/m²/year. Significant annual wet deposition occurs in the Atlantic (250-300 mg S/m²/year) due to the oxidation of DMS (released from the ocean) and long-range transport of $SO_2(g)/SO_4^{2-}$ from anthropogenic source regions. In CY49R1, the area of maximum wet S deposition increases around regions like New York State, resulting in a 40% increase in positive annual MB to 190 mg S/m²/year. This contrasts with the significant improvement in the annual MB for $[SO_4^{2-}]$, as shown in Table 7, indicating an increase in scavenging into the aqueous phase of $SO_4^{2-}$ particles due to other cumulative updates in IFS-COMPO (cf. Table 9), partly due to a 10% increase in gas-phase $SO_2$ to $H_2SO_4$ (cf. Table 9).

In Southeast Asia, EANET annual wet deposition totals show that more than double the amount of S deposition occurs compared to Europe or the U.S., reaching 1200-1300 mg S/m²/year in central China and Indonesia. The temporal distribution of stations shows a positive gradient between deposition totals in China and those extending toward Indonesia (2000-2200 mg S/m²/year, not shown). This highlights the importance of $SO_4^{2-}$ transport, considering the low regional $SO_2(g)$ precursor mixing ratios near the equator (cf. Figure A1 in the Appendix), with primary sources being infrequent volcanic eruptions that typically inject $SO_2$ above the boundary layer (thus with limited surface impact).

Along the eastern coast of China, observations show annual totals of 250-350 mg S/m²/year, contrasting with higher values in central China. This is surprising, considering that high $SO_2$ emissions are defined in IFS-COMPO around South Korea rather than central China. This implies that the regional $SO_2$ emissions employed for this region may be overestimated, given the low regional deposition values. The regional annual MB improves markedly to 8.7 mg S/m²/year, which is very low given the high values in the measurements. However, the correlation coefficient degrades from 0.72 to 0.65.





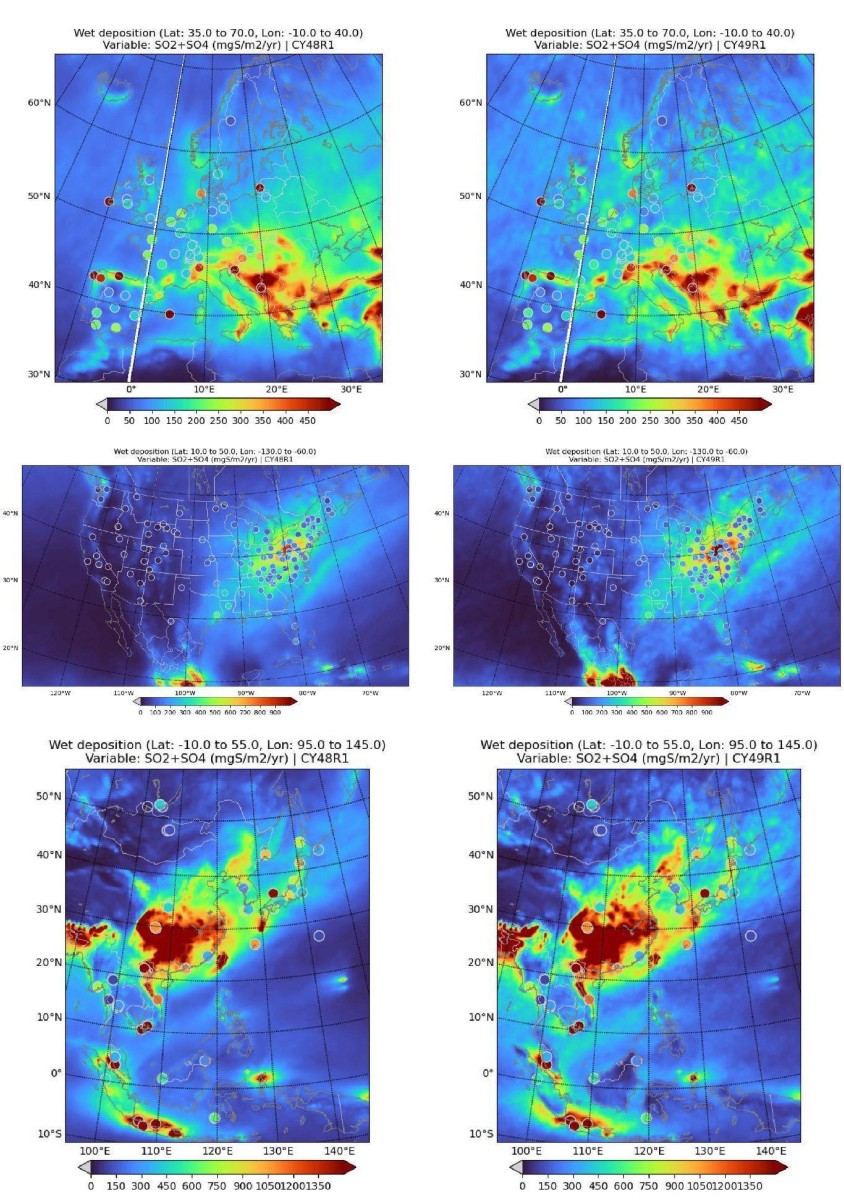

711

**Figure 8:** Annual comparisons of the cumulative wet deposition totals of dissolved $SO_2$ and $SO_4^{2-}$ aerosol (mg S/m²/year) for 2018, simulated in CY48R1 (left column) and CY49R1 (right column), shown for Europe (top), the U.S. (middle), and Southeast Asia (bottom). The corresponding statistics are provided in Table 9.

716

717

718





### 5.2 Total annual wet NH$_x$ deposition

Figure 9 shows the corresponding changes in total annual mean wet deposition of reduced N for both CY48R1 and CY49R1 during 2018. The sampling stations are the same as those used for the wet S deposition evaluation. In Table 9, the global chemical budget terms for NH$_4^+$ show that cumulative updates to IFS-COMPO increase the tropospheric burden by one third (similar to SO$_4^{2-}$, as (NH$_4$)$_2$SO$_4$ is a dominant SIA species, Seinfeld and Pandis, 2006). However, this is reversed when applying EQSAM4Clim for aerosol and cloud pH, as shown when comparing CY48R1 and CY49R1. This results in significant decreases in both global dry and wet deposition totals (>50%, cf. Table 9) across the three regions.

In Europe, where high summertime NH$_3$(g) mixing ratios are simulated (cf. Figure 4 in Section 4), EMEP observational annual wet deposition totals show peak values in the Balkans and northern Italy (Po Valley), with regional variability in France (250-350 mg N/m²/year). In regions with low NH$_3$ emissions, such as Scandinavia and the Iberian Peninsula, wet deposition totals range from 50-200 mg N/m²/year. In CY48R1, high surface NH$_3$(g) mixing ratios (5-15 ppb; see Figure A3 in the Appendix) result in relatively high NH$_x$ annual wet deposition totals of 350-500 mg N/m²/year for northwest and central Europe (e.g., Benelux, Austria). Measured annual mean values are typically exceeded, resulting in an annual MB of 61 mg N/m²/year, albeit with a high correlation (0.69, cf. Table 9). The continental distribution is well represented, though high values extend too far east and west of Europe. In CY49R1, the area with maximum values (>450 mg N/m²/year) shrinks. The reduction in [NH$_4^+$] (cf. Table 5) decreases the annual MB in wet deposition by nearly 60%, without degrading the correlation coefficient. The application of EQSAM4Clim significantly improves the simulation of reduced N wet deposition in IFS-COMPO for Europe.

In the U.S., CASTNET observations show a similar east-west gradient in total reduced N wet deposition as seen in NH$_3$(g) surface mixing ratios and [NH$_4^+$] distributions (cf. Figure A3 and Figure 9, respectively). Observed wet deposition values range from 30-400 mg N/m²/year, indicating that deposition levels are lower where local NH$_3$ emission sources are absent (lower than in Europe). In CY48R1, the continental gradient is captured, though maximum values in Iowa are not observed in the measurements (>100% MB), influenced by high local NH$_3$ emission flux (cf. Figure A3). On the East Coast, where most NH$_3$ sources are located, CY48R1 generally overestimates wet deposition. Compared to Europe, the annual MB for the U.S. is low (9 mg N/m²/year), reflecting large positive biases on the East Coast, moderated by underestimates elsewhere. A high correlation (R=0.77) is achieved in CY49R1. Although NH$_4^+$'s spatial distribution remains similar between cycles, the reduction in [NH$_4^+$] reduces the annual MB by 21%, with a slight degradation in correlation.

In Southeast Asia, EANET observational annual wet deposition totals are higher than in Europe and the U.S., ranging from 200-2400 mg N/m²/year (not shown), with the highest values in Indonesia and Vietnam. The simulated temporal distribution of reduced N wet deposition captures the variability across individual stations well across a wide area. In CY48R1, the annual MB is 12 mg N/m²/year on high annual totals, making it the lowest MB among the regions, with a high correlation (0.75). In CY49R1, there is a larger negative MB (though still relatively small compared to the large totals), despite the lower positive MB simulated for [NH$_4^+$] compared to CY48R1 (cf. Table 6).



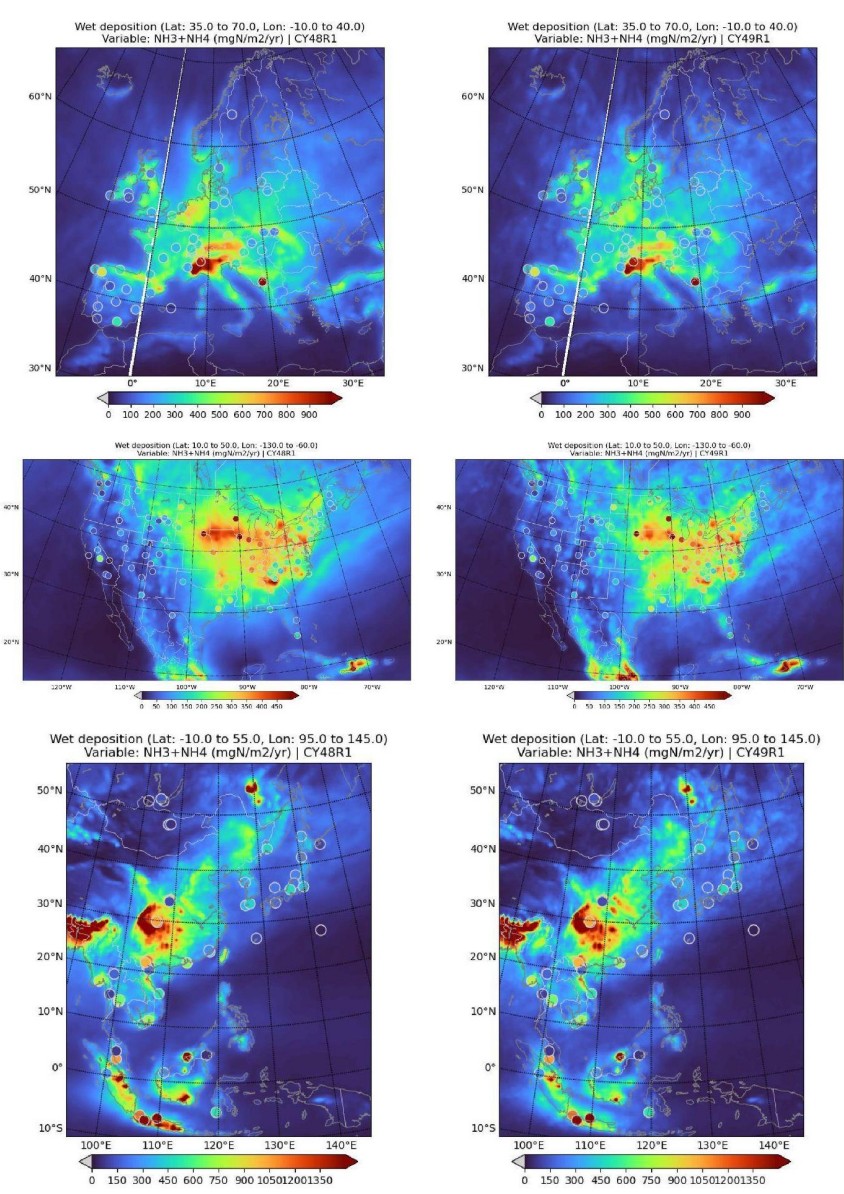

**Figure 9:** Annual comparisons of the cumulative wet deposition totals of dissolved NH₃ and NH₄⁺ aerosol (mg N/m²/year) for 2018, simulated in CY48R1 (left column) and CY49R1 (right column), shown for Europe (top), the U.S. (middle), and Southeast Asia (bottom). The corresponding statistics are provided in Table 9.



## 5.3 Total annual wet NOₓ deposition

Finally, Figure 10 shows the corresponding changes in total annual mean wet deposition of oxidized N for both CY48R1 and CY49R1 during 2018. The global chemical budget terms provided in Table 7 show an increase in the gas-phase production term for $HNO_3$, with a relatively constant heterogeneous conversion term for $N_2O_5$ when summed over various reactive surfaces. Once formed, a significant fraction of $HNO_3$ is directly scavenged into aqueous cloud droplets and deposited as wet (acidic) deposition (cf. Rémy et al., 2024). However, the large biases in $HNO_3(g)$ reveal a limit to the wet scavenging term, leaving an excess in the gas phase, impacting the results in this section. Note that particulate $NO_3^-$ takes various chemical forms in IFS-COMPO ($Ca(NO_3)_2$, $NaNO_3$, $NH_4NO_3$), so there is only partial commonality between changes in $NH_4^+$ and $NO_3^-$. Applying EQSAM4Clim reduces $NH_4^+$ concentrations and burdens, while $HNO_3(g)$ concentrations increase (cf. Table 5 and Figure 6).

In Europe, EMEP observational annual wet deposition totals of oxidized N range from 150-275 mg N/m²/year, correlating with the homogeneous distribution of $[NO_3^-]$ (see Section 4.3). A few high outliers are likely influenced by strong local NOₓ emissions. Although modest differences in cumulative wet deposition of $[NO_3^-]$ occur between CY48R1 and CY49R1, the regional bias in oxidized N wet deposition improves markedly, decreasing by 80% (from positive to negative), though no significant improvement in the (time-sensitive) correlation is observed.

In the U.S., higher values of oxidized N deposition occur on the East Coast, driven by NOₓ emissions (see Figure 13). CASTNET observational wet deposition values range from 50-500 mg N/m²/year, showing a strong longitudinal gradient. This gradient is captured well, though CY48R1 typically overestimates by 100-200 mg N/m²/year, e.g., in New York State and surrounding regions. In the western U.S., observations show values between 0-100 mg N/m²/year, with positive model biases of 100 mg N/m²/year in the northwest states in both versions. In the southern U.S., CY48R1 overestimates by 50-70 mg N/m²/year, which decreases significantly in CY49R1. CY48R1 exhibits a large positive bias of 130 mg N/m²/year, improved by 23% in CY49R1. Again, Pearson's R remains relatively unaffected, indicating the governing influence of the spatial distribution of main point sources and limited impact on forecasts (as IFS-COMPO is not employed here as a fully coupled forecasting system).

In Southeast Asia, EANET observational total wet deposition values range from 50-800 mg N/m²/year, with the highest values (>2000 mg N/m²/year) occurring on the Malaysian coast. In northern China, wet deposition totals of up to 400 mg N/m²/year occur, approximately half of what is observed near the southern coast and eastward. The highest simulated wet deposition totals occur in southwest China, correlating with high NOₓ emissions. Comparing CY48R1 and CY49R1 shows a marked 31% decrease between cycles, again with limited changes to the correlation.



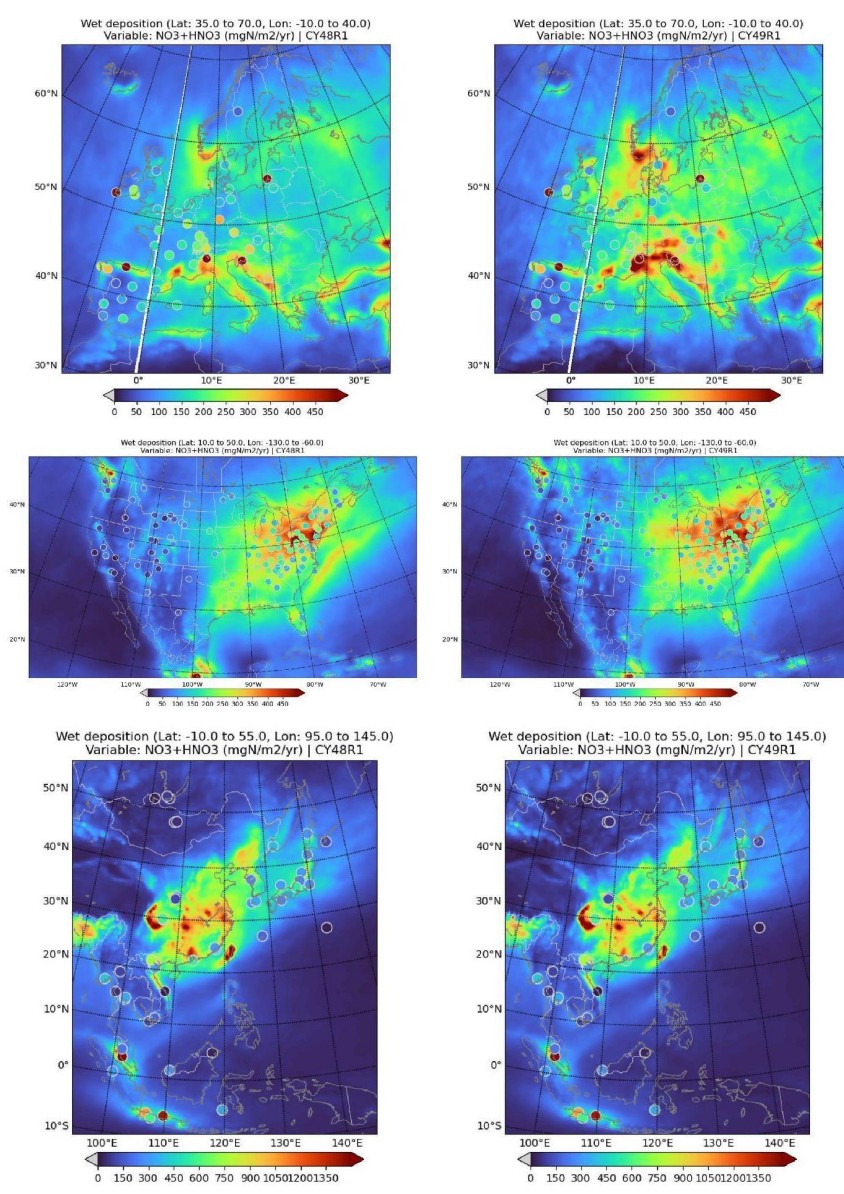

**Figure 10:** Annual comparisons of the cumulative wet deposition totals of dissolved HNO₃ and NO₃⁻ aerosol (mg
N/m²/year) for 2018, simulated in CY48R1 (left column) and CY49R1 (right column), shown for Europe (top), the
U.S. (middle), and Southeast Asia (bottom). The corresponding statistics are provided in Table 9.
**6 Conclusions**
In this paper, we build on previous evaluations of IFS-COMPO CY49R1 performance presented in Rémy et al.
(2024), which assessed the impact of EQSAM4Clim and its revised calculation of aerosol and cloud pH (Metzger
et al., 2024) on improving air quality forecasts by segregating and investigating individual inorganic components.
To scrutinize its effect on atmospheric composition, we compared the current operational IFS version, which
contains a basic description of aerosol and cloud pH (CY48R1), with the next operational IFS version (CY49R1),





which uses EQSAM4Clim in combination with a unified wet scavenging approach and other developments.
Further improvements were also made to both in-cloud and below-cloud scavenging of soluble trace gases and
aerosols through updated parameterizations, as detailed in Rémy et al. (2024).
We have shown that the most significant impacts of the IFS-COMPO updates are related to the production efficacy
of SIA and the subsequent phase partitioning of reduced/oxidized nitrogen species. Comparing simulations with
and without EQSAM4Clim reveals that changes in SIA are primarily caused by alterations in gas/aerosol
partitioning. The verification and analysis are shown for three dominant source regions—Europe, the U.S., and
Southeast Asia—by focusing on surface concentration and wet deposition observations for 2018, compared against
observational composites. Most of the simulated SIA surface concentration and wet deposition fields are improved
by the proposed CY49R1 changes, particularly by the use of EQSAM4Clim.
For $SO_2(g)/SO_4^{2-}$, only moderate changes occur in the conversion rate. For $SO_2(g)$, a 7% increase in the global
tropospheric burden indicates less phase transfer due to limitations in uptake caused by the increase in solution
pH. An increase in the gas-phase production of $H_2SO_4(g)$, which is subsequently scavenged, offsets a modest
reduction in the aqueous-phase production term. For surface $[SO_2(g)]$, this results in a lower mean annual bias for
Europe with moderate correlation, while a higher negative bias with little correlation is observed for the U.S. In
China, no appreciable impact occurs, as a high positive bias of 11.5 µg/m³ is observed with respect to CNEC and
a near-zero correlation coefficient. For $[SO_4^{2-}]$, the tropospheric burden and lifetime increase by one third due to
the IFS-COMPO updates, leading to a reduction in the annual mean biases for Europe and the U.S., along with
increases in the corresponding correlation coefficients. However, in China, performance degrades, with a positive
annual mean bias and a decrease in the correlation coefficient.
For $NH_3(g)/NH_4^+$, the changes are more substantial, resulting in beneficial improvements in global modeling of
reduced nitrogen. For $NH_3(g)$, the tropospheric burden nearly doubles due to a halving of the conversion rate into
$NH_4^+$, with more $NH_3(g)$ being directly deposited to the surface. For surface $[NH_3(g)]$, there is a contrasting change
in the simulated weekly mean bias between Europe and the U.S. In Europe, there is no significant improvement in
the persistent high weekly mean biases, which increase by 10-25% during spring and summer despite all updates.
In the U.S., the lower weekly $[NH_3(g)]$ results in an associated low bias in the simulations, meaning the increase
in the tropospheric burden improves surface comparison markedly. For $[NH_4^+]$ in Europe, EQSAM4Clim's
application results in limited changes in the simulated weekly bias during winter, while significant reductions are
observed in summer, with an associated increase in the annual mean correlation. In the U.S. and China, similar
reductions in the annual mean bias of nearly 50% occur, although the correlation is slightly reduced.
For $HNO_3(g)/NO_3^-$, the changes are similar to those for $NH_3(g)/NH_4^+$ partitioning due to the speciation of SIA,
which is mainly linked via $NH_4NO_3$. Gas-phase production of $HNO_3(g)$ increases without an associated increase
in the global tropospheric burden, due to increased loss to the surface via dry deposition. EQSAM4Clim increases
the fine aerosol component while reducing the coarse aerosol component, which decreases the fraction of $HNO_3(g)$
held in the particulate phase by 50%. In Europe and the U.S., persistent negative biases for $HNO_3(g)$ are changed
to significant positive biases. For $[NO_3^-]$, significant improvements in annual mean biases occur globally, as
illustrated by the three chosen regions, along with improvements in simulated correlation coefficients.
For the wet deposition component, changes in SIA concentrations are qualitatively similar to the annual wet
deposition totals, although regional changes are variable and species-specific. In Europe, reductions are observed
in the simulated annual mean bias for all three chemical types, with oxidized N improving markedly. In the U.S.,
the annual mean bias increases for wet S deposition, while biases for both reduced and oxidized wet N decrease.
In Southeast Asia, there is a marked improvement in wet S deposition, a moderate improvement in oxidized wet
N, and a degradation in reduced wet N. Overall, the recent improvements brought by EQSAM4Clim (Metzger et
al., 2024), as applied here and in Rémy et al. (2024), show that CY49R1 is fit for purpose in capturing regional
particle concentration and loss terms via wet deposition.





**Appendix**

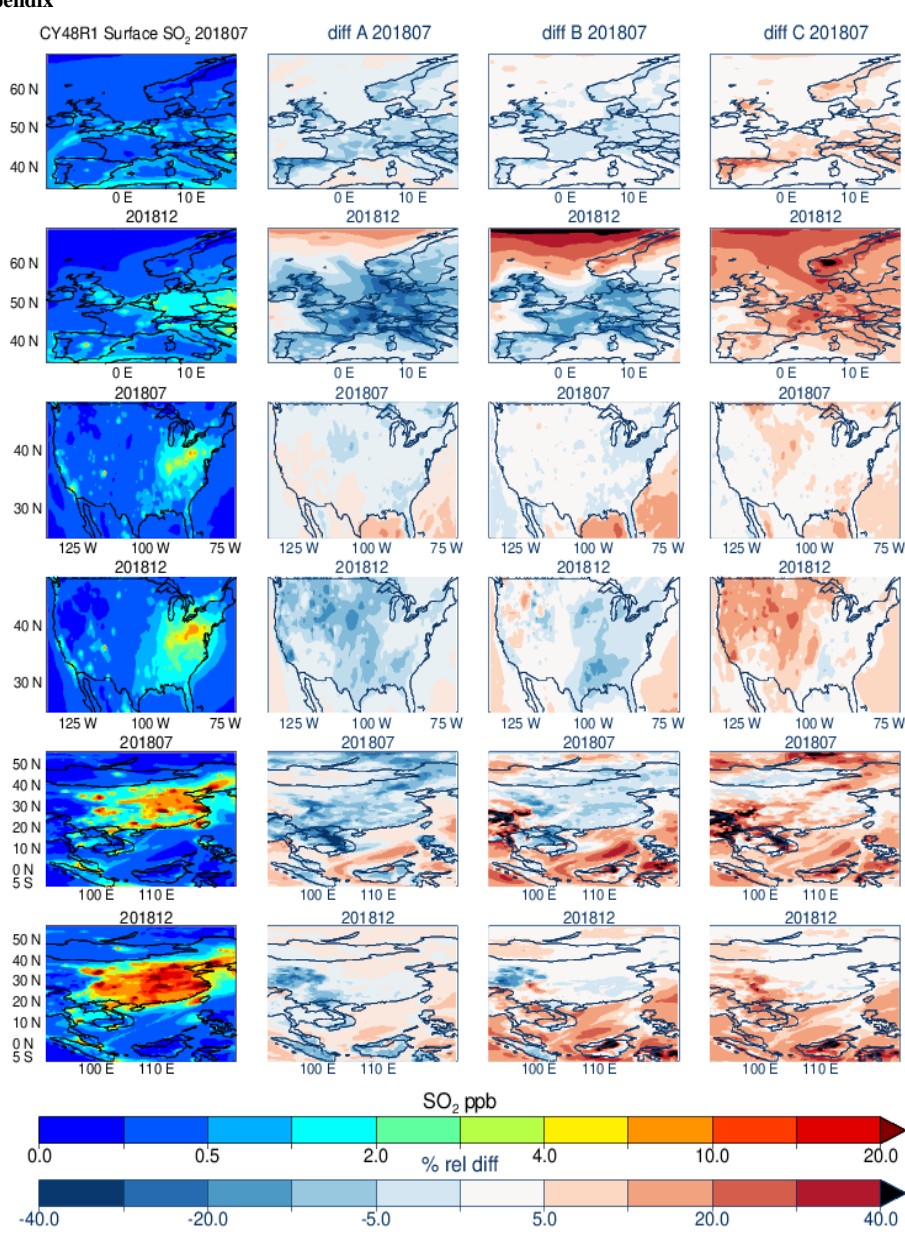

**Figure A1:** The horizontal seasonal mean distribution for surface $SO_2$ for CY48R1 for July and December 2018 for Europe (top), the United States (middle), and Southeast Asia (bottom). The corresponding relative differences are compared against the other simulations. Panel definitions: Diff A = (CY49R1_NOE4C - CY48R1)/CY48R1; Diff B = (CY49R1 - CY48R1)/CY48R1; and Diff C = (CY49R1 - CY49R1_NOE4C)/CY48R1.





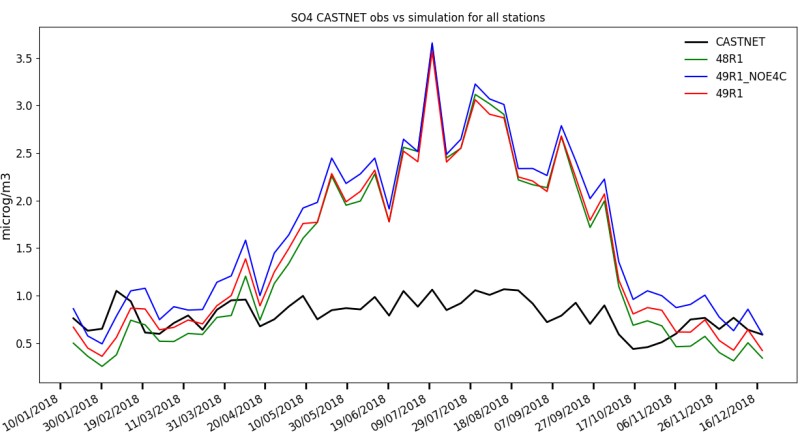

859

**Figure A2**: Comparisons of weekly $SO_4^{2-}$ concentrations (µg/m³) in the U.S. between CASTNET composites and the IFS-COMPO simulations CY48R1, CY49R1_NOE4C, and CY49R1 for 2018.

863


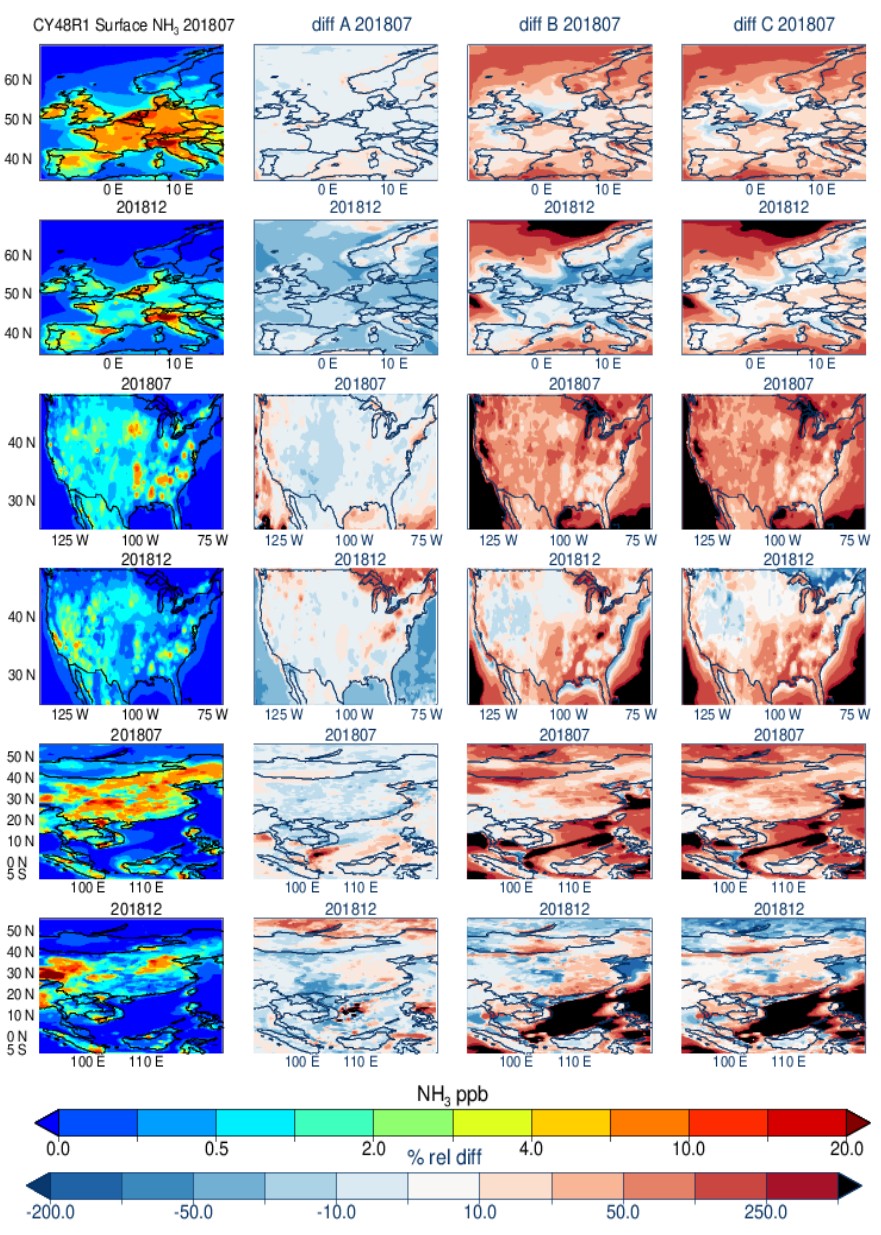

**Figure A3:** As for Figure A1, except for NH₃. Panel definitions: Diff A = (CY49R1_NOE4C - CY48R1)/CY48R1; Diff B = (CY49R1 - CY48R1)/CY48R1; and Diff C = (CY49R1 - CY49R1_NOE4C)/CY48R1.



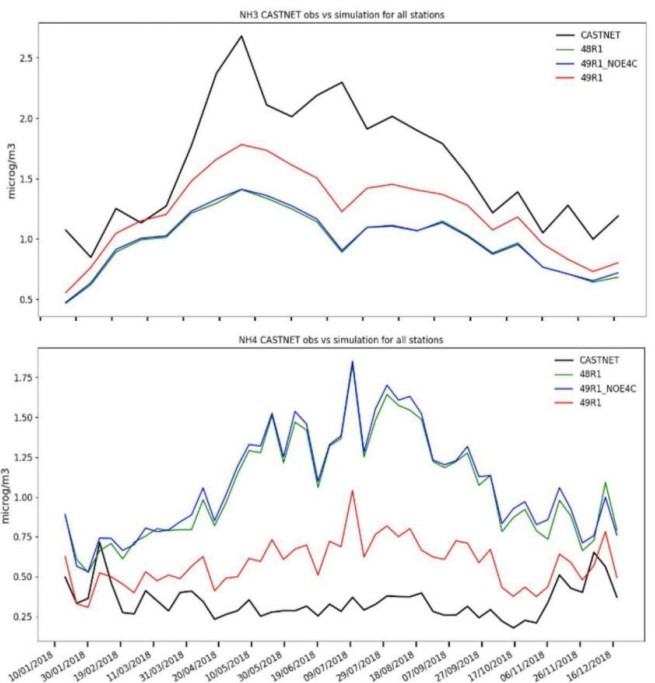

**Figure A4** : Comparisons of weekly NH₃ and NH₄⁺ concentrations (µg/m³) in the U.S. between CASTNET composites and the IFS-COMPO simulations CY48R1, CY49R1_NOE4C, and CY49R1 for 2018.

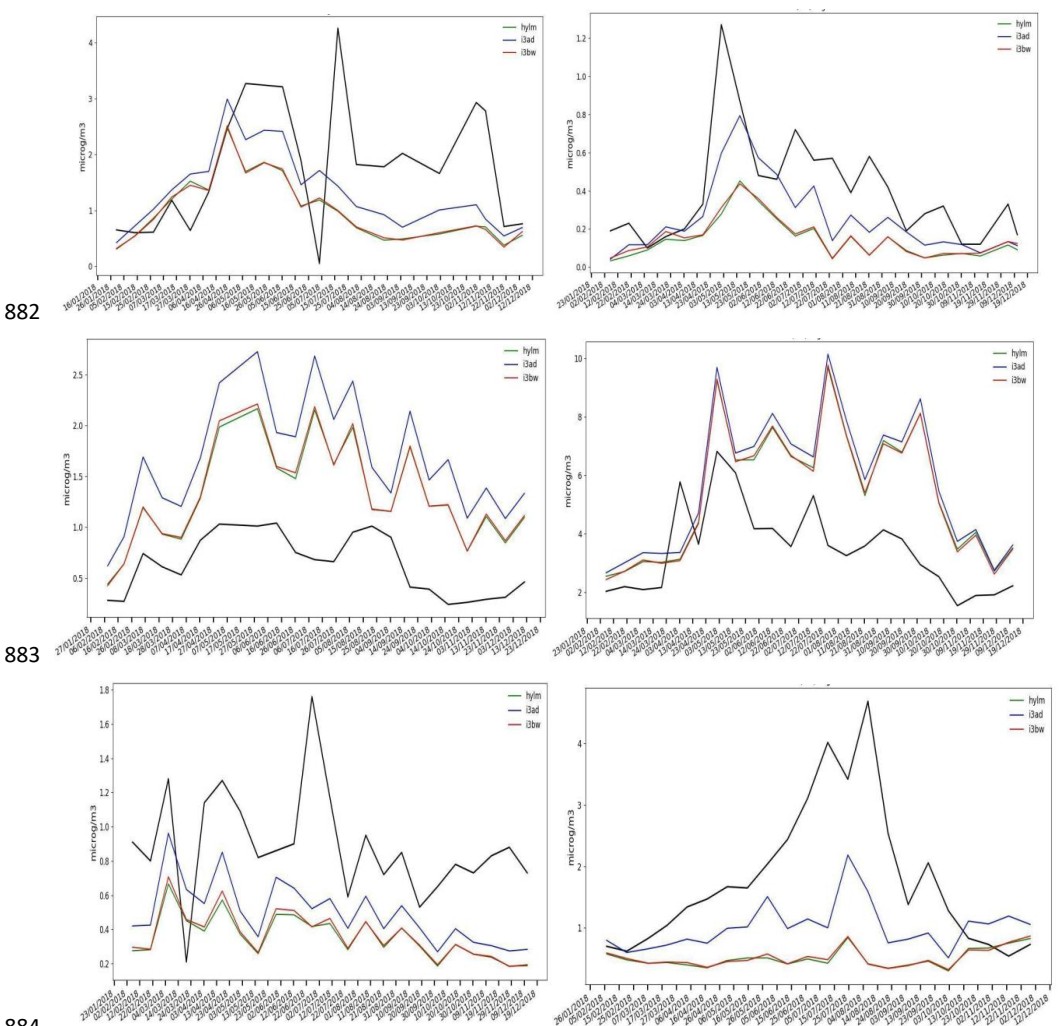

**Figure A5:** Comparisons of weekly mean NH₃ concentrations (μg/m³) from CY48R1, CY49R1_NOE4C, and CY49R1 against measurements from selected stations participating in the AMoN network for 2018. From top left to bottom right, station ID OH09 (Oxford, Ohio, 39.53°N, 84.72°W), NY98 (Whiteface Mountain, New York State, 44.39°N, 73.85°W), AR03 (Caddo Valley, Arizona, 34.17°N, 93.10°W), AL99 (Sand Mountain, Alabama, 34.29°N, 86.0°W), CA67 (Joshua Tree National Park, California, 34.1°N, 116.39°W), and FL19 (Indian River, Florida, 27.85°N, 80.45°W).

**Table A1**: Statistics for the regional distribution of gaseous NH₃ in the U.S. compared against a composite of measurements from all 18 stations participating in the AMoN measurement network for 2018. Relative percentage differences are included as (CY49R1 - CY48R1)/CY48R1.

| Diagnostics | CY48R1 | CY49R1_NOE4C | CY49R1 |
|---|---|---|---|
| MB (μg/m³) | -0.50 | -0.49 (+2.0) | -0.26 (+48.0) |
| RMSE | 1.79 | 1.79 (-) | 1.71 (+4.4) |
| Pearsons R | 0.49 | 0.49 (-) | 0.52 (+6.1) |







**Figure A6:** As for Figure 1, except for HNO₃. Panel definitions: Diff A = (CY49R1_NOE4C - CY48R1)/CY48R1; Diff B = (CY49R1 - CY48R1)/CY48R1; and Diff C = (CY49R1 - CY49R1_NOE4C)/CY48R1.





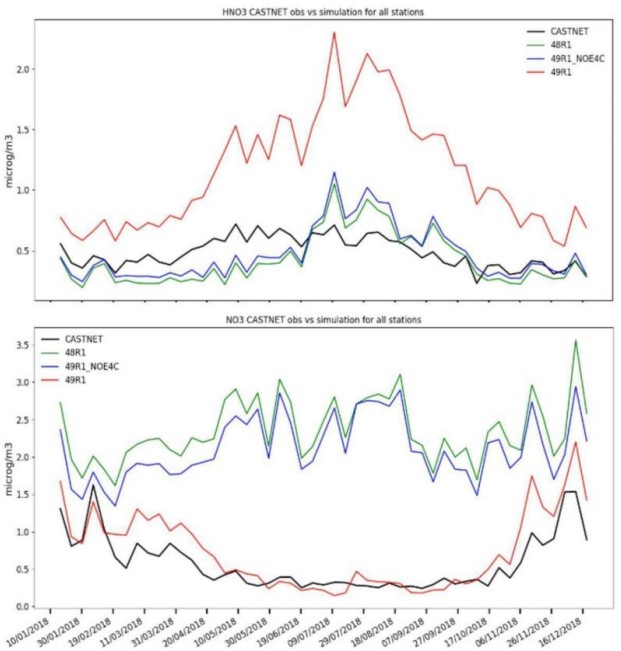

899

**Figure A7**: Comparisons of weekly $HNO_3$ and $NO_3^-$ concentrations ($\mu g/m^3$) in the U.S.
between CASTNET composites and the IFS-COMPO simulations CY48R1,
CY49R1_NOE4C, and CY49R1 for 2018.

*Author Contributions*

JEW and SM were the principal authors of the paper and produced most of the figures. SR conducted the IFS-
COMPO simulations and performed the regional comparisons made against observational datasets for evaluating
the deposition fluxes. SM provided and integrated EQSAM4Clim for the more accurate calculation of pH in
aerosols and clouds. VH updated the model towards CY49R1 and handled technical updates with respect to the
implementation of EQSAM4Clim. JF is a representative of the CAMS consortium under which this work was
conducted.

*Code and Data Availability*

Model codes developed at ECMWF are the intellectual property of ECMWF and its member states and are
therefore not publicly available. ECMWF member-state weather services and their approved partners may be
granted access. Access to a version of IFS (OpenIFS) that includes this experimental cycle may be obtained from
ECMWF under an OpenIFS license. More details can be found at
https://confluence.ecmwf.int/display/OIFS/About+OpenIFS. The surface data for IFS-COMPO used for this study
is available on Zenodo (https://doi.org/10.5281/zenodo.13902673).

*Competing Interests*

At least one of the co-authors is a member of the editorial board of Geoscientific Model Development.

*Acknowledgements*

We acknowledge funding from the Copernicus Atmosphere Monitoring Service (CAMS), which is funded by the
European Union's Copernicus Programme. We also acknowledge the EMEP, EANET, AirNow, CASTNET,
AMoN, and AirBase monitoring networks for providing access to surface observational data for $SO_2$, $SO_4^{2-}$, $NH_3$,
$NH_4^+$, $HNO_3$, and $NO_3^-$.



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
