# Peer review of "An evaluation of the regional distribution and wet deposition of"

_Geoscientific Model Development, 2024_

## Referee Comment (RC2)

**Anonymous referee comments on**

**"An evaluation of the regional distribution and wet deposition of secondary inorganic aerosols and their gaseous precursors in IFS-COMPO cycle 49R1"**

**by J.E. Williams et al.**

This publication presents the recent developments in IFS-COMPO about secondary inorganic aerosol formation representation through EQSAM4Clim module and wet deposition. After quickly presenting model configuration and used observational data, authors presents a validation of secondary inorganic aerosols and their precursors surface concentrations. Last part is about wet deposition of SIA compared to observations.

**General comments:**

This study is very similar to the Remy et al., 2024, with the first part that only focuses on the performances of SIA modelling in IFS-COMPO. The novelty that is the wet deposition treatment is barely processed. I would recommand to be more specific about the changes in the code. A detailed description of the wet deposition scheme along with explanation about the choices and the expected results. The wet deposition analysis has to be longer than the first part that can be reduced. Also the CY48R1_NOEAC which allow to separate the effect of EQSAM4Clim from the wet deposition (and other changes?) needs to be more used to explain and analyse the impact of changes on wet deposition. Also, it would be interesting to have global maps of wet deposition even if there are no measurement at global scale.

The global structuration of the text need to be revised and an in-depth proofreading needs to be done before submitting. It is frightening to see as many wrong figure and table references in a submitted paper. Also please read the GMD author guidelines and try to follow them:

https://www.geoscientific-model-development.net/submission.html

Concerning the figures, their quality is overall poor, especially in the Appendix. I would recommand to do an effort to increase their quality and readibility. Also the partitioning between the paper and appendix should be questioned. Figure A1 for example is used several times and is worth to appear in the main part of the paper.

As you are not using the "real" CY49R1, but modified version of CY48R1, I would recommand to delete all reference to CY49R1 expect from the fact that the presented modifications are meant to enter in the latter.

A section is missing to explain how are computed the statistics. Are all the data gathered in one vector and the tables represent spatio-temporal statistics?

About tables presenting budget, I would recommand to check whether the conservativity is assured and to discuss the possible explanation for differences. Generally speaking you should check different number in tables. I spotted a few problems listed in the specific comments, but I didn't bother to check everything.

I would recommand a large amount of work on this paper before publication.

**Specific comments:**

- Line 126: "which will be operational in November 2024" : Please update this sentence as the date is now expired.

- Table 1: The Experiment IDs are not used in the text. Please remove them or add a sentence to tell that they can be used to retrieve the data on the MARS storage system.

- Line 205: Please add a reference for the direct production from hot shipping exhausts.

- Line 211: AirBase is no longer used since mid 2000's. The new system is called AQeR (Air Quality e-Reporing)

- Line 214: How did you select "rural background stations"?

- Line 271: Figure A1. Please introduce the figure before referencing it. Also Fig. A1 is used a lot. Maybe it would be interesting to move it to the main part of the paper.

- Line 272: Please introduce Table 3 before using it.

- Line 272: "For CY49R1 […] by approximately.1-0.2 µg/m3" → I assume you are talking about Fig. 2 results. Please introduce it and announce it properly or move the sentence in the SO4-- paragraph.

- Line 274: Figure A1.

- Line 279: You introduce Table 3, but you used it only in the next paragraph.

- Line 306: "Statistics relate to seasonal means" → Please be more precise about this sentence. Maybe in the section 3 when introducing observational data.

- Table 2: For Europe, according to the section 3, data must be AQeR (replacing AirBase). For US it is AirNow, and China CNEMC.

- Table 2: RMSE are missing in the table.

- Table 2: Why would you omit CY49R1_NOEAC in the Table, as you put the results on Fig. 1?

- Table3: (also valid for other budget table): Maybe you cloud add a minus sign in front of figure representing a loss for the considered species.

- Line 312: Figure A1

- Line 316: Weekly or seasonal?

- Figure 1: For European figures SO2_surf → SO2 and EBAS → AQer. For US figures, AirBase → AirNow.

- Line 322: Modify AirBase and CNEC.

- Line 322: For China, is it weekly or seasonal?

- Line 326: "along with associated biases". Add "not shown" or modify Fig. 2.

- Figure 2: The format of Fig. 1 is very interesting, why not use the same for Fig. 2 and have the three area and the biases on the same figure?

- Figure 2: Change EBAS to EMEP in the title.

- Line 341: "The low MB for SO2 […] is too fast". I don't understand this sentence. What rate of oxidation for which species? Please be more precise.

- Line 354: Are you referring to direct production from hot shipping exhausts?

- Line 370: Please add that you plotted surface concentrations.

- Figure 3: Why not plotting also CY49R1_NOEAC

- Line 388: I don't understand the reference to Fig. 8-10.

- Line 392: EBAS → EMEP

- Line 401: Figure A3.

- Line 401: "The corresponding […] in Table 5.". This sentence is not properly places as you don't use Table 5 for a while.

- Line 403: "Tich?" → "Tichy"

- Figure 4: EBAS → EMEP in the title

- Line 423: Introduce Table 5 here.

- Line 434: "Although maximum […] during winter." Do you have an explanation for this phenomena?

- Line 439: Figure A5 → Fig. A4

- Line 441: "This suggests […] of HNO3" →Is this a remark true at global scale?

- Figure 4 also seems to show that the emissions of NH3 are too strong in Europe when considering NH3+NH4. What can you say about this?

- Also, in the US what are the implication of the ammonium aerosol formation together with sulfate that are overestimated according to Fig. A2?

- Line 455: Table A1

- Line 456: Figure A3?

- Line 456: "Differences between […] burden in Table 4." You are comparing surface field with 3D diagnostic. This is not necessarily relevant, especially when it comes to SIA.

- Line 459: -0.26 µg/m3

- Line 479: I didn't see a proper discussion section.

- Line 484: Table 5

- Line 486: Figure A3

- Line 488: Figure A5

- Line 503: Table 6

- Line 514: Figure A6?

- Line 519: Figure A6

- Line 520: You introduce Fig. A 7 6 now, after referencing it twice.

- Line 521: I supposed you mean relative difference in percentage.

- Line 523: Please add a reference for the sentence about direct HNO3 emissions.

- Line 536: I thought you didn't have direct shipping emissions?

- Line 545: please keep using ppb all along the paragraph.

- Figure 6: EBAS → EMEP in the title.

- Line 557: "The evolution […] bottom panel" I don't see any bias plotted.

- Line 559: Fig. A7

- Line 569: Table 8 is not introduced

- Line 578: Fig. A7

- Table 7: relative difference for coarse NO3- in CY49R1 is false.

- Line 597: I would rather write 0.2-2 µg/m3

- Line 612: please add "surface" concentrations

- Line 645: "corresponding" → "comparison with observations", or something approaching.

- Line 649: "following figures" → what figures?

- Line 664: Figure 8

- Line 664: Figure 8 should be introduced in the next paragraph as you use it.

- Line 665: "To allow […] 1000 mgS/m2/year" You should remove this sentence as it doesn't reflect the figures content.

- Line 682: "by around 10 %"

- Line 694: I suppose it is Table 4.

- Line 699: temporal → spatial

- Line 702: Do you have volcanic passive degassing emissions in IFS-COMPO? Please discuss also this aspect.

- Line 705: As Fig. A1 is quite difficult to analyse due to its poor quality, we have to believe you.

- Line 744: I don't see a measuring point in Iowa.

- Line 747: 0.77 → 0.72, or a high correlation is achieved in CY48R1.

- Figure 8, 9 and 10: Labels are unreadable for Asia

- Line 715, 760 and 799: Table 10

- Line 780: Figure 13?

- Line 788: I don't understand the sentence about the fully coupled forecasting system. Please explain.

- Line 790: "values range from 50-800 mgN/m2/year with the highest value (>2000 mgN/m2/year)" If the values range from 50 to >2000, please write 50 to 2xxx.

- Line 806: "with the next operational IFS version (CY49R1)". No you didn't use CY49R, you used CY48R1 with aerosol evolution submitted for CY49R1.

- Line 816: I would not be so direct. Indeed some aspects are better represented using EQSAM4Clim, but other don't. Please be more specific about this part of the conclusion.

- Line 854: Please modify seasonal with monthly

- Figure A4: how many stations were used?

- Figure A5: The observations are missing in the legend. Also use CY4xxx as reference instead of Experiment IDs.

- Table A1: relative difference for CY49R1 bias and RMSE should be negative.

- Line 896: As for figure A1

**Author contribution:**

I don't understand VH contribution as you did not use the "complete" CY49R1 cycle, but only a modified version of the CY48R1.

**Code and Data Availability:**

I Guess there is a missing statement for the simulation data availability. Maybe there are available through MARS.

**Acknowledgements**

Please modify AirBase with AQeR and add CNEMC.

---

## Author Comment (AC1)

We thank both referees for their comments and suggestions on our original manuscript concerning the validation of the concentrations and wet deposition of secondary inorganic aerosols in the CAMS forecasting system. This has resulted in a major revision of both text and figures in order to address the suggestions and critique from the two anonymous referees. We have ensured that the referencing to all Table and Figures is now correct. We provide a detailed response to both reviews below.

Responses to referee #1:

**1- The description of the IFS-COMPO model is very succinct, and details of its main components, particularly aerosol management, would be appreciated. Schematic figures, for both model version, showing the interaction between the various modules (IFS-COMPO, IFS-AER, EQSAM4Clim, etc.) and the changes between the two versions would be relevant.**

This paper is a companion paper to that published by Remy et al. (2024) in which a comprehensive description of the updates made between CY48r1 and CY49r1 are given, and a schematic of how all the components in the model interact can be found as Figure 1 in both Remy et al. (2024) and Metzger et al. (2024). In the interests of brevity we refrain from repeating or republishing this schematic. For instance, we make no mention of IFS-AER as this is the aerosol only version of the model which is not used in this study and would add confusion if introduced.

We include the following sentences to point the reader to the other papers: vis : "*A more comprehensive description of Cy49r1 updates is provided in Rémy et al. (2024) and of the EQSAM4Clim thermodynamic module in Metzger et al. (2024), along with a schematic showing the interaction of these different models towards providing accurate air quality forecasts.*"

**2- The observational networks (section 3) are too rapidly described. More information is needed on the different stations in each dataset (number of stations, locations, data available for the period used, etc.). Figures showing the location of the various stations and a discussion of the advantages and weaknesses of each dataset would be useful in this section. Additional information, such as the uncertainties associated with these datasets or references on their reliability, would also be useful. Putting all the necessary station information in this section will also help to lighten the text and figures in the rest of the article.**

In order to improve on the ability to directly compare results across different aerosol species we now reduce the number of measurements datasets which we use in the manuscript to three main sources : EMEP, CASTNET (with AMoN for $NH_3(g)$) and EANET. We introduce a figure in the Appendix which shows the location of all the stations used for the evaluation. The lack of frequent gas-phase measurements for South-East Asia means that the statistics are now computed for the annual means of particle concentrations and wet deposition totals only for this region.

**3- All figures in the article need to be greatly improved and standardized. The quality of the figures needs to be improved, with simpler, more legible titles and more**

**complete captions. Appendix figures should be similar to those in the article (legends, curve colors, titles, etc.) and use the names of the article simulations. Many of the figure numbers in the article are also incorrect. See specific comments for details. Some of these issues should have been fixed before publication in EGUsphere.**

We have reviewed the figures upon the request of the referee. We have improved the quality of the figures showing both the surface distributions (original figures A1, A3 and A6) and the evaluations (Figs. 3, 5, 7-10), within the limits placed by the journal on the total figure size. The gas-phase comparisons against both EMEP and CASTNET have been remapped using different software. We have ensured the clarity has been improved such that, upon viewing the manuscript at high resolution, shows the distribution in fine detail.

**4- Despite the improvements brought by the new version of the model, and in particular by the implementation of EQSAM4Clim, many biases are still present. Further discussion of ways to improve these biases would be relevant. Finally, this article focuses on 3 specific regions (Europe, the United States and Asia), but some informations on other parts of the world and on the various limitations of this study would also be interesting.**

We now add some discussion on the most probable causes of the remaining biases for both the pre-cursors and the associated particles, although without specific sensitivity studies we refrain from putting emphasis on any dominant cause. In that we are unable to validate the surface results for e.g. Africa and South America with the same accuracy as the three selected regions means we refrain from expanding this to global scale. Without a meaningful evaluation of all the components involved in the formation of SIA this would detract from the main message of the paper.

**Specific comments**

*- Page 4, lines 167-168: Please add a reference.*

"For convective precipitation, the assumed precipitation fraction has been standardized to 0.05 (whereas in CY48R1, a value of 0.1 was used for chemistry scavenging and 0.05 for aerosol scavenging)."

*- Page 5, lines 214, 227: As mentioned above, indicate the number of stations, develop selection criteria, etc.*

We have introduced a figure in the Appendix showing the distribution of measurement stations from the EMEP, CASTNET and EANET measurement networks used for the evaluation of the e.g. $SO_2(g)$ surface concentrations. For the other species (i.e. SIA particle distributions and wet deposition totals) the location of the measurement stations are shown on the respective figures (Figs. 3, 6, 9-12). We have improved these figures including the use of black circles to highlight the locations.

*- Page 5, line 232: "Acid Deposition Monitoring Network in East Asia" – > "EANET" (acronym already explained line 225).*

Second definition now removed.

*- Page 6, line 253: "Nitrate#1" – > fine mode nitrate?*

Now replaced with the correct naming.

*- Page 6, line 255: A map with station locations would be useful here. Refer to section 3 if such a map is added.*

We have introduced a figure into the Appendix showing the locations of the measurement stations used for evaluating the gas-phase precursors for Secondary Inorganic Aerosols for evaluating the performance of IFS-COMPO. For the wet deposition evaluating the location of the stations are provided on each of the deposition maps shown with a colour coded circle also giving the observed levels of deposition in ug/m3 (c.f. Figure 11).

*- Page 6, line 259: "we show monthly mean regional differences for July and December 2018" – > Why did you choose these two months?*

These months are representative of summertime and wintertime performance, where seasonality exists in regional emission estimates related to e.g. regional heating ($SO_2$) and agriculture ($NH_3$) and also meteorology.

*- Page 6, line 268: Bias already cited in the literature?*

See Remy et al. (2024) for further details of such biases. We have now made a direct reference in the text.

*- Page 7, Table 2: RMSE line missing. Indicate the relative differences (in %) between the two cycles as done in other Tables. Remove "diagnostic" and add "SO2" to harmonize with other tables.*

This table has now been removed to align the comparisons across species and the remaining tables subsequently renumbered.

*- Page 7, Table 3: Harmonize the legend of all tables in the article. "Percentage difference changes are calculated as …" or "with the associated relative differences provided in parenthese …": Choose one sentence and use it for all the Tables.*

We now ensure that the table legends are consistent with each other for the three budget tables.

*- Page 7, line 312: "Figure 1A" – > Figure A1*

This figure has now been removed from the paper.

*- Page 7, line 312: "the region with the highest surface $SO_2$ concentrations is the northeastern U.S." – > Please rephrase the sentence, not consistent with the figure.*

We have now moved this figure into the main text (becoming the new Fig 1) and we now rephrase this sentence to "Figure 1 shows the Eastern U.S. exhibits the highest surface $SO_2$"

*- Page 7, line 313: "There is little seasonality in the weekly observational composites" – > consistent with literature?*

We are referring to the observational weekly means derived from the CASTNET stations in the U.S. for this specific year therefore feel we do not need to refer to any previous study. We have significantly expanded on the description of these sites as requested by the referee.

*- Page 7, lines 318-319: "SO$_2$ emissions in the global inventory are significantly overestimated" – > What about the literature? Has this problem ever been reported or documented?*

We refer to the specific SO$_2$ emission inventory as developed in the CAMS consortium which are constantly being modified in response to model performance. The comparisons shown in this paper for both Europe and the US show consistent positive biases of > 100%, where [SO$_2$(g)] is critically dependent on emission fluxes applied (no chemical production term is included in the chemical scheme for SO$_2$). The methodology used for deriving the CAMS-GLOB-ANTH inventory is given in Soulie et al., (2024), Earth Syst. Sci. Data, 16, 2261–2279, https://doi.org/10.5194/essd-16-2261-2024, 2024.

*- Page 8, Figure 1: Please give each figure a simpler, clearer title (ex: SO2 - Europe, SO2 - US, SO2 - China). Also use a vertical title for the two figure lines, to the left of the first figure of each line. Put a title on the y axis of the figures. Complete the legend with all the necessary information, but do not include this information in the figure titles. Also describe the second line of figures in the legend. See also to put only one curve legend for each set of figures to lighten the figures. In the caption, please refer to section 3 for the description of the different observation datasets. The figure is not very sharp, please use a PDF or EPS file.*

This figure has now been removed to align the comparisons across species and the remaining tables subsequently renumbered. It has been replaced by evaluation using measurements from the EMEP and CASTNET networks including all stations.

*- Page 8, Figure 2: Please give a simpler and clearer title (ex: SO4 - Europe). Put a title on the y axis. In the caption, please refer to section 3 for the description of the different stations used.*

We have remade this figure and removed the previous Fig A2. Now European and US comparisons of SO$_2$ and SO$_4^{2-}$ are displayed side by side. We now change the figure legend to : "A comparison of weekly mean SO$_4^{2-}$ for Europe (top panel; µg/m$^3$) and the US (bottom panel) simulated in Cy48r1, Cy49r1_NOE4C and Cy49r1 as compared against measurement composites from stations representative of a rural scenario taken from EMEP and CASTNET observational networks, respectively, for 2018. The sampling frequency of the data for South-East Asia does not allow a corresponding weekly plot for this region.". We also simplify the figures titles as suggested.

*- Page 10, Figure 3: Review the titles of the various figures (unclear and illegible). Put a title for each column (ex: CY48R1 and CY49R1) and each row (ex: SO4 - Europe, SO4 - US, SO4 - Asia). Put only one colorbar per line to make it more legible and add*

*its own unit. In the caption, please refer to section 3 for the description of the different observation datasets.*

We have now completely restyled this Figure and changed the titles and provided a higher resolution version, where the location of the observational stations can be seen more clearly by adopting a black surrounding circle. We provide only one colour bar now at the bottom of each column.

*- Page 12, Figure 4: Please give a simpler and clearer title for each figure (ex: NH3 – Europe and NH4 -Europe). Put a title on the y axis. In the caption, please refer to section 3 for the description of the different stations used. Harmonize all figure captions of the article. "The corresponding biases are shown in the bottom panel." is not appropriate here.*

We have now moved Fig. 4A into the main text such that the comparison between the performance in Europe and the US can be made more easily. We have also homogenized the figures legends as for the corresponding plots for $SO_4^{2-}$ .

*- Page 15, Figure 5: Same as Figure 3.*

We have restyled this figure with similar changes as those described for Figure 3.

*- Page 16, lines 513-519: You refer twice to Figure A7, but it's Figure 6, isn't it?*

Thanks for finding the typo which is now Corrected.

*- Page 16, line 520: I think this is Figure A6 and not Figure A7.*

This figure has now been moved into the main text.

*- Page 17, Figure 6: Same as Figure 4.*

We have now moved Fig 7A into the main text such that the comparison between the performance in Europe and the US can be made more easily. We have also homogenized the figures legends between $HNO_3$ and $NO_3^-$ for both regions.

*- Page 19, Figure 7: Same as Figure 3.*

In line with Figure 3 we have improved this figure addressing the issues raised on figure legends and clarity.

*- Page 20, Table 8: Remove % from this table to harmonize with other tables. Please correct ug – > µg (also for Table 4 and 6).*

We have now homogenized the format across all tables.

*- Page 20, lines 621-630: Not the same font.*

This seems to have been introduced during the typesetting process as in our submitted version everything is formatted as Times New Roman as requested in the submission.

*- Page 22, line 664: Here too, I think it's Figure 8 and not Figure 11.*

Thanks for finding the typo which is now Corrected.

*- Page 23, Figure 8: Same as Figure 3. - Page 25, Figure 9: Same as Figure 3. - Page 27, Figure 10: Same as Figure 3.*

These three figures have been improved in line with the comments made related to clarity and the figure legends.

*- Page 26, lines 780-781: Figure 13 – > Figure 10. Please check that the text and figure numbers are consistent throughout the article.*

We have made major changes to the manuscript where we have moved some of the figures which were in the Appendix into the main body of the text on the request of the reviewers. Therefore all figures have been renumbered throughout the article, with care taken not to reference incorrect figures.

*- Page 29, Figure A1: Make a more legible figure. For example, use clear headings for each column (ex: NH3 - CY48R1, Diff A, Diff B, Diff C). Also include Europe, US and Asia, as well as July and December for the corresponding vertical rows on the left of the figure. Delete sub-figure headings for greater legibility.*

We have restyled and improved the presentation of this figure following the suggestions of the referee and moved the figure into the main text.

*- Page 30, Figure A2: Harmonize the titles (ex: NH3 – US and NH4 – US), axes, axis titles and legend with the figures in the article. Use the same curve colors as the other figures (keep the same color for observations and different versions throughout the article).*

We have now moved this figure into the main text and replotted to homogenize the figure legends and colour scheme.

*- Page 31, Figure A3: Same as Figure A1.*

See response related to original Figure A1 which has now been moved into the main text.

*- Page 32, Figure A4: Same as Figure A2.- Page 33, Figure A5: Harmonize the figure with the other figures in the article. Put the names of the versions used in the article and add observations in the caption. Use station name as sub-figure title.*

This figure has now moved into the main text and replotted to address the issues listed by the referee.

*- Page 33, Table A1: Add NH3 to the table to harmonize with the other tables. In the caption, please refer to section 3 for the description of the different stations used here.*

We have now changed the title of this table to be more compatible with the other tables presented in the manuscript.

*- Page 34, Figure A6: Same as Figure A1.; Page 35, Figure A7: Same as Figure A2.*

Both figures have now been improved and moved into the main body of the text.

---

## Author Comment (AC2)

We thank anonymous referee #2 for the comments and suggestions on our original manuscript concerning the validation of the concentrations and wet deposition of secondary inorganic aerosols in the CAMS forecasting system. This has resulted in a major revision of both text and figures in order to address the suggestions and critique from the two anonymous referees. We have ensured that the referencing to all Table and Figures is now correct. We provide a detailed response to both reviews below.

Responses to Referee #2

**Specific comments:**

*Concerning the figures, their quality is overall poor, especially in the Appendix. I would recommand to do an effort to increase their quality and readibility. Also the partitioning between the paper and appendix should be questioned. Figure A1 for example is used several times and is worth to appear in the main part of the paper.*

In response to the referee's criticism we have now worked on the quality of the figures throughout the manuscript and enhanced the readability. The three plots concerning regional changes in the gaseous precursors simulated in IFS-COMPO have now been moved into the main text and restructured the manuscript accordingly. We also address the issues raised by referee #1 regarding figure legends.

*As you are not using the "real" CY49R1, but modified version of CY48R1, I would recommend to delete all reference to CY49R1 expect from the fact that the presented modifications are meant to enter in the latter.*

This paper is a companion paper to one already published in GMD (Remy et al, 2024) and we use the same nomenclature as in this publication to avoid confusion. All the updates described and evaluated here are active in Cy49r1 therefore the results presented pertain to this version of the model. Given that Cy49r1 is operational we now change to text to reflect this e.g. by removing the word 'proposed'.

*A section is missing to explain how are computed the statistics. Are all the data gathered in one vector and the tables represent spatio-temporal statistics*

The statistics are calculated with the in-house statistics package available at ECMWf called which are used for evaluation throughout the CAMS initiative. We now add the following sentence : "All statistical metrics represent spatio-temporal averages unless otherwise noted, combining all station-time pairings into a single evaluation vector per region and species"

*- Line 126: "which will be operational in November 2024": Please update this sentence as the date is now expired.*

Thanks for the comment. Now replaced with " … to be operational in 2025).

*- Table 1: The Experiment IDs are not used in the text. Please remove them or add a sentence to tell that they can be used to retrieve the data on the MARS storage system.*

We retain the experimental ID's such that readers have the information to access the original simulations at any time in the future. We add the following sentence in the Table Legend : "The experiment ID's can be used to retrieve the original data from the MARS archiving system hosted at ECMWF."

*- Line 205: Please add a reference for the direct production from hot shipping exhausts.*

We now include a peer reviewed reference related to this statement.

***- Line 211: AirBase is no longer used since mid 2000's. The new system is called AQeR (AirQuality e-Reporing)***

In the new manuscript we remove the use of AirBase data.

***- Line 214: How did you select "rural background stations"?***

In the new analysis we now use all available stations rather than limiting the analysis to a subset.

***- Line 271: Figure A1. Please introduce the figure before referencing it. Also Fig. A1 is used a lot. Maybe it would be interesting to move it to the main part of the paper.***

We now move the original Figures A1, A3 and A6 included in the original supplement into the main text, where change the text

***- Line 272: Please introduce Table 3 before using it.***

We have amended the text to : "The associated annual mean statistics are provided in Table 2. To assess the global integrated impact on $SO_4^=$ formation, the associated global budget terms are provided in Table 3 in Tg S/year. "

***- Line 272: "For CY49R1 [...] by approximately.1-0.µg/m3" → I assume you are talking about Fig. 2 results. Please introduce it and announce it properly or move the sentence in the SO4--paragraph.***

We have now re-written this paragraph expanding on the results and to introduce the Figure in the correct place.

***- Line 274: Figure A1.***

This figure has now been moved into the main text such that this appendix Figure labelling is now obsolete.

***- Line 279: You introduce Table 3, but you used it only in the next paragraph.***

We have now introduced and defined Table 3 immediately after the introduction/definition of Table 2 to allow it to be used in later discussion.

***- Line 306: "Statistics relate to seasonal means" → Please be more precise about this sentence. Maybe in the section 3 when introducing observational data.***

We have now provided details regarding the statistics used.

***- Table 2: For Europe, according to the section 3, data must be AQeR (replacing AirBase). For US it is AirNow, and China CNEMC.***

Thank you for catching this error. We have now re-labelled the columns.

***- Table 2: RMSE are missing in the table.***

Table 2 has now been removed.

***- Table 2: Why would you omit CY49R1_NOEAC in the Table, as you put the results on Fig. 1?***

Table 2 has now been removed.

*- Table3: (also valid for other budget table): Maybe you cloud add a minus sign in front of figure representing a loss for the considered species.*

Thank you for the suggestion. However we do not provide the differences in terms of Tg species and cannot use colours (red for increase, blue for decrease) due to the journal protocol.

*- Line 312: Figure A1 ; Line 316: Weekly or seasonal?; - Figure 1: For European figures SO2_surf → SO2 and EBAS → AQer. For US figures, AirBase →AirNow. ; - Line 322: For China, is it weekly or seasonal? ; - Line 326: "along with associated biases". Add "not shown" or modify Fig. 2., - Figure 2: The format of Fig. 1 is very interesting, why not use the same for Fig. 2 and have the three area and the biases on the same figure?; Figure 2: Change EBAS to EMEP in the title.*

This figure has now been removed

*- Line 341: "The low MB for SO2 [...] is too fast". I don't understand this sentence. What rate of oxidation for which species? Please be more precise.*

Comparing the associated negative bias in SO2 and the high positive bias of SO4= for summertime (with higher OH) suggests that the lifetime (chemical oxidation rate) is too low, where less SO4= would equal more SO2 improving both comparisons. We have now rewritten the sentence to ; '*Considering the corresponding low summertime MB for SO$_2$ in Figure 2 shows that the rate of oxidation in IFS-COMPO is too fast, where a slower oxidation rate of SO$_2$ by OH and/or aqueous phase processing would be required to improve the performance of IFS-COMPO.*"

*- Line 354: Are you referring to direct production from hot shipping exhausts?*

We now extend the sentence with "....here related to missing shipping emissions of SO$_2$, which quickly converts to SO$_4^2$ in the plume (Celik, et al., 2020)."

*- Line 370: Please add that you plotted surface concentrations.*

The figure legend has now been amended.

*- Figure 3: Why not plotting also CY49R1_NOEAC*

The clarity and quality of the original figure has been significantly improved., as requested by anonymous referee #1. We feel that by including additional panels for the CY49R1_NOE4C simulation would introduce potential confusion due to the size of the measurement stations being so small

*- Line 388: I don't understand the reference to Fig. 8-10.*

Thank you for finding this error now changed to the new Fig.4.

*- Line 392: EBAS → EMEP*

We now correct this typo.

*- Line 401: Figure A3.*

We have now moved this figure into the main text.

*- Line 401: "The corresponding [...] in Table 5.". This sentence is not properly places as you don't use Table 5 for a while.*

We have now moved this sentence as requested by the referee.

*- Line 403: "Tich?" → "Tichy"*

This seems to have been related to a typesetting issue rather than a typo from our side with the representation of y with a hyphen missing.

*- Figure 4: EBAS → EMEP in the title*

We now move the name of the dataset into the figure legend.

*- Line 423: Introduce Table 5 here.*

We now introduce Table 5 (new Table 4).

*- Line 434: "Although maximum [...] during winter." Do you have an explanation for this phenomena?*

This is due to the low volatility of $NO_3NH_4$ during the colder temperatures during wintertime, resulting in a longer tropospheric lifetime than in the summertime. The impact of EQSAM4Clim only becomes apparent after April due to the $HNO_3$ limiting particle formation as described in Tang et al, ACP, Atmos. Chem. Phys., 21, 875–914, https://doi.org/10.5194/acp-21-875-2021 2021. We now include extra discussion on this issue.

*- Line 439: Figure A5 → Fig. A4*

This figure has now been moved into the main section of the text and the numbering of figures updated accordingly.

*- Line 441: "This suggests [...] of HNO3" →Is this a remark true at global scale?*

This is a physical limitation to $NO_3NH_4$ formation and the sensitivity to $HNO_3(g)$ has been determined by comparing multiple species associated with SIA formation (Tang et al, ACP, 2021). For the tropics the high temperatures mean that the lifetime of $NO_3NH_4$ is very short and somewhat limits maximal concentrations.

*- Figure 4 also seems to show that the emissions of NH3 are too strong in Europe when considering NH3+NH4. What can you say about this? Also, in the US what are the implication of the ammonium aerosol formation together with sulfate that are overestimated according to Fig. A2?*

Independent studies have shown that the emission estimates for NH3(g) are realistic where we include the following sentence : "The CAMS_GLOB_ANT v5.3 (Soulie et al, 2024) emission inventory has recently been validated for $NH_3$ against top-down estimates providing confidence in the quality of the estimates for Europe (Ding et al., 2024). ". The fraction of SO4= bound the NH4+ is accounted for in EQSAM4Clim therefore the excess in SO4= shown in the new Fig. 2 is the form of non-bound SO4=.

*- Line 455: Table A1 - Line 456: Figure A3?*

Table A1 has now been removed and we have also moved the Figure from the Appendix into the main text.

*- Line 456: "Differences between [...] burden in Table 4." You are comparing surface field with 3D diagnostic. This is not necessarily relevant, especially when it comes to SIA.*

The vertical profile of $NH_3$ means that most of the particle formation occurs in the lower troposphere for $NH_4^+$ therefore we can use these changes in a qualitative sense.

*- Line 459:-0.26 µg/m3*

Now corrected in the text.

*- Line 479: I didn't see a proper discussion section.*

We now modify this text to remove the suggestion of further discussion.

*- Line 484: Table 5*

We now correct this typo.

*- Line 486: Figure A3 - Line 488: Figure A5*

Both figures have now been moved into the main text and the numbering changed in the text.

*- Line 503: Table 6*

We have now changed the reference to the correct table in the figure legend.

*- Line 514: Figure A6? - Line 519: Figure A6 - Line 520: You introduce Fig. A6 now, after referencing it twice.*

This figure has now been moved into the main text and the Figure renumbered as Figure 8.

*- Line 521: I supposed you mean relative difference in percentage.*

We have changed the text to relative differences.

*- Line 523: Please add a reference for the sentence about direct HNO3 emissions.*

We now add : Vinken, G. C. M., Boersma, K. F., Jacob, D. J., & Meijer, E. W.: Accounting for non-linear chemistry of ship plumes in the GEOS-Chem global chemistry transport model. Atmospheric Chemistry and Physics, 11(22), 11707-11722. https://doi.org/10.5194/acp-11-11707-2011, 2011.

*- Line 536: I thought you didn't have direct shipping emissions?*

We have emission of CO, $SO_2$, $NO_2$ and VOC as integrated into the anthropogenic global emission inventories, but we do not have direct emission of either $SO_4^=$ or $HNO_3$ as described in the text. We now clarify this in the text.

*- Line 545: please keep using ppb all along the paragraph.*

We have now changed ppt to ppb throughout.

*- Figure 6: EBAS → EMEP in the title.*

We now move the name of the dataset into the figure legend.

*- Line 557: "The evolution [...] bottom panel" I don't see any bias plotted.*

We now improve these figures by including corresponding bias plots for these type of comparisons in our amended manuscript.

*- Line 559: Fig. A7*

This figure has now been moved into the main section of the text and the numbering of figures updated accordingly.

*- Line 569: Table 8 is not introduced*

Table 8 should have been Table 7 (new Table 6). This typo is now corrected.

*- Line 578: Fig. A7*

This figure has now been moved into the main section of the text and the numbering of figures updated accordingly.

*- Table 7: relative difference for coarse NO3- in CY49R1 is false.*

We apologize for the error and have now corrected the change in the respective budget term.

*- Line 597: I would rather write 0.2-2µg/m3*

Now changed in the text

*- Line 612: please add "surface" concentrations*

Now added in the text.

*- Line 645: "corresponding" → "comparison with observations", or something approaching.*

This sentence has now been changed.

*- Line 649: "following figures" → what figures?*

This pertains to the regional comparisons of the wet deposition totals which follow the tables. We now specify which figures.

*- Line 664: Figure 8 should be introduced in the next paragraph as you use it.*

We have now renumbered the figures and introduce the comparison of total S wet deposition in the correct place.

*- Line 665: "To allow [...] 1000 mgS/m2/year" You should remove this sentence as it doesn't reflect the figures content.*

We have now changed the text for 1000 mgS/m2/year to 500 mgS/m2/year so as not to be misleading.

*- Line 682: "by around 10%"*

Now changed in the text.

*- Line 694: I suppose it is Table 4.*

The tables have been renumbered and we ensure that we refer to the correct table in the text.

*- Line 699: temporal → spatial*

Now changed in the text.

*- Line 702: Do you have volcanic passive degassing emissions in IFS-COMPO? Please discuss also this aspect.*

Volcanic outgassing is included in the simulations as detailed here https://atmosphere.copernicus.eu/eruptive-emissions, which does contribute to the background $SO_2$ concentrations in specific (typically remote) locations. Given the lack of measurement stations in Italy we do not think that this impacts our results in any way for the chosen simulation year. We now mention this in the description of the simulations.

*- Line 705: As Fig. A1 is quite difficult to analyse due to its poor quality, we have to believe you.*

This figure has now been provided in more high resolution as .eps format and amended in accordance with the requests of referee #1.

*- Line 744: I don't see a measuring point in Iowa.*

There is a measurement station situated in the bottom left corner of the state but we agree that this is difficult to see on the figure. We have improved the visibility of the location of the measurement stations on the new version of the figure.

*- Line 747: 0.77 → 0.72, or a high correlation is achieved in CY48R1.*

We have now corrected the value of the correlation coefficient for CY49r1 in the text.

*- Figure 8, 9 and 10: Labels are unreadable for Asia*

These figures have now been replotted and provided at a higher resolution, where modifications have been made according to the suggestions of referee #1. The location of the stations used for validation have also been made clearer.

*- Line 715, 760 and 799: Table 10*

All tables have been renumbered and we ensure that we reference the correct table in the new figure legends.

*- Line 780: Figure 13?*

We have now renumbered most of the figures due to shifting those in the Appendix into the main body of the text.

*- Line 788: I don't understand the sentence about the fully coupled forecasting system. Please explain.*

This refers to the lack of data assimilation in the IFS-COMPO simulations used in the manuscript. We now provide more clarity in the text.

*- Line 790: "values range from 50-800 mgN/m2/year with the highest value (>2000 mgN/m2/year)". If the values range from 50 to >2000, please write 50 to 2xxx.*

We have now modified the text.

*- Line 806: "with the next operational IFS version (CY49R1)". No you didn't use CY49R, you used CY48R1 with aerosol evolution submitted for CY49R1.*

We refer the referee to our answer given above regarding the nomenclature of model versions.

*- Line 816: I would not be so direct. Indeed some aspects are better represented using EQSAM4Clim, but other don't. Please be more specific about this part of the conclusion.*

We have now removed this sentence and present the conclusions in a less generic manner.

*- Line 854: Please modify seasonal with monthly*

We have now modified the figure legend.

*- Figure A4: how many stations were used?*

We have now moved this figure into the main text such that direct comparisons can be made against the EMEP results. The figure has been replotted, formatting homogenised and clarity improved.

*- Figure A5: The observations are missing in the legend. Also use CY4xxx as reference instead of Experiment IDs.*

In the updated manuscript we have reflected on the inclusion of these comparisons and have decided to remove it as we only presented this for NH3 and, although they are included in the regional statistics for the US, feel they do not contribute to the main flow of the manuscript.

*- Table A1: relative difference for CY49R1 bias and RMSE should be negative, - Line 896: As for figure A1*

This table has been removed and the figure moved into the main text.

*Author contribution: I don't understand VH contribution as you did not use the "complete" CY49R1 cycle, but only a modified version of the CY48R1.*

VH was a PI of the CAMS35_2 project which funded this work and was involved in updating the infrastructure of the code as used here. This is stated clearly in the author contribution section.

*Please modify AirBase with AQeR and add CNEMC.*

We have removed this analysis from the manuscript therefore we no longer use the datasets.

---

## Referee Report (RR1)

**Anonymous referee comments on:**

**An evaluation of the regional distribution and wet deposition of secondary inorganic aerosols and their gaseous precursors in IFS-COMPO Cycle 49R1**

**by J.E. Williams et al.**

This comment is referee number 2 responses on authors responses after the first evaluation round.

**Comments on Referee #2 responses**

Authors didn't answer to referee #2 general comment. Could you please do so and especially give details about some important points such as:

- The changes of the wet deposition scheme and analysis. You indicate that the companion paper already describe the changes made to the code. Nevertheless I consider this in not precise enough.

- The naming of the code version (cy48 vs cy49). I understand your point, even if you do not treat the possible interactions between the composition compartment and the other changes in the IFS model between CY48 and Cy49. Also, I would suggest to at least modulate the title to suggest the ambiguity. Also I would recommand to be more precise for future publications. Concerning the use of Cy49r1 in the present publication I will leave it to the editor to decide.

- Did you check conservativity in the budget tables?

Simulation Cy49r1_NOE4C usage is quite under, it might be useful to exploit it more to be able to make the distinction between different impacts, or to remove it completely.

Is there a signification in the colours of the modified manuscript?

Also there are still figure and table references that are not correct...

**Specific comments**

Authors wrote:

*Line 272: Please introduce Table 3 before using it. We have amended the text to : "The associated annual mean statistics are provided in Table 2. To assess the global integrated impact on SO4= formation, the associated global budget terms are provided in Table 3 in Tg S/year."*

I can't find this sentences in the updated text.

*Table3: (also valid for other budget table): Maybe you cloud add a minus sign in front of figure representing a loss for the considered species.*

*Thank you for the suggestion. However we do not provide the differences in terms of Tg species and cannot use colours (red for increase, blue for decrease) due to the journal protocol.*

I asked for a minus sign, not colours.

*Figure A4: how many stations were used?*

Figure A4 is now a part of figure 5, but you didn't answer the question.

**Additional or new specific comments**

- Line 349: "No filtering has been applied to these measurements" Does it mean you applied filtering to the other datasets?

- Line 443: "The global budget" You forgot to introduce Table 2.

- Line 793: "Europe, US and SE Asia"

- Lines 995+ and 1080+ are developing the same point.

- Line 1095: Maybe also say that AMON data are used for top right panel.

- Line 1107: significant decreases → Significant decrease

- Line 1121: "Unfortunately the lack […] can be shown." maybe you should add a paragraph on how selected stations or data, because on the EBAS website when selecting 2018 and ammonium data from the EMEP "Framework" there is a station in South France at Pic du Midi which has data for the aerosol matrix and a few stations including Cyprus and Italy in the pm10 matrix as shown in figure below.

[Figure]

[Figure]

[Figure]

[Figure]

[Figure]

- Line 1334: "Table 8" not sure which table you are referencing here.

- Line 1346: IFS-COMPO: which version?

- Line 1457: "Comparison surface comparison"

- Line 1478: "panel"

- Line 1230: Figure 5 has been referred to instead of figure 4.

- Line 1260+: "This can lead to […] dynamically limited." This sentence suggests that Cy48r1 uses EQSACM4Clim while it is not the case.

- Line 1268: "with a strong seasonal cycle in maximal values peaking in July" It is quite hazardous to assert this based on two monthly plots.

- Line 1597: Table 2?

- Line 1664: I guess you mean figure 4 or 5.

- Line 1665: Do you mean Benelux instead of Balkans?

- Line 1806 and 2124: CNEMC data are not used any more.

**Comments on Referee #1 responses**

The numbers corresponds to the number of Referee#1 comments.

2- You said you added a figure in the Appendix showing the location of all the station used for evaluation. This would have been indeed a good idea, nevertheless you only showed stations used for SO2 surface concentrations. If the stations are the same for all gaseous measurements please change the legend otherwise please change the figure. You could use a colour code to make the distinction between stations.

4. You indeed discuss the reason for biases being still present, but you don't discuss on possible way to improve these biases.

---

## Editor Decision (ED1)

Dear Author,

I have the following comments at this stage of the review:

- you will find the second round of reviews from both reviewers on the GMD webpage. Please address their new comments carefully in your response

- reviewer 1 recommends accepting the manuscript as is. However, based on the comments provided, which I have repeated just below, I believe the manuscript should be accepted pending minor revisions.

  " I just have a minor technical corrections: in Figures 3, 6, 9, 10, 11, 12 the visibility of the color bar should be improved and the captions corrected (write the charge as a superscript). As for figures 2, 5 and 8, the legends can be enlarged to make them more visible."

- please note that I agree with the following reviewer 2 comment:

  "As you are not using the "real" CY49R1, but modified version of CY48R1, I would recommend to delete all reference to CY49R1 expect from the fact that the presented modifications are meant to enter in the latter."

  And I find your response to be insufficiently convincing. Several changes to the text should be made in your article to clarify things:

  - regarding the title of your article: Even though S. Remy used the terms 'ECMWF IFS-COMPO 49R1' in the title of his 2024 GMD publication, the title of your article should be modified to improve clarity. I have the following suggestion, though you may have another one:

    "An evaluation of the regional distribution and wet deposition of secondary inorganic aerosols and their gaseous precursors in IFS-COMPO preparatory to cycle 49R1"

  - the following renaming of the simulations should be applied throughout your article, as suggested below:

    Cy48r1 : unchanged

    Cy49r1: Cy49r1compo

    Cy49r1_NOE4C: Cy49r1compo_NOE4C

  - in the abstract: change " The application of the EQSAM4Clim simplified thermodynamic module in IFS-COMPO cycle 49R1" into " The implementation of the EQSAM4Clim simplified thermodynamic module in IFS-COMPO, for use in cycle 49R1"

  - In Section 2, the introductory paragraph has been revised in the updated version of your article, partly reflecting the fact that cycle 49r1 is now operational. However, the text still contains some unclear sections. I recommend rewriting it, particularly to avoid conflating the description of the IFS-COMPO system with that of the specific simulations you performed. Also, Rémy et al., 2022 and Williams et al., 2022 analyse Cy47r1. I therefore have the following proposal presented below:

    "The IFS-COMPO global composition model (previously known as C-IFS) is used for operational air quality analyses and forecasts as part of CAMS. The modelling and data assimilation framework is regularly updated. During 2023, IFS-COMPO was based on the Cy48r1 version of IFS (see https://www.ecmwf.int/en/elibrary/81374-ifs-documentation-cy48r1-part-viii-atmospheric-composition; last access: 20 February 2024). Since the end of 2024, IFS-COMPO has moved to Cy49r1. IFS-COMPO Cy49r1 has been shown to reduce the biases previously identified in key products such as $O_3$ and $NO_2$ (Huijnen et al., 2016; Huijnen et al., 2019; Williams et al., 2022; https://atmosphere.copernicus.eu/eqa-reports-global-services; last access: 17 February 2025). Several updates were introduced in Cy49r1 to improve the aerosol component, the wet deposition scheme, and the representation of pH in clouds and aerosols. These include the application of the EQSAM4Clim approach and other cloud scavenging-related developments (Metzger et al., 2016; Metzger et al., 2024; Rémy et al., 2024).

For brevity, we provide only a brief description below of the updates in Cy49r1 relevant to this study. Further details are available in Rémy et al. (2024), which also includes a schematic representation of the model component interactions, and in Metzger et al. (2024) for the EQSAM4Clim thermodynamic module. The full documentation of IFS Cy49r1 is available at https://www.ecmwf.int/en/publications/ifs-documentation, lass access 23 July 2025."

– line 150: change "Updates in IFS-COMPO Cy49r1" into "IFS-COMPO Cy49r1 updates of interest for this study"

- other remarks:

  – is it correct that you present results for 2018, while Remy et al. 2024 present results for 2019?

  – you indicate that in your simulations meteorology is "initialised every 24 hours based on ERA5 reanalysis data". Is it the same protocol in Remy et al. 2024?

  – what is the reference for the climatology of DMS emissions you used?

  – please detail how biomass burning emissions and SO2 emissions are applied

  – two additional remarks/questions about your simulations: please indicate whether meteorology can differ across simulations, depending on whether aerosol–radiation interactions are activated or not. And what about the gazeous chemistry scheme, is it identical across all your simulations?

  – line 18 : please write in full "IFS"

  – line 29: I propose to change: "There is also a shift in the size of particles towards the fine mode nitrate away from the coarse mode. " into "There is also a shift in particle size distribution, with nitrate moving from the coarse mode toward the fine mode."

  – line 52: change "The SIA occurs" into "Secondary inorganic aerosols (SIA) are found throughout the troposphere."

  – line 53: change "and the concentrations" into "and concentrations"

  – line 54: no capital letter to sulphur dioxide, ammonia, nitric acid, particulate matter .
    And add a "," before ammnonia

  – line 55: please refer to the exact definitions of PMP1.0, etc... particles smaller than ...

  – line 62: in the form "OF" NO...

  – line 63: no capital letter to sulphur and nitrogen

  – line 65: change " SIA being from NH4NO3" in " SIA consists of NH4NO3"

  – line 129 : change "Model description of IFS-COMPO versions" into "General information on the most recent IFS-COMPO versions"

  – line 66: please reformulate 'increased meteorological instability'

  – line 67: do you mean: 'reducing the potential for long-range transport out of the source regions"?

  – line 68: most SIA: consider it as singular

  – line 68: "At high RH values": please specify a threshold

  – line 70: change "the optical properties" into "its optical properties"

  – line 71: change ' and interactions' into 'and its interactions"

  – line 73: change "-1" into "1"

  – line 83: change 'whose cumulative oxidation rate is dependent on the prescribed pH in solution' into 'whose overall oxidation rate varies according to the specified pH of the solution."

  – line 85: "for the determination of trends": please be more precise

  – line 89: change 'solution pH' into 'the pH of the solution'

- line 93: please reformulate to clarify the sentence: 'The more buffering of solution by NH3, the faster the conversion rate as dictated by the reaction of HSO3- being less than that for SO3 = '
- line 95: please change 'One dominant loss term for SIA is wet deposition in precipitation to the surface ' into 'SIA is primarily removed from the atmosphere through wet deposition, as it is scavenged by precipitation and transported to the surface'
- line 101: please precise what you mean by 'the distribution of the cloud liquid water content' and by the 'cloud Surface Area Density, SAD'
- line 104: please move the reference Peuch et al 2022 after "CAMS". Similarly, William et al 2022 does not refer to IFS-COMPO. Please reformulate.
- line 110: change "is the reduction of biases and increase in correlation for aerosol products" into 'was the reduction of biases and increase in correlation of aerosol products with observations'
- line 111: please reformulate : 'In that, acidic deposition and N-loading can also be output from the model means such that improving the deposition term via an improved distribution in PM will foster the development of this IFS-COMPO future product. '
- line 114: please reformulate the first sentence of the paragraph, with shorter sentences. Please specify now that you assess the performance of both IFS-COMPO Cy48r1 and an evolution of this version with updates that will be part of Cy49r1.
- line 118: please change ' This work is complementary to the recent evaluation of the performance of IFS-COMPO Cy48r1 and Cy49r1 and of the impact of using EQSAM4Clim with respect to regional PM2.5 distributions and Aerosol Optical Depth presented in Rémy et al (2024). into 'This work is both parallel and complementary to the recent assessment of global and regional PM2.5 and Aerosol Optical Depth (AOD) distributions presented in Rémy et al (2024).
- line 121: please clarify what you mean by 'upgrades to EQSAM4Clim'
- line 124 "we provide details of the changes": which changes are you referring to?
- line 125: please rewrite: 'in Sect. 5, we compare annual mean wet deposition fluxes over Europe, the U.S., and Southeast Asia using model outputs and observations...'
- line 154: no capital letters for "desert dust" and "sea-salt"
- line 156: change "has also" to "have also", "which improves" into "which improve"
- ...

- I will stop here with my suggestions for textual revisions. I note that in general the readability of the article could be greatly improved by enhancing the fluency of the English. Numerous tools are available for this purpose... Thank you for making all efforts in that direction. For some parts of the text, please rely on the formulations used by Remy, GMD 2024, which appear very clear.

I do hope that the next version of the manuscript will be easy to read, yet include all the necessary details that contribute to the quality of a scientific article. Moreover, it will reflect the second round of comments from the reviewers, along with my own remarks. I will then reread the entire article for any final suggestions.

Regards, Martine Michou

---

## Author Response (AR2)

Response to Topical Editor

We thank the topical editor for conducting a review of the revised manuscript and making some clear suggestions regarding the minor revision that must be made before publication. Here we respond to these points below;

*Regarding the title of your article: Even though S. Remy used the terms 'ECMWF IFS-COMPO 49R1' in the title of his 2024 GMD publication, the title of your article should be modified to improve clarity. I have the following suggestion, though you may have another one:*
*"An evaluation of the regional distribution and wet deposition of secondary inorganic aerosols and their gaseous precursors in IFS-COMPO preparatory to cycle 49R1"*

We have now changed the title to that given by the Topical Editor. However, we do note that GMD has a strict policy with respect to model versions and this introduces some inconsistency between studies performed with the same model version.

*The following renaming of the simulations should be applied throughout your article, as suggested below: Cy48r1 : unchanged, Cy49r1: Cy49r1compo and Cy49r1 NOE4C: Cy49r1compo NOE4C*

We have retained Cy48r1 and changed the nomenclature of the other two simulations to pre-CY49r1 and pre-CY49r1_NOE4C, which we find matches the change made to the title of the manuscript. This allows the manuscript to be found when searching whilst indicating that this is not the final version of Cy49r1, but a version of Cy49r1 in a preparatory phase.

*in the abstract: change " The application of the EQSAM4Clim simplified thermodynamic module in IFS-COMPO cycle 49R1" into " The implementation of the EQSAM4Clim simplified thermodynamic module in IFS-COMPO, for use in cycle 49R1"*

The abstract has now been changed as requested.

*In Section 2, the introductory paragraph has been revised in the updated version of your article, partly reflecting the fact that cycle 49r1 is now operational. However, the text still contains some unclear sections. I recommend rewriting it, particularly to avoid conflating the description of the IFS-COMPO system with that of the specific simulations you per- formed. Also, R´emy et al., 2022 and Williams et al., 2022 analyse Cy47r1.*

We now adopt the amended text provided by the topical Editor and remove the references.

Specific comments:

*Is it correct that you present results for 2018, while Remy et al. 2024 present results for 2019? you indicate that in your simulations meteorology is "initialised every 24 hours based on ERA5 reanalysis data". Is it the same protocol in Remy et al. 2024?*

Yes this is correct we now include the sentences in Sect. 2.2: "These three simulations use a configuration and emissions as those used for simulations presented in Remy et al., (2024) for

evaluating PM. We select the year 2018 to provide further evaluation which is complimentary to the results presented for 2019 in Remy et al., 2024).

*you indicate that in your simulations meteorology is "initialised every 24 hours based on ERA5 reanalysis data". Is it the same protocol in Remy et al. 2024?*

The 2018 simulations have exactly the same model set up as in Remy et al. 2024.

*what is the reference for the climatology of DMS emissions you used?*

We now introduce references for both SO2 outgassing and DMS oceanic emissions in Sect. 2.2.

*please detail how biomass burning emissions and SO2 emissions are applied.*

We now include the following sentence : "For biomass burning and SO2 emissions vertical profiles are used representing pyrogenic convection or industrial stack heights, with other emissions applied at the lowest model levels. A diurnal cycle is imposed for isoprene and biomass-burning emissions to capture either photolytic activity or the tropical burning cycle".

*two additional remarks/questions about your simulations: please indicate whether meteorology can differ across simulations, depending on whether aerosol–radiation interactions are activated or not. And what about the gazeous chemistry scheme, is it identical across all your simulations?*

We feel we already address this with the following sentences in Sent 2.2 of the revised manuscript : "The meteorological component is the same between simulations and corresponds to Cy48r1." and "Meteorology is initialized every 24 hours based on ERA-5 reanalysis data i.e. IFS-COMPO is run in cyclic forecast mode". Regarding chemistry : "A 15-minute chemical time-step is used for solving a modified version of CB05 tropospheric chemistry (Williams et al., 2022)."

*line 18 : please write in full "IFS"*

Now expanded in the abstract.

*line 29: I propose to change: "There is also a shift in the size of particles towards the fine mode nitrate away from the coarse mode. " into "There is also a shift in particle size distribution, with nitrate moving from the coarse mode toward the fine mode."*

The sentence has now been updated.

*Line 52 : change "The SIA occurs" into "Secondary inorganic aerosols (SIA) are found throughout the troposphere."*

Now changed to give the correct definition of the acronym.

*Line 53: change "and the concentrations" into "and concentrations"*

We have now modified this sentence.

*line 54: no capital letter to sulphur dioxide, ammonia, nitric acid, particulate matter . And add a ","
before ammnonia.*

We have now removed all capitals before the definition of the chemical species and the grammar has
been corrected.

*line 55: please refer to the exact definitions of PMP1.0, etc... particles smaller than ..*

We have now changed this sentence to : High concentrations of SIA contribute to total Particulate
Matter concentrations that are smaller than various predefined sizes : namely 1.0µm (PM1.0), 2.5µm
(PM2.5) and 10µm (PM10).

*line 62: in the form "OF" NO..*

We have now corrected the grammar.

*line 65: change " SIA being from NH4NO3" in " SIA consists of NH4NO3"*

We have now corrected the text.

*line 66: please reformulate 'increased meteorological instability'*

We now change to " … due to the increased instability of NH4NO3 due to variations in RH and
temperature … "

line 68: most SIA: consider it as singular

We have now changed the tense and checked the manuscript with a grammar checker

*line 68: "At high RH values": please specify a threshold*

Now changed to "At high RH values of 80-100% … "

*line 70: change "the optical properties" into "its optical properties"*

Now changed as suggested.

*line 71: change ' and interactions' into 'and its interactions"*

The microsoft grammar tool shows the proposed change would not improve the grammer

*line 73: change "-1" into "1"*

This is not the value given in the publication which is referenced, where a range of pH values is given
as we have written in the manuscript.

*line 83: change 'whose cumulative oxidation rate is dependent on the prescribed pH in solution' into
'whose overall oxidation rate varies according to the specified pH of the solution.*

We have now replaced "cumulative" with "overall".

*line 85: "for the determination of trends": please be more precise*

We have now expanded this to : ".. determination of long-term trends with respect to resident concentrations … "

*line 89: change 'solution pH' into 'the pH of the solution.*

We have now updated this throughout the manuscript.

*line 93: please reformulate to clarify the sentence: 'The more buffering of solution by NH3, the faster the conversion rate as dictated by the reaction of HSO3- being less than that for SO3 = '*

We have now rephrased to : "The more buffering of solution due to the scavenging and dissolution of NH3, the faster conversion rate is dictated by the rate of reaction of HSO3- being less than that for SO3=".

*line 104: please move the reference Peuch et al 2022 after "CAMS". Similarly, William et al 2022 does not refer to IFS-COMPO. Please reformulate.*

We have now moved/ removed these specific references.

*line 110: change "is the reduction of biases and increase in correlation for aerosol products" into 'was the reduction of biases and increase in correlation of aerosol products with observations'*

We have now changed this sentence in line with the direction of the Topical Editor.

*line 111: please reformulate : 'In that, acidic deposition and N-loading can also be output from the model means such that improving the deposition term via an improved distribution in PM will foster the development of this IFS-COMPO future product. '*

We now use the sentence : "Acidic deposition and nitrogen loading can also be future products from IFS-COMPO which will benefit from the improved simulation and distribution of PM."

*line 118: please change ' This work is complementary to the recent evaluation of the performance of IFS-COMPO Cy48r1 and Cy49r1 and of the impact of using EQSAM4Clim with respect to regional PM2.5 distributions and Aerosol Optical Depth presented in R´emy et al (2024) into 'This work is both parallel and complementary to the recent assessment of global and regional PM2.5 and Aerosol Optical Depth (AOD) distributions presented in R´emy et al (2024).'*

We have now modified this sentence as suggested.

*line 121: please clarify what you mean by 'upgrades to EQSAM4Clim'*

We now modify the sentence to " … application of EQSAM4Clim (Metzger et al., 2024) … " where the details of the recent modifications can be found in this recent publication.

*line 124 "we provide details of the changes": which changes are you referring to?*

The changes on which the manuscript is based i.e. the modifications to CY48r1 -> CY49r1.

*line 125: please rewrite: 'in Sect. 5, we compare annual mean wet deposition fluxes over Europe, the U.S., and Southeast Asia using model outputs and observations...'*

We have modified the sentence accordingly and changed South East to Southeast throughout the manuscript.

*line 129 : change "Model description of IFS-COMPO versions" into "General information on the most recent IFS-COMPO versions"*

We now change the title of Sect. 2 as requested.

*I will stop here with my suggestions for textual revisions. I note that in general the readability of the article could be greatly improved by enhancing the fluency of the English. Numerous tools are available for this purpose. . . Thank you for making all efforts in that direction. For some parts of the text, please rely on the formulations used by Remy, GMD 2024, which appear very clear.*

As suggested by the Topical Editor, we have used Microsoft grammar and fluency checker and corrected whenever there were any issues detected such that we view this as an independent arbitrator which avoids focusing on differences in writing style. We have implemented nearly all editorial comments on wording and phrasing (e.g., lowercase of chemical species, reformulated unclear expressions, redefined regional descriptions and checked the values of the comparisons.
Redundant or ambiguous phrases have been rewritten, including the improved articulation of simulation assumptions (e.g., consistent meteorology, use of aerosol-radiation interactions, and chemical schemes). We ensured that the paper adheres to the requested terminological consistency and enhanced readability. This has resulted in substantial modifications to the text throughout the manuscript. We refrain from the use of e.g. chatGPT for the reformation of our manuscript.

Response to Referee #1

We thank Referee #1 for the positive evaluation of our revised manuscript and for the detailed suggestions to further improve the clarity and quality of the figures and captions. We will submit amended figures for the final version.

Response to Referee #2

We thank Referee #2 for the further review of our manuscript and the constructive suggestions.

*The changes of the wet deposition scheme and analysis. You indicate that the companion paper already describe the changes made to the code. Nevertheless I consider this in not precise enough.*

We have essentially re-written the entire content concerning the update of the wet deposition scheme in Remy et al (2024) in the manuscript such that no further content is available as the entire update has been included. Although changes to e.g. the lifetime of BC particles have no relevance to this

study, and we do not wish to give a historical narrative on the description of aerosol processes in previous IFS cycles, we do now expand on the description of changes made to the other aerosol species, vis;

IFS-COMPO Prep-cy49r1 is built on the previous operational cycle (Cy48r1) and contains 8 distinct aerosol types with multiple bins for size segregation, namely sea salt, desert dust, organic carbon, black carbon, $SO_4^=$, fine and coarse $NO_3^-$, $NH_4^+$ and Secondary Organic Aerosol. For Prep-cy49r1 updates have been made to the aerosol component of the model in the form of modifying both the description and properties of desert dust and sea-salt, the aging of carbonaceous aerosol and an update to the aerosol optics by changing the assumptions used for the $SO_4^=$ aerosol when referencing the lookup table for Mie scattering. This impacts the resident lifetimes, radiative properties and the long-range transport component for each of the aerosol species. The modifications to the description of the aerosol optics has also been implemented, which has been shown to improve the simulation with respect to the Aerosol Optical Depth (AOD) and Ångström exponent when compared against regional observations (Rémy et al., 2024). The gas-phase chemistry, photolysis and dry deposition are identical to that used in Rémy et al. (2024) and as described in Williams et al. (2022). For brevity, and that this study is only concerned with soluble aerosols, we refer the reader to Rémy et al. (2024) for more explicit details related to the other aerosol types.

We do not fully understand the comment related to the analysis as the components on the analysis of precursors, particulates and wet deposition in our paper are described in full in e.g. Sect. 3. We do not analyse AOD here and perform a different analysis as that presented in Rémy et al. (2024) . In response to the first round of reviews we have replotted most figures to aid clarify and homogenize the style across species and measurements.

*The naming of the code version (cy48 vs cy49). I understand your point, even if you do not treat the possible interactions between the composition compartment and the other changes in the IFS model between CY48 and Cy49. Also, I would suggest to at least modulate the title to suggest the ambiguity. Also I would recommand to be more precise for future publications. Concerning the use of Cy49r1 in the present publication I will leave it to the editor to decide.*

We agree that there is no present convention for referring to the version of IFS-COMPO which should be changed for future publications, even though we now change the naming convention in this paper. We also note that we are somewhat going against GMD policy where the naming of model versions should be clear and well defined across publications. Please see the response to the Topical Editor when we have both modified the title and changed the nomenclature of some of the simulations.

*Did you check conservativity in the budget tables?*

Yes the global budget terms are closed and comparable across runs.

*Simulation Cy49r1_NOE4C usage is quite under, it might be useful to exploit it more to be able to make the distinction between different impacts, or to remove it completely.*

Regarding the use of Cy49r1_NOE4C, we agree that this simulation could theoretically be used more extensively to dissect the individual contributions from EQSAM4Clim versus other updates. However, as noted, such a configuration will not be used for any forecasting or in a research context. We therefore chose to limit its use to key comparisons that isolate the effect of EQSAM4Clim and

believe this is sufficient for clarity without overcomplicating the narrative. We do however use the comparisons to differentiate that the precursor gases and SIA types are affected differently when using the EQSAM4Clim methodology. For NHx and NOx SIA we have shown strong experimental evidence that without accounting for the chemical content of aerosols, correct particle concentrations are difficult to achieve.

*Is there a signification in the colours of the modified manuscript?*

There is no specific significance associated with the colour highlighting in the tracked-changes version , where it is purely an artefact of the document editing software used and does not convey any information beyond standard editorial marking.

*Line 349: "No filtering has been applied to these measurements" Does it mean you applied filtering to the other datasets?*

No filtering has been used for any of the datasets for the comparisons shown. We now include this explicit sentence: "No filtering has been applied to any of the observational data used in this study"

*Line 443: "The global budget" You forgot to introduce Table 2.*

We now change the sentence to : "Table 2 provides the global budget terms for $SO_2(g)$, which shows that in addition to primary emission, approximately one third of $SO_2$ in the troposphere comes from the oxidation of DMS by the hydroxyl radical (OH), with DMS originating from biogenic activity in the oceans.

*Line 793: "Europe, US and SE Asia"*

We now replace "SE Asia" with "southeast Asia" throughout the manuscript as requested by the topical editor.

*Lines 995+ and 1080+ are developing the same point.*

We have now made significant changes to the text to remove this.

*Line 1095: Maybe also say that AMON data are used for top right panel.*

We now amend the figure legend to mention the use of AMoN data for the top right panel

*Line 1107: significant decreases → Significant decrease*

We have now amended this sentence.

*Line 1121: "Unfortunately the lack […] can be shown." maybe you should add a paragraph on how selected stations or data, because on the EBAS website when selecting 2018 and ammonium data from the EMEP "Framework" there is a station in South France at Pic du Midi which has data for the aerosol matrix and a few stations including Cyprus and Italy in the pm10 matrix as shown in figure below.*

We thank the referee for the link and example. As noted in our first-round responses, we chose to use a consistent set of stations across all species to enable direct interspecies comparison. Adding stations selectively for individual compounds would complicate this consistency. This is now explicitly stated in Section 3, where we have added the sentence : "We chose sites which monitor both pre-cursor gases and associated SIA simultaneously to ensure valid comparisons."

---

## Editor Decision (ED2)

Dear Author,

I have reviewed your responses to both the second round of reviews and to my comments. At this stage I now have the following comments:

- thank you for a number of improvements in the paper that respond to the reviews

- you indicate that :

  "We have now changed the title to that given by the Topical Editor. However, we do note that GMD has a strict policy with respect to model versions and this introduces some inconsistency between studies performed with the same model version."

  Thank you for your comment, but I remain convinced that the new title and the new simulation names help clarify things for the scientific community.

- I have noticed a number of inaccuracies in your responses in the file with your responses dated 8 August. You claim to have made certain modifications to the text, but these changes are not necessarily reflected in the new version of the text. This indicates a certain lack of thoroughness. This is somehow frustrating and annoying. Here are below my points:

  - I had indicated: " please indicate whether meteorology can differ across simulations, depending on whether aerosol–radiation interactions are activated or not.

    You respond: "The meteorological component is the same between simulations and corresponds to Cy48r1."

    I'm sorry, but your answer is not sufficient: are there or are there not interactions between the gaseous species and/or aerosols with the radiative scheme? If there are, then the meteorology, even if reinitialized every day, is not identical across all simulations, and this should be clearly stated in the text. The meteorological component and the meteorological conditions are two different things.

  - I had indicated: line 29: I propose to change: "There is also a shift in the size of particles towards the fine mode nitrate away from the coarse mode. " into "There is also a shift in particle size distribution, with nitrate moving from the coarse mode toward the fine mode."

    You respond: "The sentence has now been updated."

    I'm sorry, but the sentence has not been updated

  - I had indicated: Line 53: change "and the concentrations" into "and concentrations"

    You respond: "We have now modified this sentence."

    I'm sorry, but the sentence has not been updated

  - I had indicated: line 54: no capital letter to sulphur dioxide, ammonia, nitric acid, particulate matter . And add a "," before ammonia.

    You respond: "We have now removed all capitals before the definition of the chemical species and the grammar has been corrected."

    Not done in the abstract, and "comma " not added

  - I had indicated "line 111: please reformulate : 'In that, acidic deposition and N-loading can also be output from the model means such that improving the deposition term via an improved distribution in PM will foster the development of this IFS-COMPO future product."

    You respond: We now use the sentence : "Acidic deposition and nitrogen loading can also be future products from IFS-COMPO which will benefit from the improved simulation and distribution of PM."

    OK, but you've added a sentence while keeping the original one, which results in a redundancy.

  - I had indicated "line 121: please clarify what you mean by 'upgrades to EQSAM4Clim'

    You respond: "We now modify the sentence to " . . . application of EQSAM4Clim (Metzger et al., 2024) . . . "where the details of the recent modifications can be found in this recent publication.

    I don't see this sentence in the text

– Referee 2 indicated: "Did you check conservativity in the budget tables?"

  You respond: "Yes the global budget terms are closed and comparable across runs." So please indicate that in the text of the paper as well.

• I had already previously made a large number of comments regarding syntax, which in my opinion made the article easier to read. At this stage, I believe that even though the author indicates that they used Microsoft tools with regard to the use of English, many sentences or parts of sentences in the text could be written more correctly, regardless of writing style. I am once again making a few comments regarding the syntax/grammar in this new round of feedback. However, at this stage, I feel it is no longer my responsibility to point out all the remaining imperfections throughout the article.

• I noticed that the line numbering is different in the ATC2 file and in manuscript-version3. I use below the line numbering in the ATC2 file for the following additional comments:

  – line 18: add "(IFS-COMPO)" after Integrated Forecast System-COMPOsition

  – line 20: add ',' after "cycle 49R1"

  – lines 165 and 169: change "CY49R1" into "pre-CY49R1"

  – line 390: remove ',' between "particles" and "are"

  – line 420: I propose to remove ' which is used operationally'.

  – line 437: : you indicate that "update includes adjustments to the below-cloud scavenging parameters" and then "Additionally, a below-cloud scavenging model has been implemented.". Do you mean that there is now in "pre-CY49R1" a separate routine for below-cloud scavenging? Please clarify

  – line 480: please replace "The details of the sensitivity experiments" by "The details of the experiments"

  – line 492: please replace "The emissions adopted" by "Anthropogenic emissions"

  – line 493: remove "with"

  – line 500: remove ',' after ≪ based ≫

  – line 549: you write ' Unfortunately for S. E. Asia measurements of both precursors and SIA at a weekly time frequency ...'. Please rather indicate : "Unfortunately, conducting a similar analysis in Southeast Asia was not possible owing to the unavailability of both precursor and SIA weekly measurements."

  – line 549: '.' after study

  – line 578: the sentence "A direct link...": this last sentence of the paragraph does not logically follow from the previous ones

  – line 586: you write: 'The maps for December show higher mixing ratios towards the East, with a significant contribution from shipping.'. Is indeed shipping responsible for these higher mixing ratios?

  – line 678: "by 1.5%" or by "by 1.4%" as indicated in Table 2?

  – line 699: please clarify 'The yearly mean bias (MB) value decreases by around 25%, with a moderate correlation."

  – line 791 : legend of Fig. 2: please specify what the grey band means, and provide details on the variability criteria used (the latter comment applies also to Fig. 5 and Fig. 8)

  – line 801 : "Although some smaller differences" : smaller than what?

  – line 854: "which would then lower the negative bias" : don't you mean the positive bias?

  – line 856: "means of surface [SO2(g)]": I don't see any SO2(g) presented in Fig. 3; please correct the sentence

  – line 944: "in Fig 4": isn't it rather Fig 3?

- line 931: legend of Fig. 3, and in the legends of all equivalent Figures in the paper, change: "The site locations used are shown in each pane and taken from the EMEP, CASTNET and EANET networks, respectively." into "Site locations and corresponding observed values, taken from the EMEP, CASTNET and EANET networks respectively, are shown in circles".

- all throughout the paper, please remove capital letter for northern, southern, southeast, etc... when they are adjectives, and when they don't refer to an official region

- line 1354 : please indicate in a couple of words what an aerosol dynamical model is

- line 1356 : please clarify: "imposed in pre-CY49R1 occurs"

- line 1363: please clarify: "In contrast to Europe no seasonal decreases occur for any location."

- line 1364: what do you mean by: "are the high across all regions"?

- line 1433: change "op" to "top"

- line 1444: please move the reference (Metzger et al., 2018) when first mentioning "aerosol dynamical model"

- line 1752: "(bottom right panel of Fig. 8)": is this the correct figure?

- line 2321: "towards Iowa are not seen in the measurements", it's not clear to me what you are pointing here

- line 2324: please reformulate "yearly a result of the large positive MB towards the east being moderated by negative in other parts of the U.S"

- line 2328: "This provides " : what does "This" refers to?

- line 2673: "show"

- line 2698 "do reveal"

- line 2704: high values in the observations are not due to missing sources. Please reformulate the sentence.

- line 2724: "and has little effect on the forecasts itself": please explicit what has little effect

- line 2739 "an quantify"?

- line 3033 : please remove 'that have'

- line 3040: in the sentence "For surface [SO2(g)], no significant impact has occurred with respect to the MB for Europe or the U.S. and with little correlation. ": please be more precise for "MB" and what you mean by "and with little correlation."
  More generally, all through the conclusions, precise when you provide numbers what they refer to, that they are for yearly means or for ...

- line 3044: "For the yearly wet deposition of oxidized S, results are mixed with reductions in the MB for Europe and China but increasing markedly for the U. S.". Please correct the grammar

- line 3052: please reformulate "from a reduction in the efficacy of particle"

- line 3061: "persistent" in what sense?

- line 3159 : I understood you did not get weekly observations from EANET. Please correct the legend of the figure

Looking forward to an improved paper.
Regards, Martine Michou

---

## Author Response (AR3)

We thank the Topical Editor for her patience with conducting a final review of our manuscript and we provide a detailed response below.

[1] I had indicated: " please indicate whether meteorology can differ across simulations, depending on whether aerosol–radiation interactions are activated or not. You respond:" The meteorological component is the same between simulations and corresponds to Cy48r1." I'm sorry, but your answer is not sufficient: are there or are there not interactions between the gaseous species and/or aerosols with the radiative scheme? If there are, then the meteorology, even if reinitialized every day, is not identical across all simulations, and this should be clearly stated in the text. The meteorological component and the meteorological conditions are two different things.

*The source code for the meteorological part of IFS used in the simulations corresponds to cycle 48R1, with atmospheric composition updates intended for cycle 49R1. In the simulations shown in this work, the meteorological initial conditions are taken every day at 0h00 from an independent analysis (ERA5), but the 24-hour forecast can indeed be impacted by interaction between gaseous species and aerosols with the radiation scheme. This means that the meteorological forecasts can indeed differ between the simulations, although the analysis is the same. Due to modifications being limited to the inorganic component the influence on meteorology is considered small. We now add in the manuscript: "Still, marginal differences may occur between simulations due to a coupling of gases and aerosol with radiation, which is active by default. "*

[2] I had indicated: line 29: I propose to change: "There is also a shift in the size of particles towards the fine mode nitrate away from the coarse mode. " into "There is also a shift in particle size distribution, with nitrate moving from the coarse mode toward the fine mode." You respond: " The sentence has now been updated." I'm sorry, but the sentence has not been updated

*We have now modified the abstract in line with the requests of the editor.*

[3] I had indicated: Line 53: change "...and the concentrations" into "and concentrations" You respond: "We have now modified this sentence." I'm sorry, but the sentence has not been updated.

*We now remove the word "the".*

[4] line 54: no capital letter to sulphur dioxide, ammonia, nitric acid, particulate matter. And add a "," before ammonia. You respond: "We have now removed all capitals before the definition of the chemical species and the grammar has been corrected." Not done in the abstract, and "comma " not added.

*Capitals are now removed. "We now write: '… sulphur dioxide ($SO_2$), ammonia ($NH_3$) and nitric acid ($HNO_3$).'"*

[5] I had indicated "line 111: please reformulate : 'In that, acidic deposition and N-loading can also be output from the model means such that improving the deposition term via an improved distribution in PM will foster the development of this IFS-COMPO future product." You respond: We now use the sentence: "Acidic deposition and nitrogen loading can also be future products from IFS-COMPO which will benefit from the improved simulation and distribution of PM." OK, but you've added a sentence while keeping the original one, which results in a redundancy.

*We apologize for this oversight. The original sentence has now been removed, and we now modify the new sentence to: Reduced and oxidised nitrogen deposition can also be future CAMS products which will benefit from the improved speciation between gaseous and particulate nitrogen.*

[6] I had indicated "line 121: please clarify what you mean by 'upgrades to EQSAM4Clim' You respond: "We now modify the sentence to " . . . application of EQSAM4Clim (Metzger et al., 2024) . . . "where the details of the recent modifications can be found in this recent publication. I don't see this sentence in the text

We have changed this sentence towards: "The influence on regional wet and dry deposition terms are subsequently evaluated to assess the application of both EQSAM4Clim (Metzger et al, 2024) and the deposition schemes." The actual changes with respect to the inorganic aerosol modeling are described as part of Sec. 2.1

[7] Referee 2 indicated: "Did you check conservativity in the budget tables?" You respond: "Yes the global budget terms are closed and comparable across runs." So please indicate that in the text of the paper as well.

*We now declare this specifically above each of the budget tables.*

[8] line 18: add "(IFS-COMPO)" after Integrated Forecast System-COMPOsition

*Acronym now added.*

[9] line 20: add ',' after "cycle 49R1"

*Comma now added.*

[10] lines 165 and 169: change "CY49R1" into "pre-CY49R1"

*Now corrected.*

[11] line 390: remove ',' between "particles" and "are"

*This comma is now removed.*

[12] line 420: I propose to remove ' which is used operationally'.

*Due to the rapid progress in IFS cycles and that the submission date of this paper was last year, we agree that this text doesn't necessarily provide the most up-to-date information, with the text now being removed.*

[13] line 437: you indicate that "update includes adjustments to the below-cloud scavenging parameters" and then "Additionally, a below-cloud scavenging model has been implemented.". Do you mean that there is now in "pre-CY49R1" a separate routine for below-cloud scavenging? Please clarify

*We apologize for this additional this sentence "additionally a below-cloud scavenging model has been implemented" should not have appeared. The changes to the scavenging parameterization as evaluated here are exactly as described just before this sentence. We now remove this sentence.*

[14] line 480: please replace" The details of the sensitivity experiments" by "The details of the experiments"

*We have now removed the word 'sensitivity'*

[15] line 492: please replace "The emissions adopted" by "Anthropogenic emissions"

*We have changed the text as requested. We have slightly updated the specification of the emissions in the model, to improve the readability:*

*The anthropogenic emissions adopted in these configurations are taken from CAMS_GLOB_ANT v5.3 (Soulie et al., 2024), biogenic emissions taken from the CAMS_GLOB_BIO v3.1 dataset (Sindelarova et al., 2022; http://eccad.aeris-data.fr/) and biomass burning emissions taken from GFAS v1.2 (Kaiser et al., 2012). All emissions are provided at 0.1 x 0.1 resolution at a monthly frequency, except for the biomass burning emissions which are prescribed daily. For biomass burning and anthropogenic emissions from specific sectors vertical profiles are used representing pyrogenic convection or industrial stack heights, with other emissions being applied in the lowest model level.*

[16] line 493: remove "with"

*We have reformulated this sentence and removed the word 'with', see the response to query '15' above.*

[17] line 500: remove ',' after ≪ based ≫

*The comma is now removed.*

[18] line 549: you write ' Unfortunately for S. E. Asia measurements of both precursors and SIA at a weekly time frequency ...'. Please rather indicate: "Unfortunately, conducting a similar analysis in Southeast Asia was not possible owing to the unavailability of both precursor and SIA weekly measurements."

*We now replace the sentence as requested.*

[19] line 549: '.' after study

*We have now updated the grammar.*

[20] line 578: the sentence "A direct link...": this last sentence of the paragraph does not logically follow from the previous ones

*We agree with the topical editor and have now removed this sentence.*

[21] line 586: you write: 'The maps for December show higher mixing ratios towards the East, with a significant contribution from shipping.'. Is indeed shipping responsible for these higher mixing ratios?

*We agree that this description is not clear enough and now modify the sentence accordingly:*
*The maps for July show higher mixing ratios towards the east, with a significant contribution from shipping in the Mediterranean and the English Channel. For December, the highest mixing ratios occur over the continent, most notably Germany and Poland.*

[22] line 678:" by 1.5%" or by" by 1.4%" as indicated in Table 2?

*Now corrected towards 1.4% as in the Table, thank you*

[23] line 699: please clarify 'The yearly mean bias (MB) value decreases by around 25%, with a moderate correlation."

*We now remove this sentence and limit the discussion related to the evaluation of $SO_2(g)$ to lines 352-405.*

[24] line 791: legend of Fig. 2: please specify what the grey band means, and provide details on the variability criteria used (the latter comment applies also to Fig. 5 and Fig. 8)

*Fig. 2, 5, 8 (grey band): "As requested, we now specify that the grey band denotes the station range (min–max) of observed weekly means at each time. This is applied consistently in Figs. 2, 5, and 8."*

[25] line 801 : "Although some smaller differences" : smaller than what?

*We now modify the sentence: Although some small differences in the $SO_2(g)$ weekly values occur between simulations, there is no improvement in pre-Cy49r1 with respect to the weekly MB. The positive summertime MB for $SO_2$ implies that there is either an overestimate in the regional source terms or underestimate in the depositional loss term (c.f. $SO_4^{2-}$] below).*

[26] line 854: "which would then lower the negative bias" : don't you mean the positive bias?

*We have now modified this sentence into "…. which would then lower the positive bias seen for [SO2(g)]."*

[27] line 856: "means of surface [SO2(g)]": I don't see any SO2(g) presented in Fig. 3; please correct the sentence.

*We have now corrected this sentence and removed reference to [SO2(g)].*

[28] line 944: "in Fig 4": isn't it rather Fig 3?

*Thanks for highlighting this typo. We now correct the table 3 legend to refer to Fig 3.*

[29] line 931: legend of Fig. 3, and in the legends of all equivalent Figures in the paper, change: "The site locations used are shown in each pane and taken from the EMEP, CASTNET and EANET networks, respectively." into "Site locations and corresponding observed values, taken from the EMEP, CASTNET and EANET networks respectively, are shown in circles".

*The figure legend has been replaced, also for Figs. 6, 9, 10, 11 and 12.*

[30] all throughout the paper, please remove capital letter for northern, southern, southeast, etc... when they are adjectives, and when they don't refer to an official region

*We homogenised the naming convention (e.g., 'Southern U.S.' → 'southern U.S.'; Northwest → 'northwest U.S.'), harmonized all occurrences to 'pre-Cy49r1' (and 'Cy48r1'), including section headers and table labels. We retain Southeast Asia as this is a named region in the manuscript.*

[31] line 1354: please indicate in a couple of words what an aerosol dynamical model is

*"Here, an aerosol dynamical model refers to a microphysical framework that prognoses particle number/size distributions (via condensational growth, coagulation, nucleation and inter-mode transfer), thus enabling kinetically limited gas-to-particle uptake and condensation processes beyond the instantaneous thermodynamic equilibrium assumed by EQSAM4Clim."*

[32] line 1356 : please clarify: "imposed in pre-CY49R1 occurs"

*We have now modified the text to: "For December, the relative differences in pre-Cy49R1 compared to Cy48r1 show increases in $HNO_3(g)$ at latitudes below 60°N, together with decreases of between 25-75% at latitudes higher than 60°N, i.e. at lower temperatures, and under a relatively low $NO_x$ environment."*

[33] line 1363: please clarify:" In contrast to Europe no seasonal decreases occur for any location."

*To better clarify we now change this sentence into:"In contrast to Europe no significant decreases in [HNO$_3$(g)] occur for any location or season."*

[34] line 1364: what do you mean by: "are the high across all regions"?
*We change the formulation of this sentence into: Finally, in comparison to the other regions the surface [HNO$_3$(g)] are highest for Southeast Asia. Maximal values reach the order of 4-5 ppb towards the eastern coast (July) and central regions (c.f. December).*

[35] line 1433: change "op" to "top"
*We have now corrected this typo.*

[36] line 1444: please move the reference (Metzger et al., 2018) when first mentioning "aerosol dynamical model"
*We have now moved the reference to the first mention of coupling to an aerosol dynamic model.*

[37] line 1752: "(bottom right panel of Fig. 8)": is this the correct figure?
*Figure 8 pertains to the comparison of both HNO3 and NO3- against EMEP and CASTNET weekly values such that we do reference the correct figure here.*

[38] line 2321:" towards Iowa are not seen in the measurements", it's not clear to me what you are pointing here
*We now replace Iowa with central U.S. to indicate the location of maximum concentrations.*

[39] line 2324: please reformulate "yearly a result of the large positive MB towards the east being moderated by negative in other parts of the U.S"
*We have now reformulated the sentence as requested.*

[40] line 2328: "This provides " : what does "This" refers to?
*We now replace the beginning of the sentence: "The yearly deposition values range between … "*

[41] line 2673: "show"
*We now modify the sentence to: "The global chemical budget terms provided in Table 6 reveal that there is an increase in the gas-phase production term for HNO3, ..."*

[42] line 2698 "do reveal"
*We now correct the grammar of the sentence.*

[43] line 2704: high values in the observations are not due to missing sources. Please reformulate the sentence.
*We agree that emphasis should not be put on missing source terms and we now remove the reference to "missing local emission sources".*

[44] line 2724: "and has little effect on the forecasts itself": please explicit what has little effect
*We now shorten the sentence to remove the implied effect on chemical forecasts.*

[45] line 2739 "an quantify"?
*Now corrected to "... and quantify .. "*

[46] line 3033 : please remove 'that have'
*We have now reformulated the sentence as follows: The gaseous precursors, SIA surface concentrations and wet deposition totals for the three dominant global source regions (Europe, the U.S., and southeast Asia) have been evaluated against weekly/yearly observational composites for the year 2018."*

[47] line 3040: in the sentence "For surface [SO2(g)], no significant impact has occurred with respect to the MB for Europe or the U.S. and with little correlation. ": please be more precise for "MB" and what you mean by "and with little correlation." More generally, all through the conclusions, precise when you provide numbers what they refer to, that they are for yearly means or for ...
*We have now changed the sentence to: "For surface [SO$_2$(g)], no significant impact has occurred with respect to the mean bias over Europe and the U.S.. Also the correlation in the weekly variability improved only marginally"*

[48] line 3044: "For the yearly wet deposition of oxidized S, results are mixed with reductions in the MB for Europe and China but increasing markedly for the U. S.". Please correct the grammar
*We have now reformulated the sentence to: Reductions in the MB are found for Europe and China, whilst for the U. S. the MB increases markedly.*

[49] line 3052: please reformulate "from a reduction in the efficacy of particle"
*We have now reformulated this sentence towards: "For [NH$_4^+$], the application of EQSAM4Clim significantly reduces the associated MB against observational yearly means by approximately 45% for all three global regions. This is explained by a reduction in the efficacy of particle formation, especially during summertime."*

[50] line 3061: "persistent" in what sense?
*Persistent with respect to the length of time for which the bias exists. We have now reformulated the sentence to: "For HNO$_3$(g) in Europe and the U.S., there is a persistent negative bias during the summertime which is changed to a significant positive bias due to a larger fraction remaining in the gas phase. For the U. S., there is a degradation with a relatively low bias changing to a large positive bias for the entire year, which indicates an incomplete process coupling when considering mineral cations without aerosol dynamics. "*

[51] line 3159 : I understood you did not get weekly observations from EANET. Please correct the legend of the figure
*We now remove the word weekly from the figure legend.*

---

## Editor Decision (ED3)

Dear Author,

I have reviewed your responses and the new version of your paper. Thank you for a large number of evolutions in the paper that respond to the majority of my comments.

At this stage I still have a few remaining comments.

Note that the paragraph numbering below refers to the paragraph numbering in your response, and the line numbering refers to your tracked changes file.

I'm still surprised that there are once again differences between what is written in your answer sheet and what is actually in the text of the article.

- para 9 line 20 : ',' not added after 'cycle 49R1'

- para 11 line 187 : ',' not removed before 'are outlined'

- para 25 line 486: add 'an' before 'underestimate' and line 487 remove bracket after $SO_4^{2-}$

- line 1512 : remove "." before "which"

- Table 3: the MB, RMSE, and Pearson coeff for Europe/cy48r1 are now mixed

- in the conclusion, you sometimes use the terms 'mean bias' and sometimes 'MB'. I recommend using 'mean bias' throughout the conclusion

I leave it to the GMD publication team to ensure that my latest comments are taken into account. Regards, Martine Michou